# Cholesterol removal improves performance of a model biomimetic system to co-deliver a photothermal agent and a STING agonist for cancer immunotherapy

Lin Li[1], Mengxing Zhang[2], Jing Li[1], Tiantian Liu[2], Qixue Bao[1], Xi Li[1], Jiaying Long[3], Leyao Fu[1], Zhirong Zhang[3], Shiqi Huang[2], Zhenmi Liu[1] ✉ & Ling Zhang [ID][1,2] ✉

Biological membranes often play important functional roles in biomimetic drug delivery systems. We discover that the circulation time and targeting capability of biological membrane coated nanovehicles can be significantly improved by reducing cholesterol level in the coating membrane. A proof-of-concept system using cholesterol-reduced and PD-1-overexpressed T cell membrane to deliver a photothermal agent and a STING agonist is thus fabricated. Comparing with normal membrane, this engineered membrane increases tumor accumulation by ~2-fold. In a melanoma model in male mice, tumors are eliminated with no recurrence in >80% mice after intravenous injection and laser irradiation; while in a colon cancer model in male mice, ~40% mice are cured without laser irradiation. Data suggest that the engineered membranes escape immune surveillance to avoid blood clearance while keeping functional surface molecules exposed. In summary, we develop a simple, effective, safe and widely-applicable biological membrane modification strategy. This "subtractive" strategy displays some advantages and is worth further development.

As an important type of biomimetic drug delivery system, cell membrane-coated nanoparticles have been widely studied in recent years and displayed various advantages in targeted delivery of drugs[1]. However, many of these works were conducted in nude mice which have a compromised immune system, and the efficiency of these delivery systems in immunocompetent mice remains to be improved[2–4]. As these membranes usually act as interface between in vivo environment and the delivery vehicle, they largely determine the performance of the delivery system[5]. However, the sizes, morphology, and proteins of cell membranes used in delivery system are different from those of normal cells in the blood, and it is difficult for them to avoid being recognized as foreign objects and cleared by phagocytes[6,7]. For example, in the blood flow, complement proteins could deposit on the surface of nanoparticles in a process called opsonization[8,9]. These proteins then prime the particle for removal by immune cells like phagocytes in the blood, which could lead to diminished targeted accumulation and adverse effects[10]. Hence, reducing the adsorption and activation of complements on the surface of cell membrane-coated nanoparticles may significantly improve their performance[11].

Various membrane engineering methods have been applied and tested with some success, such as surface polyethylene glycol

[1]Institute of Systems Epidemiology, West China School of Public Health and West China Fourth Hospital, Sichuan University, Chengdu 610041, China. [2]Med-X center for Materials, College of Polymer Science and Engineering, Sichuan University, Chengdu 610065, China. [3]Key Laboratory of Drug Targeting and Drug Delivery Systems of Ministry of Education, West China School of Pharmacy, Sichuan University, Chengdu 610065, China. ✉e-mail: liuzhenmi1983@hotmail.com; zhangling83@scu.edu.cn

modification (i.e., PEGlaytion), shape modification, or combined usage with complement inhibitors[2,8,9,12,13]. Among these, PEGlaytion is the most widely used nanomedicine modification method and has become an essential part of many successful products, such as the BNT162b2 vaccine developed by Pfizer-BioNTech. However, these methods mostly rely on adding functional molecules to the nanomedicine, which is more difficult to apply on biological membranes as they possess highly heterogenetic surface motifs and are intolerable towards many reaction conditions[14]. So far, there is still concern that PEG may cause adverse effects in vivo, which may activate complements in the blood and induce adverse injection reactions[8]. Besides, when long chains of PEG or other molecules are added to membrane surface, the targeted ligand on the membrane may be masked and suffer efficient loss[2]. Thus, alternative strategy for biological membrane engineering is highly demanded to enhance their anti-clearance efficiency and improve their clinical translation probability[15].

Liposomes share a similar basic surface structure with biological membranes. We noticed that the elimination of liposomes from blood circulation was quickened by increasing the cholesterol content in liposomes, with unclear mechanism[16]. Besides keeping fluidity of membranes and reducing membrane permeability, it is also debated that cholesterol molecules could be attracted to high-melting point lipids and proteins to form "lipid rafts" on membranes[17]. It is possible that the presence of cholesterol increases the rate of protein interaction with membrane components[17]. Nevertheless, it appears worth testing whether reducing cholesterol content in biological membrane could improve their performance. (2-hydroxypropyl)-β-cyclodextrin (β-CD) is a biocompatible compound widely used as a drug solubilizer in the clinic and in various drug delivery systems, and has been reported to scavenge cholesterol through host-guest interaction[18].

As a proof-of-concept, we planned to apply this cholesterol-removing method in a biological membrane-based bionic system aiming to treat melanoma which has high metastasis and recurrence rate as well as other types of tumors[19]. Immunotherapy, especially immune checkpoint blockade (ICB) therapy, has got some success in treating a portion of cancer patients[20,21]. However, current ICB therapies are inefficient for a majority of cancer patients for various reasons, such as low programmed cell death-Ligand 1 (PD-L1) expression level[22–26]. Therefore, combined treatments with ICB have been proposed and tested to improve the outcome[27]. The cytosolic DNA sensing STING pathway which plays a key role in initiating anti-tumor immunity, is one such treatment target[28]. It was found that STING agonists cGAMP could activate STING and induce the secretion of interferon (IFNs) to trigger anti-tumor immune responses, and the released IFNs in turn upregulate PD-L1 in tumor cells, which then enhance the therapeutic effect of ICB[29,30]. However, cGAMP is unstable in vivo[31], although the recently reported analog SR-717 has improved stability[29], it still lacks tumor-targeting ability and may induce systemic toxicity. Thus, a tumor-targeting delivery system is needed to improve the efficiency of STING agonist[32].

It was reported that T cell membrane-coated nanoparticles could naturally bind to tumor cells, and the programmed death-1 (PD-1) on T cell membrane-targeted and blocked PD-L1 on tumor cells, offering ICB effect[33]. The phenolic-metal nanoparticle based on quercetin and iron (i.e., QFN) was reported to have good drug-loading capacity and photothermal conversion efficiency[19,34], which could be used to load SR-717.

In this work, the β-CD-treated membrane sourced from T cells overexpressing PD-1 is used to coat SR-717-loaded QFN. The system is named CISP, which denotes low-cholesterol membrane coat ICB agent, STING agonist, and photothermal agent. Membranes with lower cholesterol content reduce the uptake of CISP by monocytes in the blood by ~50%, while maintaining its tumor-targeting capability. As a result, CISP delivers SR-717 to tumor sites to activate the local STING pathway, while PD-1 on CISP surface blocks PD-L1 in tumor cells. This method is different from most surface modification strategies, as it removes molecules rather than adding molecules to the object surface such as PEGlaytion, which is easier to apply and potentially safer.

## Results

### Preparation and characterization of CISP

First, mouse T cell line CTLL2 overexpressing PD-1 was engineered (CTLL2-PD1) (Fig. 1a, b). Then, the living CTLL2-PD1 cells were treated with (2-hydroxypropyl)-β-cyclodextrin (β-CD) before the cell membrane was extracted (Fig. 1c). Results showed that the content of cholesterol in cell membrane treated with β-CD (20 mM) was reduced to ~20% of the original level (Fig. 1d). The extracted cell membrane was then coated on quercetin-ferrum nanoparticles (QFN) which encapsulated SR-717 (e.g., $QFN_{717}$)[19]. For experimental purposes, several different bionic nanosystems were prepared: normal CTLL2 cell membrane-coated $QFN_{717}$ (nCSP, denoting normal-Cholesterol cell membrane-coated STING agonist, and Photothermal agent), normal CTLL2-PD1 cell membrane-coated $QFN_{717}$ (nCISP, with added ICB agent) and cholesterol-deficient CTLL2-PD1 cell membrane-coated $QFN_{717}$ (CISP, as described above). Transmission electron microscope (TEM) images showed membrane-like structures on the surface of nCSP, nCISP, CISP (Fig. 1e and Supplementary Fig. 1), and >80% $QFN_{717}$ were coated with the cell membrane in both nCISP and CISP (Supplementary Fig. 2). Fluorescence images also showed that the cell membrane signal co-localized with $QFN_{717}$ core, suggesting that the cell membrane was successfully coated on $QFN_{717}$ (Fig. 1f).

The hydration particle sizes of $QFN_{717}$, nCSP, nCISP, and CISP were all ~150 nm, and the zeta potential was approximately -10 mV (Supplementary Fig. 3a–c). The encapsulation rate of SR-717 in $QFN_{717}$, nCSP, nCISP, and CISP was ~90%, while it was ~70% in PLAG nanoparticles loaded with SR-717 (e.g., $PLGA_{717}$), indicating QFN performed well to load SR-717 (Supplementary Fig. 3d). Western blot results showed that the PD-1 level of nCISP and CISP was significantly higher than that of nCSP (Fig. 1g). SDS-PAGE and proteomics results illustrated that the membrane protein composition of nCISP and CISP were similar (Supplementary Figs. 4, 5). In addition, QFN, $QFN_{717}$, nCSP, nCISP, and CISP all showed similar photothermal effects with 808 nm laser irradiation (Supplementary Fig. 6 and Fig. 1h), suggesting that membrane coating and SR-717 loading did not affect the photothermal conversion efficiency of QFN[14,19,35].

### CISP blocked the elevated PD-L1 induced by SR-717 in tumor cells

Compared with nCSP and $QFN_{717}$, nCISP and CISP showed stronger targeting ability to B16F10 cells in vitro (Fig. 2a–d). When PD-L1 expression was knocked out in B16F10 cells (B16F10[PD-L1 KO]), its uptake of CISP was significantly reduced, and the ratio of tumor cells with nanoparticles dropped from ~45% to ~25% (Supplementary Fig. 7a, b). Hence, the binding of PD-1 and PD-L1 played an important role in the tumor targeting of PD-1-expressing membrane.

In fact, both CISP and nCISP could bind to PD-L1 on B16F10 cells and transport it to lysosomes (Fig. 2e), which was consistent with the previous report[33]. Also consistent with published data is that free SR-717 increased PD-L1 expression on tumor cells in this study (Fig. 2f)[29]. Here, both nCISP and CISP could reduce the expression of PD-L1 on tumor cells even with loaded SR-717 (Fig. 2f), indicating that the overexpressed PD-1 on nCISP and CISP blocked the PD-L1 induced by SR-717. Compared with nCSP, CISP, and nCISP enhanced the killing effect of CD8 T cells against tumor cells in vitro (Fig. 2g, h and Supplementary Fig. 8a, b)[36]. Additionally, CISP and nCISP showed similar cytotoxicity to B16F10 cells under laser irradiation (Supplementary Fig. 9a, b).

Taken together, these results suggest that CISP could enhance the tumor targeting of nanoparticles by binding to PD-L1 on tumor cells, and PD-L1 is delivered to lysosomes along with CISP, thereby reducing

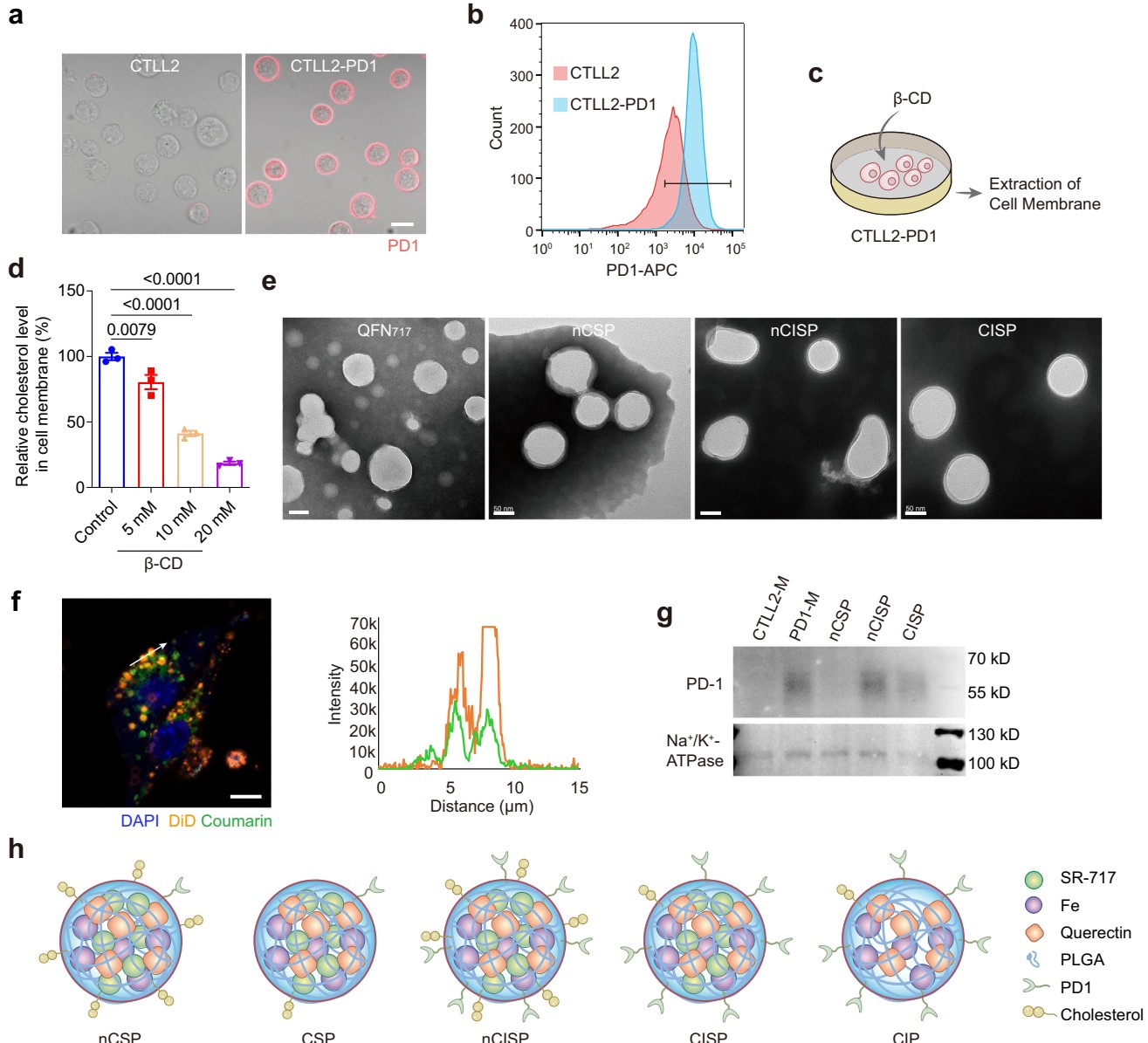

**Fig. 1 | Preparation and characterization of CISP. a** Representative fluorescence images of PD-1-APC on cell membrane (*n* = 5 technical replicates). Scale bar = 10 μm. **b** The expression of PD-1 on cell membrane detected by flow cytometry (*n* = 3 technical replicates). **c** Schematic illustration showing the process to prepare low-cholesterol membrane. **d** Relative level of cholesterol in cell membrane after cells incubated with different concentration of (2-hydroxypropyl)-β-cyclodextrin (β-CD) for 0.5 h (*n* = 3 independent samples). **e** Representative TEM images of QFN$_{717}$, nCSP, nCISP and CISP (*n* = 3 independent samples). Scale bar = 50 nm. **f** The localization of cell membrane (marked with DiD) and QFN$_{717}$ core (marked with coumarin) in CISP, and the fluorescence intensity of DiD and coumarin on the arrow line. B16F10 cells were incubated with CISP for 4 h (*n* = 3 independent samples). Scale bar = 5 μm. **g** Western blot analysis of PD-1 and Na$^+$/K$^+$-ATPase in membrane of

CTLL2 (CTLL2-M), membrane of CTLL2-PD1 (PD1-M), nCSP, nCISP and CISP. **h** Schematic illustration showing nCSP, CSP, nCISP, CISP and CIP used in this study. Data represent mean ± SEM (**d**). Statistical significance was determined by one-way ANOVA test in **d**, and it was two-sided and adjustments were made for multiple comparisons. The experiments for **a**, **b**, **d**–**f**, **g** were repeated three times independently with similar results. Source data are provided as a Source Data file. MFI mean fluorescence intensity, QFN Quercetin-ferrum nanoparticles, QFN$_{717}$ QFN loaded with SR-717, nCSP normal-Cholesterol cell membrane-coated QFN$_{717}$, CSP low-cholesterol membrane-coated QFN$_{717}$, nCISP normal-Cholesterol cell membrane-coated ICB agent and QFN$_{717}$, CISP low-cholesterol membrane-coated ICB agent and QFN$_{717}$, CIP low-cholesterol membrane-coated ICB agent and QFN.

PD-L1 on tumor cells membrane induced by SR-717, which enhanced the activity of T cells to kill B16F10 cells[33]. Moreover, CISP and nCISP showed similar tumor-targeting ability in vitro, indicating that reducing the cholesterol content of T-cell membrane to an adequate level did not affect the targeted function of CISP in vitro.

## CISP efficiently targeted tumors in vivo

Next, the membrane-coated nanomedicines were tested in tumor-bearing mice models. Here, B16F10 cells (7 × 10$^5$ cells/mouse) were

inoculated subcutaneously in C57BL/6 mice, and mice were intravenously injected with different nanomedicines at 8 days post-cell inoculation. Biodistribution assay showed that the tumor accumulation and penetration of CISP was significantly higher than that of nCSP, nCISP, and non-coated QFN$_{717}$ (Fig. 3a, b and Supplementary Fig. 10a, b). More CISP could be detected in dendritic cells (DCs) in tumors (Supplementary Fig. 11a, b). With laser irradiation, the tumors in CISP group experienced rapid temperature rise and reached a higher degree than other groups (Supplementary Fig. 12). No obvious tumor targeting

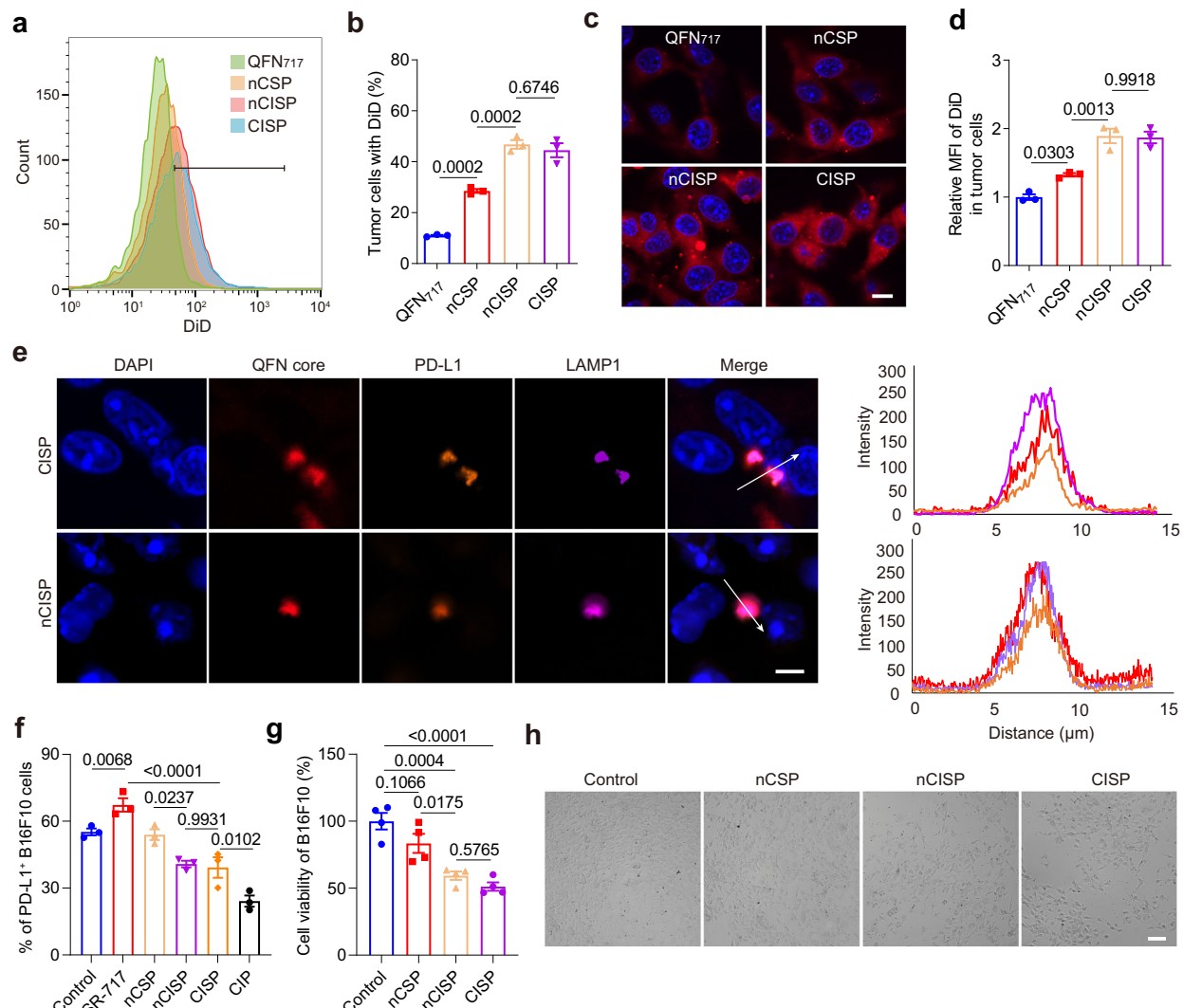

**Fig. 2 | CISP blocked the elevated PD-L1 in tumor cells induced by STING agonists in vitro. a, b** The uptake of QFN717, nCSP, nCISP, and CISP (marked with DiD) by B16F10 cells detected by flow cytometry (**a**), and the proportion of cells with DiD (**b**) (n = 3 independent samples). **c, d** Representative fluorescence images of B16F10 cells treated with QFN717, nCSP, nCISP, and CISP (marked with DiD, red) (**c**), and the relative mean fluorescence intensity of DiD in cells (**d**) (n = 3 independent samples). Scale bar = 10 μm. **e** Representative images of QFN core, PD-L1, and LAMP1 in B16F10 cells, and the fluorescence intensity on the arrow line (n = 3 independent samples). Scale bar = 5 μm. **f** Membrane PD-L1 level detected by flow cytometry (n = 3 independent samples). **g, h** Cell viability of B16F10 detected with CCK-8 kit (**g**), and representative images of B16F10 cells (**h**). B16F10 cells were treated with

nCSP, nCISP, and CISP, and then incubated with activated CD8+ T cells for 48 h (n = 4 independent samples). Scale bar = 100 μm. Data represent mean ± SEM (**b, d, f, g**). Statistical significance was determined by a one-way ANOVA test in **b, d, f, g** and it was two-sided and adjustments were made for multiple comparisons. The experiments for **a, b, e–h** were repeated three times independently with similar results. Source data are provided as a Source Data file. MFI mean fluorescence intensity, QFN Quercetin-ferrum nanoparticles, QFN717 QFN loaded with SR-717, nCSP normal-Cholesterol cell membrane-coated QFN717, CSP low-cholesterol membrane-coated QFN717, nCISP normal-Cholesterol cell membrane-coated ICB agent and QFN717, CISP low-cholesterol membrane-coated ICB agent and QFN717, CIP low-cholesterol membrane-coated ICB agent and QFN.

ability was observed in nCISP group even though it worked very well in vitro (Fig. 2a, b and Fig. 3a, b). Additionally, there was no significant difference in the biodistribution of nCISP and CISP in the major organs (heart, liver, spleen, lung, and kidney) (Supplementary Figs. 13a, b, 14).

CSP (denotes low-cholesterol membrane-coated STING agonist and Photothermal agent) was fabricated and used as a control as well (Fig. 1h and Supplementary Fig. 15). Results showed that CSP had weaker tumor targeting capability than CISP (Fig. 3c, d). CISP could significantly increase the level of SR-717 in tumors, and CSP delivered less SR-717 to tumors (Fig. 3e). It appears that, consistent with in vitro experiments, CISP bound PD-L1 on tumor cells and brought it to lysosomes, while CSP failed to show such ability (Fig. 2e and Fig. 3f). These results indicated that cholesterol-deficient membrane overexpressed PD-1 could promote the tumor-targeting ability of nanoparticles in vivo.

Cell membrane vesicle isolated from CTLL2-PD1 cells treated with β-CD also showed improved tumor targeting ability, indicating that this strategy is also applicable to cell membrane vesicle and the relatively stiffer inner core is not solely responsible for their clearance (Supplementary Fig. 16a–c). Additionally, high concentration of β-CD (30 mM) was found to reduce the viability of CTLL2-PD1 cells after incubation (Supplementary Fig. 17), and the encapsulation efficiency decreased to ~40% when QFN717 was coated with membrane source from T cells treated with β-CD (30 mM) (Supplementary Figs. 1, 2). CISP (30 mM) failed to demonstrate the excellent tumor-targeting effect of CISP (20 mM) (Fig. 3c). Hence, it appears that there is an optimal concentration of β-CD to be used in this strategy.

For further comparison, we PEGlayted the CTLL2-PD1 cell membranes (PEG-nCISP), hoping to improve the tumor targeting of nCISP by

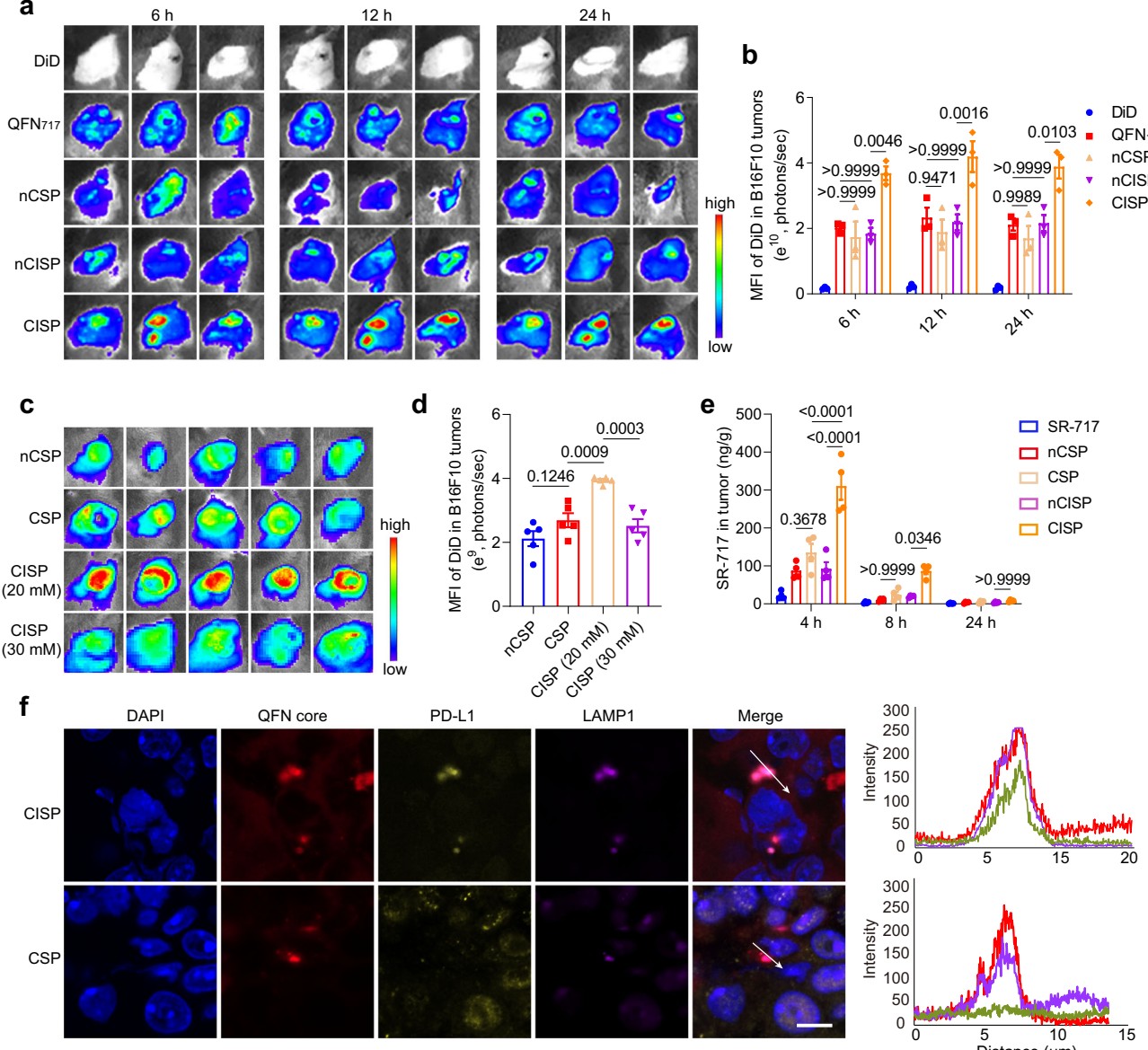

**Fig. 3 | CISP targeted to tumors in vivo. a**, **b** Fluorescence images of B16F10 tumors in mice treated with free DiD, QFN_{717}, nCSP, nCISP, and CISP (**a**) and the mean fluorescence intensity of DiD in tumors (**b**) (*n* = 3 mice). **c**, **d** Fluorescence images of B16F10 tumors at 6 h post injection of nCSP, nCISP (20 mM), and nCISP (30 mM) (**c**), and the mean fluorescence intensity of DiD in tumors (**d**) (*n* = 5 mice). **e** Concentration of SR-717 in tumors detected by LC-MS/MS (*n* = 4 mice). **f** Representative images of QFN core, PD-L1, and LAMP1 in tumors and the fluorescence intensity on the arrow line (*n* = 3 independent samples). Scale bar = 10 μm. Data represent mean ± SEM (**b**, **d**, **e**). Statistical significance was determined by one-way ANOVA test in **b**, **d**, **e**, and it was two-sided and adjustments were made for multiple comparisons. The experiments for **a**, **f** were repeated three times independently with similar results. Source data are provided as a Source Data file. MFI mean fluorescence intensity, QFN Quercetin-ferrum nanoparticles, QFN_{717} QFN loaded with SR-717, nCSP normal-Cholesterol cell membrane-coated QFN_{717}, CSP low-cholesterol membrane-coated QFN_{717}, nCISP normal-Cholesterol cell membrane-coated ICB agent and QFN_{717}, CISP low-cholesterol membrane-coated ICB agent and QFN_{717}, CISP (20 mM) low-cholesterol membrane (treated with 20 mM β-CD) coated ICB agent and QFN_{717}, CISP (30 mM) low-cholesterol membrane (treated with 30 mM β-CD) coated ICB agent and QFN_{717}.

prolonging its blood circulation time[2,37]. Although the uptake of PEG-nCISP by phagocytes was strongly reduced, the uptake of nanoparticles by B16F10 cells also dropped significantly (Supplementary Fig. 18a–d). As a result, PEG-nCISP failed to show tumor targeting ability in mice either (Supplementary Fig. 18e, f). Thus, at least in this design, the cholesterol removal strategy performed better than PEGlaytion.

We also evaluated the performance of low-cholesterol tumor cell membrane (using B16F10 cells), as published data showed that tumor cell membrane-coated nanoparticles could target the same tumor (though many of these works were conducted in nude mice which have compromised immune system)[2]. We again found that the low-

cholesterol B16F10 cell membrane-coated PLGA (e.g., CBM) outperformed their normal counterpart (e.g., nCBM) (Supplementary Fig. 19a, b). Macrophage membrane was reported to exhibit inflammatory site targeting capability[38]. Thus, we constructed a macrophage membrane-wrapped PLGA nanosystem. We again found that the low-cholesterol RAW264.7 cell membrane-coated PLGA (e.g., CRM) possessed better inflammation site accumulation than particles coated with normal membrane (e.g., nCRM) and naked PLGA nanoparticles in an arthritis model (Supplementary Fig. 20a–d). These results imply that the cholesterol removal strategy is also applicable to other sources of biological membranes and other disease models. Interestingly, low-

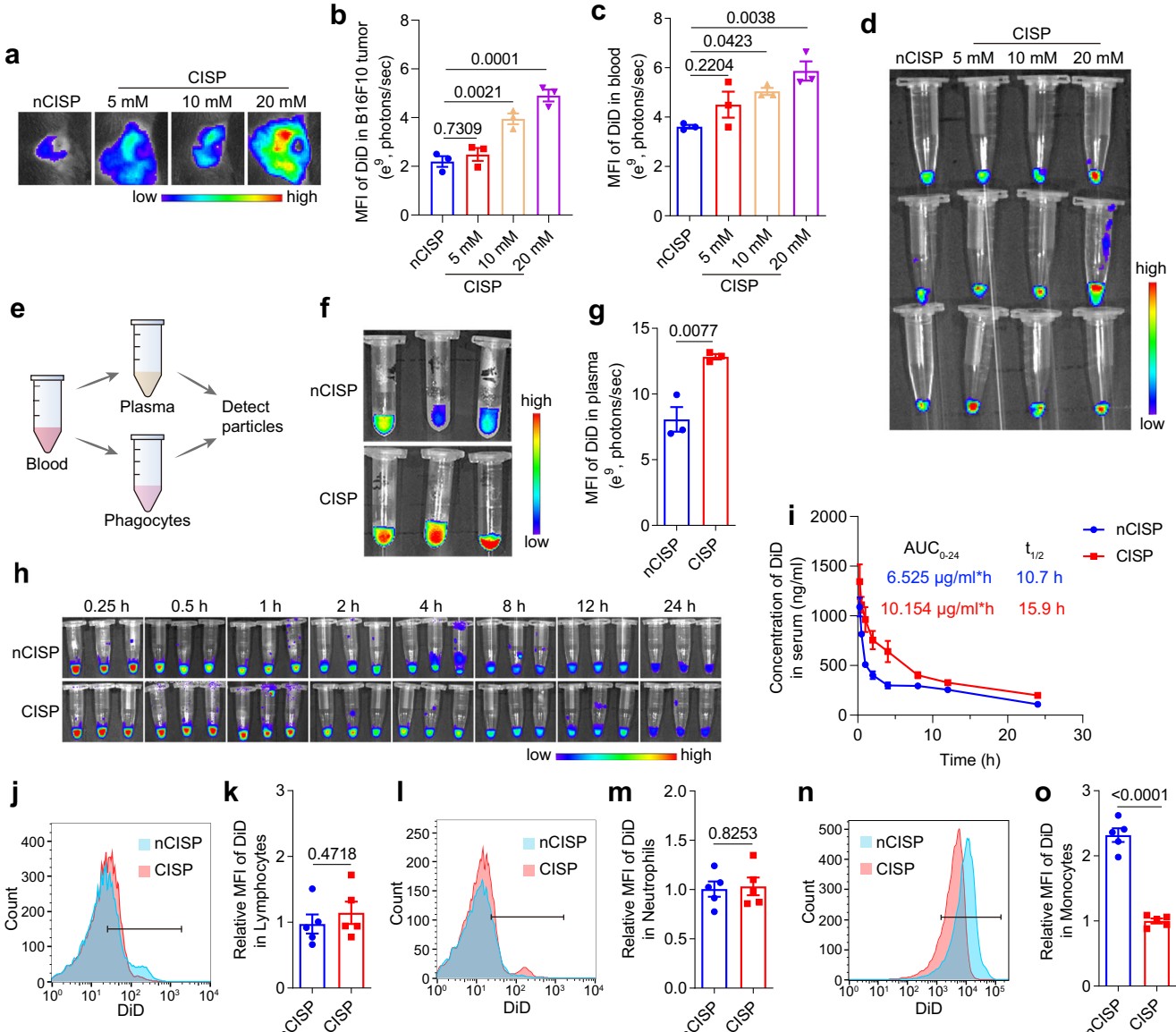

**Fig. 4 | The cholesterol-deficient cell membrane inhibited the clearance of CISP by monocytes in the blood. a, b** Fluorescence images of tumors in mice treated with CISP (**a**) and the mean fluorescence intensity of DiD in tumors (**b**) ($n = 3$ mice). Membrane of CISP were sourced from CTLL2-PD1 that were treated with different concentration of β-CD. **c, d** mean fluorescence intensity of DiD in blood (**c**) and fluorescence images of blood at 4 h post injection (**d**) ($n = 3$ mice). **e** Schematic illustration showing the process to detect nanoparticles in blood. **f, g** Fluorescence images of plasma at 0.5 h post injection (**f**) and the mean fluorescence intensity of DiD in plasma (**g**) ($n = 3$ mice). **h** Fluorescence images of serum in mice ($n = 3$ mice). **i** Concentration of DiD in serum detected by fluorescence spectrophotometer ($n = 3$ mice). **j, k** Uptake of nCISP and CISP by lymphocytes in the blood detected by flow cytometry at 0.5 h post injection (**j**) and the relative mean fluorescence intensity of DiD in lymphocytes (**k**) ($n = 5$ mice). **l, m** Uptake of nCISP and CISP by

neutrophils in the blood detected by flow cytometry (**l**) and the relative mean fluorescence intensity of DiD in neutrophils (**m**) ($n = 5$ mice). **n, o** Uptake of nCISP and CISP by monocytes in the blood detected by flow cytometry (**n**) and the relative mean fluorescence intensity of DiD in monocytes (**o**) ($n = 5$ mice). Data represent mean ± SEM (**b, c, g, i, k, m, o**). Statistical significance was determined by one-way ANOVA test in **b, c**, and it was two-sided and adjustments were made for multiple comparisons. Student's two-sided *t* test was used for the statistical analysis in **g, k, m, o**. The experiments for **c, d, f, g, n, o** were repeated three times independently with similar results. Source data are provided as a Source Data file. MFI mean fluorescence intensity, QFN Quercetin-ferrum nanoparticles, QFN₇₁₇ QFN loaded with SR-717, nCISP normal-Cholesterol cell membrane-coated ICB agent and QFN₇₁₇, CISP low-cholesterol membrane-coated ICB agent and QFN₇₁₇.

cholesterol membrane sourced from CTLL2-PD1 cells failed to show improved targeting ability in the arthritis model (Supplementary Fig. 21a, b), indicating that the functionality of the engineered membrane itself is essential, which is consistent with the results mentioned above (Fig. 3c).

## The cholesterol-deficient cell membrane inhibited the clearance of CISP by monocytes in the blood

Next, we investigated why cholesterol removal improved in vivo performance of biological membranes while no significant difference was

identified in vitro. One of the most prominent differences between in vivo and in vitro environment is the complex components in the blood flow[39]. Considering that nCISP and CISP showed similar tumor targeting ability to tumor cells in vitro (Fig. 2a–d), we next mainly focused on the situation of nCISP and CISP in the blood flow. Fluorescence images show that the tumor targeting ability of CISP was reversely correlated with cholesterol level in the membrane (Fig. 1d and Fig. 4a, b). The circulation time of CISP in blood was longer than that of nCISP, and CISP with lower cholesterol content had higher

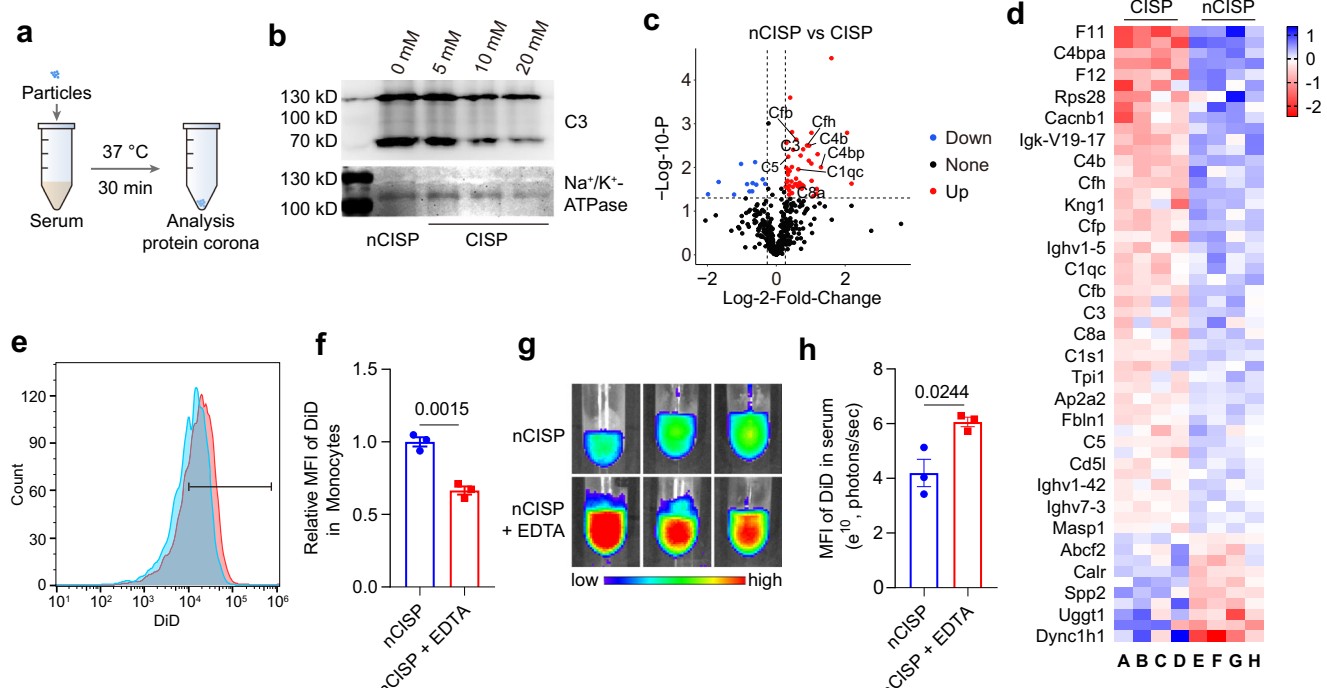

**Fig. 5 | CISP absorbed less complement C3 in serum. a** Schematic illustration showing the process to detect protein corona absorbed on CISP. **b** Western blot analysis of C3 absorbed on nCISP and CISP. **c** Volcano plot showing the serum proteins absorbed on CISP and nCISP. Fold_change means nCISP/CISP (*n* = 4 independent samples). **d** Heat map showing the serum proteins absorbed on CISP and nCISP with significant difference (fold change >1.2, *p* < 0.05) (*n* = 4 independent samples). **e**, **f** Uptake of nCISP by monocytes detect by flow cytometry (**e**) and the relative mean fluorescence intensity (MFI) of DiD in monocytes (**f**) (*n* = 3 independent samples). Blood of mice was collected and incubated with EDTA (2 mM) for

20 min, and nCISP was added into blood and incubated for 30 min. **g**, **h** Fluorescence images of plasma (**g**) and the mean fluorescence intensity (MFI) of DiD in plasma (**h**) (*n* = 3 independent samples). Data represent mean ± SEM (**f**, **h**). Student's two-sided *t* test was used for the statistical analysis in **f**, **h**. The experiments for **b**, **e**–**h** were repeated three times independently with similar results. Source data are provided as a Source Data file. C3 complement 3, QFN Quercetin-ferrum nanoparticles, QFN717 QFN loaded with SR-717, nCISP normal-Cholesterol cell membrane-coated ICB agent and QFN717, CISP low-cholesterol membrane-coated ICB agent and QFN717.

blood retention in the blood at 4 h post injection (Fig. 1d and Fig. 4c, d). These results indicate that the lowered cholesterol may delay the clearance of membrane-coated nanoparticle in the blood stream in a context-dependent manner.

Phagocytes in the blood (mainly including monocytes, lymphocytes, and neutrophils) are important players in the direct clearance of nanoparticles in the body[9,12]. Therefore, we analyzed the uptake of CISP and nCISP by phagocytes in the blood (Fig. 4e). Results showed that the amount of CISP (marked with DiD) in plasma or serum was higher than that of nCISP (Fig. 4f–h and Supplementary Fig. 22a, b), with the Area Under Curve (AUC) of CISP being ~55% higher and the half-life time (*t*1/2) prolonged (Fig. 4i and Supplementary Fig. 23). The AUC of SR-717 in CISP group was also ~65% higher than that in nCISP group (Supplementary Fig. 24). While the amount of CISP in monocytes was only half of nCISP, the phagocytosis rate of CISP and nCISP by neutrophils and lymphocytes were similar but lower than monocytes (Fig. 4j–o). Similar results were also observed in cell membrane vesicle and macrophage membrane-coated nanoparticles (Supplementary Figs. 16d, e, 20e–j). Therefore, cholesterol level seemed mostly affected monocyte clearance, and the clearance of nCISP by monocytes might be responsible for its poor performance in vivo.

To test the phenomenon in a more relevant clinical setting, human blood from healthy male donors was used to assess the uptake of nCISP and CISP by monocytes (Supplementary Fig. 25a). After incubating the particles with human blood at 37 °C for 0.5 h, the uptake of CISP by monocytes was also significantly lower than nCISP, and more CISP were left in the plasma (Supplementary Fig. 25b–e). Cell membrane sourced from human melanoma cells A375 was also used here[2],

results showed that low-cholesterol membrane-coated PLGA nanoparticles (e.g., CAM) and their normal counterpart (e.g., nCAM) had similar targeting ability to A375 cells in vitro, and more CAM remained in the plasma after incubation with human blood (Supplementary Fig. 25f–i). Hence, the cholesterol removal strategy displayed good potential for clinical translation.

It was reported that complement proteins could deposit on the surface of nanoparticles and prime them for removal by immune cells like monocytes[8,11,12,40]. Indeed, we found that the adsorption of complement C3 on CISP in mice serum is positively correlated with cholesterol level in the membrane, and the content of activated complement C5a in blood was also lower (Fig. 1d, Fig. 5a, b and Supplementary Fig. 26). The proteomics results showed that nCISP absorbed more complement proteins (C3, C5, C1s1, CD4b, C8a, etc.) in serum compared with CISP (Fig. 5c, d). When liposomes with different levels of cholesterol were incubated with a serum of mice, higher levels of cholesterol led to stronger C3 absorption in serum (Supplementary Fig. 27a, b), which was consistent with the previous report[16]. These results indicated the level of cholesterol in cell membrane-coated nanoparticles may affect complement proteins absorbed by them. Thus, we used ethylenediaminetetraacetic acid (EDTA), a known inhibitor of complement[41], to test the relationship between complement protein function and blood clearance. Results showed that EDTA indeed reduced the uptake of nCISP by monocytes in the blood, and more nCISP remained in the plasma (Fig. 5e–h). These results indicate that the complement proteins absorbed on biological membrane-wrapped nanoparticles may enhance their clearance by monocytes in the blood, and reduced adsorption and activation of the complement system by removing cholesterol may impede this clearance[16].

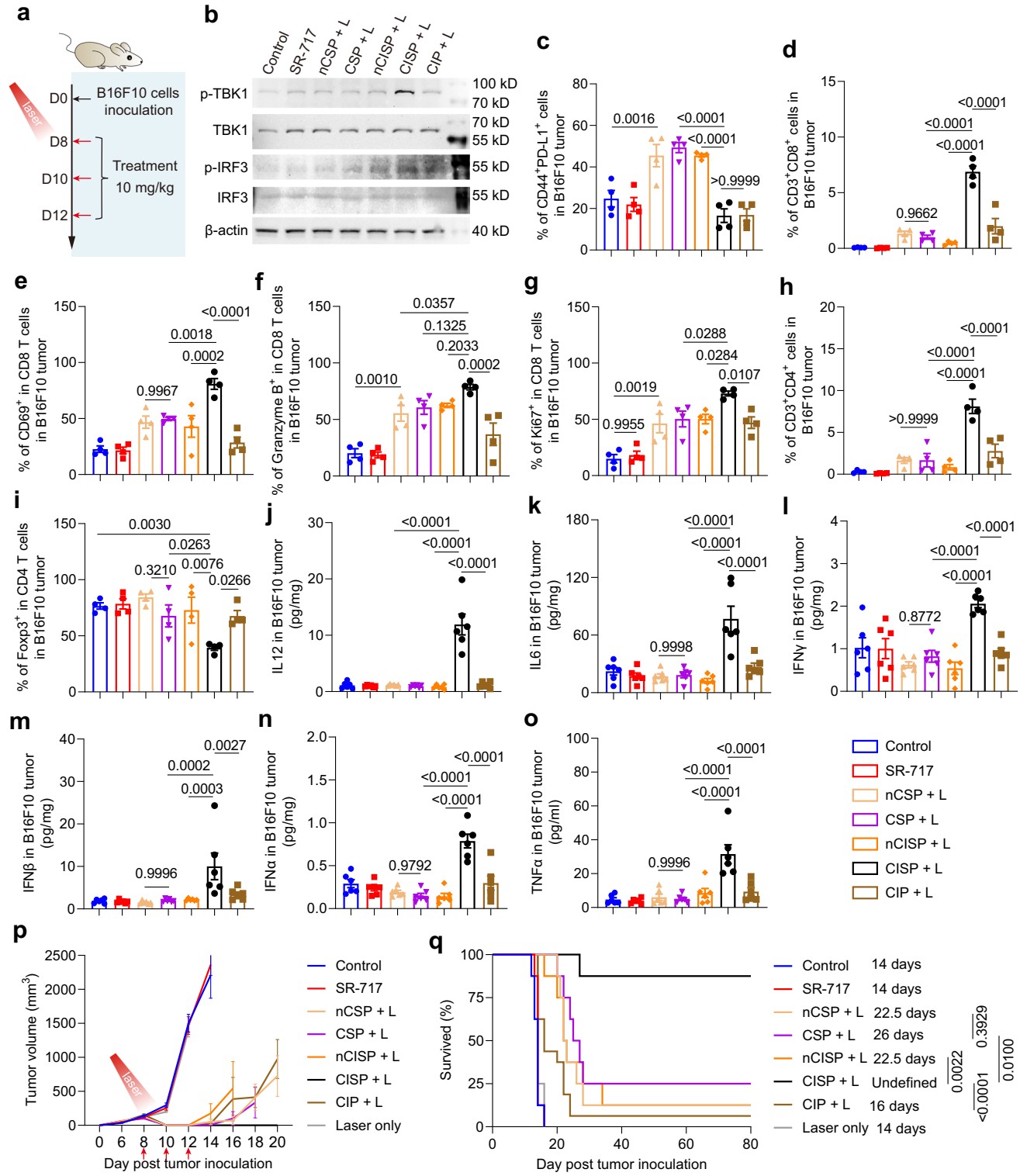

The hydrophobicity of the nanoparticle has a significant impact on the adsorption of proteins such as complements[42]. We found that CISP and liposomes with lower levels of cholesterol had better hydrophilicity (Supplementary Figs. 27c, d, 28a, b). However, the detailed mechanism of CISP in reducing complement adsorption is yet to be investigated in the future.

### The anti-tumor effect of CISP in mice

Encouraged by these results, we first tested the anti-tumor effect of CISP against melanoma in mice. B16F10 cells ($7 \times 10^5$ cells/mouse) were

inoculated subcutaneously in C57BL/6 mice. Drugs were intravenously injected on day 8, 10, and 12, and one laser irradiation was applied to activate photothermal therapy at 12 h post the first injection (Fig. 6a)[19]. Western blot results demonstrated that CISP significantly increased the expression of p-TBK1 and p-IRF3 in tumors, indicating that STING signal pathway in tumor sites was activated (Fig. 6b). In comparison, free SR-717, QFN$_{717}$, nCSP, and nCISP all displayed limited effect (Fig. 6b). Similarly, CISP also significantly reduced the expression of PD-L1 in tumor cells while nCSP, CSP, and nCISP showed weaker effects (Fig. 6c and Supplementary Fig. 29a). With the combined therapeutic

**Fig. 6 | CISP enhanced anti-tumor immunity in mice bearing melanoma.**
**a** Schematic illustration showing the experiment process. Red arrows indicate the administration of nanomedicines. The dosage of SR-717 is 10 mg/kg, and tumors were irradiated with 808 nm laser for 5 min (1.0 W/cm²) at 12 h post the first injection. **b** Western blot analysis of p-TBK1, TBK1, p-IRF3 and IRF3 in tumors at 6 days post laser irradiation. **c** Flow-cytometric analysis of CD44⁺PD-L1⁺ cells in B16F10 melanoma at 6 days post laser irradiation ($n = 4$ mice). **d** Flow-cytometric analysis of CD3⁺CD8⁺ cells in B16F10 melanoma ($n = 4$ mice). **e** Flow-cytometric analysis of CD69⁺ cells in CD8 T cells in B16F10 melanoma ($n = 4$ mice). **f** Flow-cytometric analysis of Granzyme B⁺ cells in CD8 T cells in B16F10 melanoma ($n = 4$ mice). **g** Flow-cytometric analysis of Ki67⁺ cells in CD8 T cells in B16F10 melanoma ($n = 4$ mice). **h** Flow-cytometric analysis of CD3⁺CD4⁺ cells in B16F10 melanoma ($n = 4$ mice). **i** Flow-cytometric analysis of Foxp3⁺ cells in CD4 T cells in B16F10 melanoma ($n = 4$ mice). **j–o** IL-12 (**j**), L-6 (**k**), IFNγ (**l**), IFNβ (**m**), IFNα (**n**), and TNFα (**o**) in tumors at 6 days post laser irradiation ($n = 6$ mice). **p** Tumor growth curve of mice bearing melanoma. Red arrows indicate the administration of nanomedicines ($n = 8$ mice), and tumors were irradiated with 808 nm laser at 12 h post the first injection. **q** Survival curve of mice bearing melanoma, $n = 8$ mice. Data represent mean ± SEM (**c–p**). Statistical significance was determined by one-way ANOVA test in **c–o**, and it was two-sided and adjustments were made for multiple comparisons. Survival was measured using the Kaplan–Meier method and statistical significance was calculated by log-rank test in **q**. The experiments for **b** were repeated three times independently with similar results. Source data are provided as a Source Data file. QFN Quercetin-ferrum nanoparticles, QFN₇₁₇ QFN loaded with SR-717, nCSP normal-Cholesterol cell membrane-coated QFN₇₁₇, CSP low-cholesterol membrane-coated QFN₇₁₇, nCISP normal-Cholesterol cell membrane-coated ICB agent and QFN₇₁₇, CISP low-cholesterol membrane-coated ICB agent and QFN₇₁₇, CIP low-cholesterol membrane-coated ICB agent and QFN.

effects, CISP significantly increased the number of CD8 T cells, CD69⁺ CD8 T cells, Granzyme B⁺ CD8 T cells, Ki67⁺ CD8 T cells, and CD4 T cells in tumors (Fig. 6d–h and Supplementary Fig. 29b, c). The maturation of DCs, M1 macrophages, memory T cells in tumors, and activated T cells in the tumor-draining lymph nodes (TdLNs) were also significantly increased (Supplementary Figs. 30a–c, 31a, 32a–d, 33a, b). Besides, Treg cells and M2 macrophages in tumors were decreased (Fig. 6i, Supplementary Fig. 31b), and the level of IL-12, IL-6, IFNγ, IFNβ, IFNα, and TNFα in tumors was elevated (Fig. 6j–o). In contrast, other groups only showed a limited effect on activating anti-tumor immune response (Fig. 6d–o). It is worth noting that although CIP (denotes low-cholesterol membrane-coated ICB agent and Photothermal agent) could also reduce PD-L1 in tumors, the final outcome is unsatisfactory, indicating that SR-717 is necessary to boost the immune responses (Figs. 1h, 2f, 6c–o). Ultimately, ~85% of the mice in CISP group were cured with no tumor recurrence on day 80, which is the best outcome (Fig. 6p, q, Supplementary Figs. 34, 35). In contrast, nCSP, CSP, and nCISP displayed weaker therapeutic efficiency, with a survival rate of ~20%; and no mice survived with free SR-717 treatment, reflecting the importance of the delivery system (Fig. 6q). Noticeably, the tumor recurrence rate in CISP group was lower than CIP, again illustrated the important role of SR-717 in this system (Fig. 6q). Taken together, the CISP which utilized low-cholesterol membrane showed higher delivery performance and better therapeutic outcomes combined with PTT in mice bearing melanoma.

Considering that PTT is not suitable for some cancers in clinical practice, we then examined the effect of CISP in some other tumor models. Here, colon cancer cells (MC38) ($5 \times 10^5$ cells/mouse) were inoculated subcutaneously in C57BL/6 mice (Fig. 7a). Fluorescence images showed that tumor accumulation of CISP was higher than that of nCISP, CSP, and nCSP (Supplementary Fig. 36a, b). Similarly, CISP and CIP also significantly reduced the expression of PD-L1 in tumor cells while nCSP, CSP, and nCISP showed weaker effects (Fig. 7b and Supplementary Fig. 37a). CISP increased CD8 T cells in MC38 tumors, and CSP also improved it to some extent, but is weaker than CISP (Fig. 7c and Supplementary Fig. 37b). More specifically, CISP treatment resulted in the strongest increase of CD8 T cells, activated CD8 T cells, CD4 T cells, M1 Macrophages, NK cells, and matured DCs in MC38 tumors (Fig. 7c–g, Supplementary Figs. 37c–f, 38a–f). Maturation of DCs, activated CD4 T cells, and CD8 T cells in the TdLNs (Supplementary Fig. 39a–h); as well as activated CD4 T cells and CD8 T cells in splenocytes all displayed the highest increase in CISP group (Supplementary Fig. 40a–c). CISP caused significant up-regulation of the CD8⁺/CD4⁺ ratio in the TdLNs, which was reported to associated with the prognosis of a variety of cancers and the response to immunotherapy[43] (Supplementary Fig. 41a, b). The changes in IL-12, IL-6, IFNγ, IFNβ, TNFα, and Treg cells in tumors were also the greatest in CISP group (Fig. 7h, Supplementary Figs. 37g, 42a–e). In contrast, other groups including CIP and CSP showed limited activation of anti-tumor immune response (Fig. 7c–h and Supplementary Fig. 42a–e).

Ultimately, CISP caused the strongest MC38 tumor suppression, and ~40% of treated mice survived on day 80, while all mice in other groups died (Fig. 7i–j, Supplementary Figs. 43, 44). Thus, CISP is also significantly more effective than other systems in treating MC38 colon cancer, even without the introduction of PTT.

Additionally we also detected the effect of CISP in mice bearing Lewis lung cancer. Results showed that CISP could also target LLC tumors (Supplementary Fig. 45a, b). To examine whether loading SR-717 into CISP leads to better outcome, CIP administrated with free SR-717 (CIP + SR-717) was used as a comparison in this tumor model (Supplementary Fig. 45c). Measurements showed that CIP+ SR–717 along with CISP and CIP significantly reduced the expression of PD-L1 in LLC tumor cells while nCSP, CSP, and nCISP showed weaker effects (Supplementary Fig. 45d). However, only CISP resulted in the strongest increase of CD8 T cells, NK cells, M1 macrophage and anti-tumor cytokines including IL-6, IFNγ, IFNβ, TNFα and IFNα in LLC tumors (Supplementary Fig. 45e–l). Ultimately, only CISP inhibited the growth of LLC tumors, while other groups basically failed to show therapeutic effects in this tumor model (Supplementary Figs. 45m, n, 46, 47). These results indicated that CISP exhibited good therapeutic outcomes in LLC tumors, and support the necessity of loading SR-717 into CISP to activate anti-tumor immune responses.

Metastasis is a major cause of death for cancer patients and lung metastasis is common for melanoma[44]. Hence, we also examined the effect of CISP in a lung metastatic melanoma mice model. Fluorescence images showed that CISP could target lungs with metastasis sites (Supplementary Fig. 48a, b). Further examinations showed that CISP performed better than CIP, nCISP, CSP, and nCSP in reducing metastatic foci in lung (Supplementary Fig. 49a–c). Significantly higher CD8 T cells, matured DCs, M1 macrophages, and IFNα could also be detected in CISP-treated mice (Supplementary Fig. 49d–h). These results indicated that CISP could induce anti-tumor immune responses to inhibit metastatic melanoma.

The biocompatibility of CISP was also briefly evaluated[19,45]. Body weight of mice was unaffected by CISP (Supplementary Fig. 50a–c). Blood routine examination and blood biochemistry (including ALT, AST, LDH, CREA, UREA, and CHE) of mice bearing MC38 tumor were all in the normal range treated with CISP (Supplementary Figs. 51a–f, 52a–f). Blood biochemistry of mice bearing LLC tumor was also in the normal range treated with CISP (Supplementary Fig. 53a–f). Hematoxylin-eosin staining of organs showed that CISP did not induce clear kidney injury, pulmonary toxicity, cardiac injury, or inflammatory infiltrates in the spleen (Supplementary Figs. 54, 55). Hence, CISP appears to be biocompatible and safe in mice.

## Discussion

As mentioned, "additive strategies" are common in nano-delivery system design. Indisputably, this type of strategies has been successful in various applications, but it is still worth exploring alternative strategies which may be advantageous in some aspects. Here, we tested a

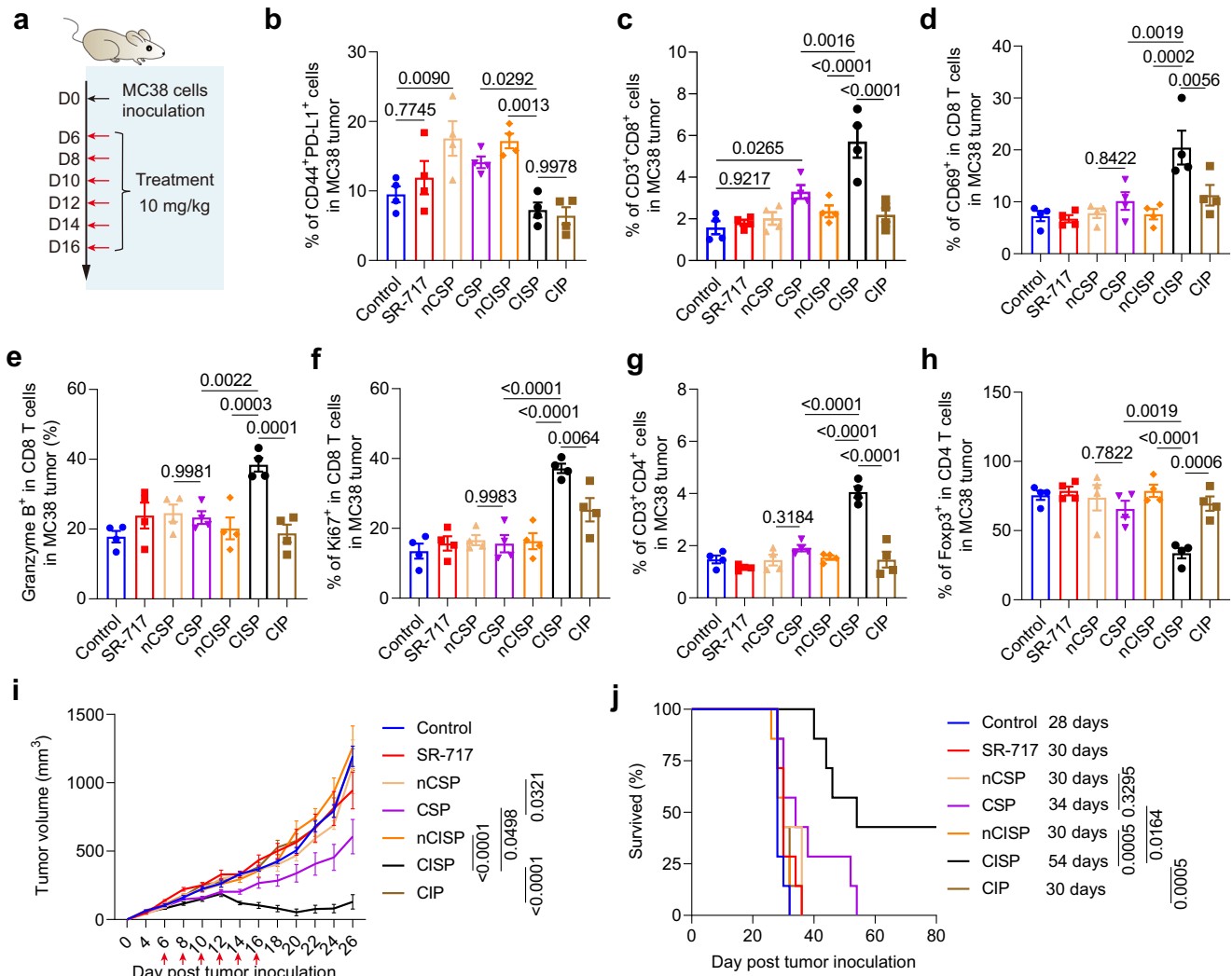

**Fig. 7 | CISP enhanced anti-tumor immunity in mice bearing colon cancer.**
**a** Schematic illustration showing the experiment process. Red arrows indicate the administration of nanomedicines. The dosage of SR-717 is 10 mg/kg. **b** Flow-cytometric analysis of CD44+PD-L1+ cells in MC38 tumors on day 18 ($n = 4$ mice).
**c** Flow-cytometric analysis of CD3+CD8+ cells in MC38 tumors ($n = 4$ mice). **d** Flow-cytometric analysis of CD69+ cells in CD8 T cells in MC38 tumors ($n = 4$ mice).
**e** Flow-cytometric analysis of Granzyme B+ cells in CD8 T cells in MC38 tumors ($n = 4$ mice). **f** Flow-cytometric analysis of Ki67+ cells in CD8 T cells in MC38 tumors ($n = 4$ mice). **g** Flow-cytometric analysis of CD3+CD4+ cells in MC38 tumors ($n = 4$ mice). **h** Flow-cytometric analysis of Foxp3+ cells in CD4 T cells in MC38 tumors ($n = 4$ mice). **i** Tumor growth curve of mice bearing MC38 tumors. Red arrows indicate the administration of nanomedicines ($n = 7$ mice). **j** Survival curve of mice

bearing MC38 tumors ($n = 7$ mice). Data represent mean ± SEM (**b**–**i**). Statistical significance was determined by one-way ANOVA test in **b**–**h** and it was two-sided and adjustments were made for multiple comparisons. Statistical significance was determined by two-way ANOVA test in **i**, and it was two-sided and adjustments were made for multiple comparisons. Survival was measured using the Kaplan-Meier method and statistical significance was calculated by log-rank test in **j**. Source data are provided as a Source Data file. QFN Quercetin-ferrum nanoparticles, QFN717 QFN loaded with SR-717, nCSP normal-Cholesterol cell membrane-coated QFN717, CSP low-cholesterol membrane-coated QFN717, nCISP normal-Cholesterol cell membrane-coated ICB agent and QFN717, CISP low-cholesterol membrane-coated ICB agent and QFN717, CIP low-cholesterol membrane-coated ICB agent and QFN.

"subtractive strategy" that focuses on cholesterol in biological membranes. In this study, we discovered the reversed relationship between clearance time and cholesterol level in biological membranes, and this cholesterol depletion did not affect the short-term targeting efficiency nor surface PD-1 function. This may help avoid the dilemma that adding more and longer surface disguise molecules could improve circulation time but affect surface functional molecules at the same time (Supplementary Fig. 18a–f)[15]. Interestingly, a recent study also reported that lowered cholesterol levels could indeed enhance membrane permeability and improve molecule transfer across biological membranes, showing another potential of subtractive strategy[46].

It is known that the deposition of complements, as part of immune surveillance mechanism, initializes foreign body removal process and the nanomedicines are rapidly cleared from blood flow or

body fluids. So far, plenty of clinical failures of nanomedicine could be attributed to the protein deposition on the surface, and the clearance of nanoparticles by phagocytes in blood may be an important reason[8,10,11,15,37,47–49]. Thus, it is worth noting that our further mechanistic investigations indicate that the delayed clearance of low-cholesterol membrane-coated nanoparticles is likely related to the reduced deposition of complements on these particles. The buildup of complements on foreign bodies could be triggered by many events, and is a complicated matter to articulate. For example, recent studies have indicated that binding of IgM on nanoparticle surface is positively related to deposition of complements and following clearance[50]. The hydrophobicity of nanoparticle was reported to impact formation of protein corona on nanoparticles[42], and the enhanced hydrophilicity may partially explain the decreased absorption of complement on

CISP (Supplementary Figs. 27c, 28a, b). However, there is still plenty more to explore in this field in order to clarify the relationships between cholesterol level and the interactions of biological membranes in vivo environments[11,16]. Currently, we are testing more delivery systems and exploring the mechanism in more detail.

Nevertheless, in animal models, our proof-of-concept system using the substrative strategy did reduce complements absorption on particles, inhibit the uptake of nanoparticles by monocytes in the blood, and retain the PD-1 activity, hence in many ways verified the value of the strategy. Clearly, this strategy may also be used for other medical conditions other than cancer, such as improving the targeting of neutrophil membrane-coated nanoparticles to inflamed tissues[51] (Supplementary Fig. 20), or achieving long-term controlled release in blood[52]. Different delivery vehicles, such as bacteria, could be coated to suppress immune surveillance and enhance the enrichment of bacteria in target sites[53,54].

CTLL2 cell membrane-coated nanoparticles showed different tumor-targeting capability in the BALB/c and C57 mice bearing 4T1 tumors, which suggest that the blood clearance mechanism is somewhat different in different type of mice[33]. This is an important reminder of the heterogeneity between different animals, and especially, between animal models and human[11]. The fast clearance of nanoparticles by phagocytes in body is one of the major reasons that causes unsatisfactory clinical outcome of nanomedicines even when they display good targeted efficiency in preclinical experiments[10,11]. On one hand, this again calls for caution on clinical translation of new therapies. On the other hand, it prompts that the nanomaterial dynamics and interactions in different animals worth more attention.

Additionally, based on the key functions of cholesterol in lipid membranes, how much cholesterol could be removed is an essential issue. Too low level of cholesterol may cause problem on membrane morphology maintenance. After systematic trials, the highest concentration of β-CD used in this study was 20 mM, as higher concentration of β-CD could affect both the cell viability of CTLL2-PD1, which may affect the integrity of T cell membrane and the targeting ability of CISP (Supplementary Figs. 17, 2, and Fig. 3c, d). These results were consistent with previous report show that the cell membrane coating integrity affects the targeting ability of membrane-coated nanoparticles[55], and the method established in this work may also be used to detect the membrane coating integrity on CISP in the future.

In summary, we found that membranes with lower cholesterol content reduced the uptake rate of CISP by monocytes in the blood by ~50%, while maintaining its tumor-targeting capability. As a result, CISP successfully delivered SR-717 to tumor site to activate local STING pathway, while PD-1 on CISP surface blocked the up-regulated PD-L1 in tumor cells induced by STING agonist (Fig. 8). Due to the strong anti-tumor immune activity, >80% of the mice showed no sign of recurrence with the combination of PTT, and CISP could also boost anti-tumor immunity to inhibit colon cancer, Lewis lung cancer and metastatic melanoma in mice without laser irradiation. The whole system also displayed good biosafety, and this camouflage strategy has good potential for further development. A new system could also be designed to further exploit this strategy or to explore other possibly subtractable elements.

## Methods

This research complies with all relevant ethical regulations approved by the Institutional Animal Care and Ethics Committee of Sichuan University and the Ethics Committee of West China Fourth Hospital and West China School of Public Health, Sichuan University.

### Materials

Quercetin, FeCl₂.4H₂O, (2-hydroxypropyl)-β-cyclodextrin, Coumarin, 1,1'-Dioctadecyl-3,3,3',3'-Tetramethylindodicarbocyanine, 4-Chlorobenzenesulfonate Salt (DiD), Ethylene Diamine Tetraacetic Acid (EDTA) and Nafamostat mesylate were brought from MACKLIN in China. Protease Inhibitor Cocktail, SR-717, and DAPI were purchased from MedChemExpress. 0.67 dL/g carboxy-terminated 50:50 poly(-lactic-co-glycolic acid) (PLGA) was obtained from Chongqing Yusi Pharmaceutical Technology in China. FITC anti-mouse CD3 (catalog number 100204, clone: 17A2, dilution: 1:100), APC anti-mouse CD4 (catalog number 100412, clone: GK1.5, dilution: 1:100), Pacific Blue anti-mouse CD69 (catalog number 104524, clone: H1.2F3, dilution: 1:100), PE anti-mouse CD8a (catalog number 100708, clone: 53-6.7, dilution: 1:200), Brilliant Violet 421 anti-mouse FOXP3 (catalog number 126419, clone: MF-14, dilution: 1:50), PE anti-mouse CD274 (B7-H1, PD-L1) (catalog number 124308, clone: 10 F.9G2, dilution: 1:100), FITC anti-mouse/human CD44 (catalog number 103006, clone: IM7, dilution: 1:200), Pacific Blue anti-human/mouse Granzyme B (catalog number 515408, clone: GB11, dilution: 1:20), Pacific Blue anti-mouse Ki67 (catalog number 652422, clone: 16A8, dilution: 1:100), APC anti-mouse/human CD11b (catalog number 101212, clone: M1/70, dilution: 1:100), PE anti-mouse F4/80 (catalog number 111704, clone: W20065D, dilution: 1:100), Pacific Blue anti-mouse CD80 (catalog number 104724, clone:16-10A1, dilution: 1:200), FITC anti-mouse CD86 (catalog number 105110, clone: PO3, dilution: 1:50), APC anti-mouse NK-1.1 (catalog number 156506, clone: S17016D, dilution: 1:100), APC anti-mouse CD11c (catalog number 117310, clone: N418, dilution: 1:100), FITC anti-mouse CD11c (catalog number 117306, clone: N418, dilution: 1:200), PE anti-mouse CD40 (catalog number 157506, clone: FGK45, dilution: 1:50), FITC anti-mouse CD206 (catalog number 141704, clone: C068C2, dilution: 1:400), Pacific Blue anti-mouse CD62L (catalog number 161208, clone: W18021D, dilution: 1:200), and PE anti-mouse CD279 (PD-1) (catalog number 135206, clone: 29 F.1A12, dilution: 1:50) were purchased from Biolegend. Alexa fluor 488 anti-mouse cd274 (PD-L1) was obtained from BD Biosciences (catalog number 568304, clone: 10 F.9G2(RUO), dilution: 1:100). Anti-PD-1 (ab214421, clone: EPR20665, dilution: 1:1000), anti-PD-L1 (ab213480, clone: EPR20529, dilution: 1:1000), anti-C3 (ab200999, clone: EPR19394, dilution: 1:1000), anti-TBK1 (ab40676, clone: EP611Y, dilution: 1:5000) and anti-β actin (ab8226, clone: mAbcam 8226, dilution: 1:1000) for western blot were obtained from Abcam. Anti-p-TBK1 (catalog number 5483, dilution: 1:1000) and anti-p-IRF3 (catalog number 29047, dilution: 1:1000) were purchased from Cell Signaling Technology. Anti-IRF3 (catalog number A19717, clone: ARC0198, dilution: 1:1000) was purchased from Abclonal Technology in China. Anti-Na⁺/K⁺ ATPase (catalog number ET1609-76, dilution: 1:10000) was purchased from HuaBio in China. PE-conjugated anti-mouse IgG(H + L) secondary antibody (catalog number, SA00008-9, dilution: 1:50), HRP-conjugated Affinipure Goat anti-mouse IgG(H + L) secondary antibody (catalog number, SA00001-1, dilution: 1:8000), HRP-conjugated Affinipure Goat anti-rabbit IgG(H + L) secondary antibody (SA00001-2, dilution: 1:8000), and Anti-LAMP1 (catalog number 67300-1-Ig, dilution: 1:100) was purchased from Proteintech in China. RPMI-1640 medium and trypsin were purchased from KeyGEN BioTECH in China. Mouse Monocyte Extraction Kit (catalog number P5230), Human Monocyte Extraction Kit (catalog number P8680), Mouse Neutrophil extraction kit (catalog number P9201), Mouse Lymphocyte extraction kit (catalog number P8620), and Puromycin (catalog number P8230) were purchased from Solarbio in China. Cholesterol Assay Kit (catalog number E1015-50) was obtained from ApplyGEN in China. C5a ELISA Kit was obtained from Redotbiotech. DSPE-PEG2000 was purchased from Ponsure Biological. Ammonium-chloride-potassium (ACK) lysing buffer (catalog number R1010) and CCK-8 (CA1210) were purchased from Solarbio in China. Complete Freund's Adjuvant (catalog number 7027) was purchased from Chondrex, Washington DC, USA. MojoSort™ Isolation Kits were obtained from Biolegend (catalog number B301590). Mouse IL-6 ELISA Kit was obtained from DAKEWE (catalog number 1210602). Mouse IL-12p70 ELISA Kit was obtained from DAKEWE (catalog number 1211202). Mouse IFNγ ELISA Kit was obtained from DAKEWE (catalog number

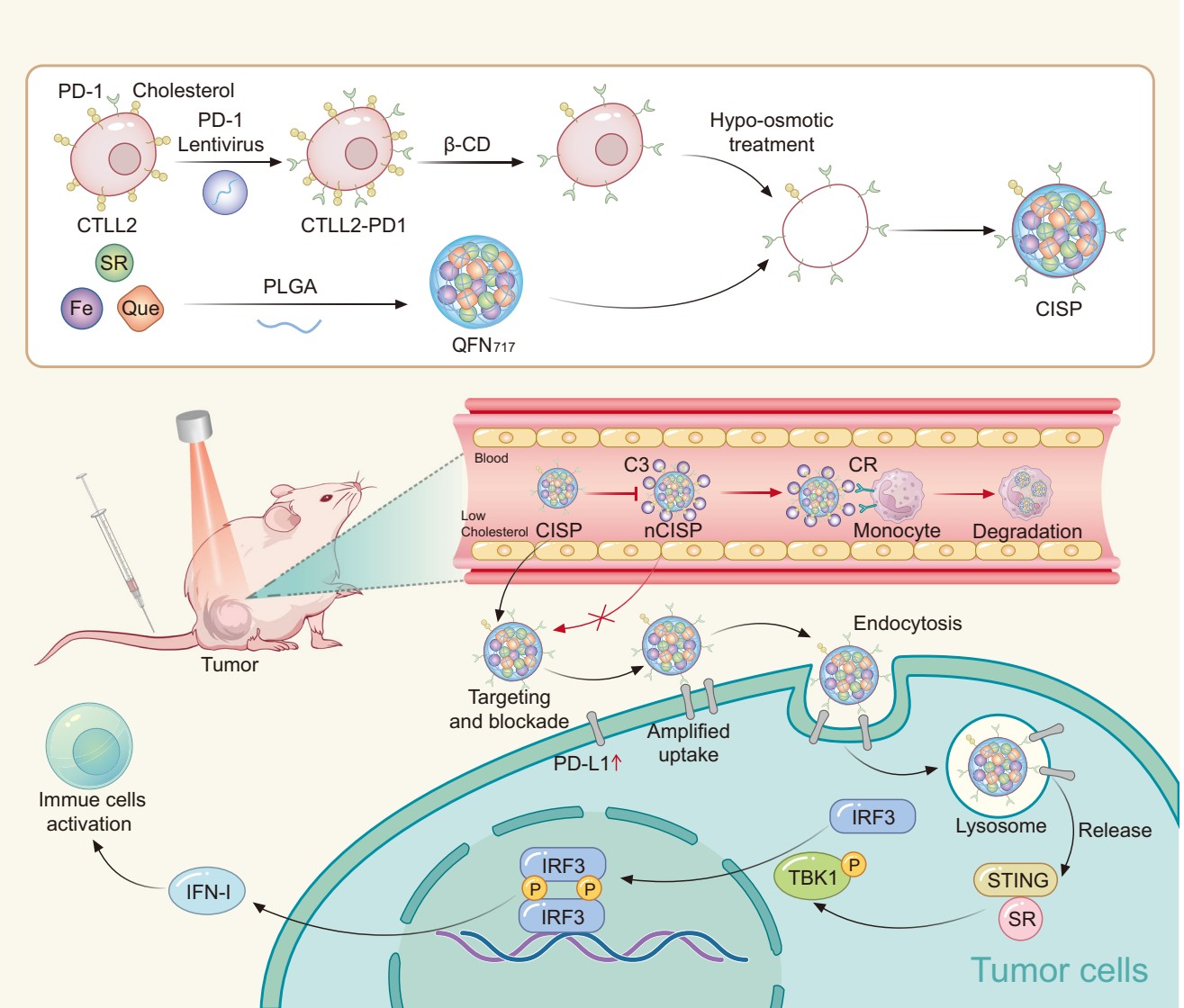

**Fig. 8 | Schematic illustration of the preparation of CISP and CISP enhanced the tumor-targeted delivery of SR-717.** PD-1 programmed death-1, PD-L1 programmed cell death-Ligand 1, SR SR-717, β-CD (2-hydroxypropyl)-β-cyclodextrin, Que Quercetin, PLGA poly(lactic-co-glycolic acid), QFN$_{717}$ quercetin-ferrum nanoparticles (QFN) encapsulated SR-717, CISP low-cholesterol membrane-coated ICB agent and QFN$_{717}$, nCISP normal-Cholesterol membrane-coated ICB agent and QFN$_{717}$, C3 complement 3, CR complement receptor, IFN-I type I interferon.

1210002). Mouse TNFα ELISA Kit was obtained from DAKEWE (catalog number 1217202). Mouse IFN-β ELISA Kit was obtained from Beyotime (catalog number PI568). Mouse IFN-α1 ELISA Kit was obtained from Biolegend (catalog number 447904).

## Animals and ethics statement

Male C57BL/6 mice (18 ± 2 g, 6–8 weeks) were obtained from Gem-Pharmatech in China. Mice were housed in constant environmental conditions (room temperature, 21 ± 1 °C; relative humidity, 40–70%, and a 12 h light–dark cycle). Mice were access to food and water free. All animal experiments were approved by the Institutional Animal Care and Ethics Committee of Sichuan University. The experiment was designed without considering the sex of mice, and male mice were selected to ensure gender uniformity. The maximum tumor burden in mice permitted by the Institutional Animal Care and Ethics Committee of Sichuan University was 2000 mm³. In some cases, this limit has been exceeded by the last day of measurement and the mice were immediately euthanized.

## Cells lines

MC38 and B16F10 were brought from American Type Culture Collection and were cultured in a complete RPMI-1640 medium. B16F10 cells transfected with Control Double Nickase Plasmid (Scramble B16F10), and B16F10 cells transfected with PD-L1 Double Nickase Plasmids (B16F10$^{PD-L1 KO}$ cells) were established in our previous work[19] (a kind gift from Prof. Zhirong Zhang, West China School of Pharmacy, Sichuan University, Chengdu, China), and they were cultured in complete RPMI-1640 medium. CTLL2 (catalog number CL-0331) was purchased from Procell Life Science & Tech-nology.Co.,Ltd, and it was cultured in a complete RPMI-1640 medium with IL-2 (100 U mL⁻¹). A375 (catalog number CL-0014) and LLC (catalog number CL-0140) were brought from Procell Life Science & Tech-nology.Co.,Ltd, and they were cultured in complete DMEM medium. RAW264.7 (catalog number CL-0190) was brought from Procell Life Science & Tech-nology.Co.,Ltd, and it was cultured in complete RPMI-1640 medium. CTLL2-PD1 cell was established by Shanghai Genechem in China, and it was cultured in a complete RPMI-1640 medium with IL-2 (100 U mL⁻¹). All cells were

maintained in a humidified atmosphere incubator containing 5% $CO_2$ at 37 °C.

## Preparation of PLGA nanoparticles

Oil phase: PLGA (5 mg) was dissolved $CH_2Cl_2$ (1 mL), and stirred for 5 min. Water phase: Polyvinyl alcohol (PVA, 20 mg) and Tween 80 (10 mg) were dissolved in water (2 mL). Mixed oil phase and water phase, and treated with a probe ultrasonic for 10 min (300 W) in ice bath, then removed dichloromethane with a rotary evaporator. To prepare DiD-labeled PLGA nanoparticles, DiD (60 μg) was dissolved in the oil phase.

## Preparation of QFN$_{717}$ and QFN

Oil phase: Quercetin (1 mg), $FeCl_2.4H_2O$ (1 mg), SR-717 (1 mg), and PLGA (5 mg) were dissolved in ethanol (50 μL) and $CH_2Cl_2$ (1 mL), stirred for 5 min. Water phase: Polyvinyl alcohol (PVA, 20 mg) and Tween 80 (10 mg) were dissolved in water (2 mL). Mixed oil phase and water phase, and treated with a probe ultrasonic for 10 min (300 W) in an ice bath, then removed dichloromethane with a rotary evaporator. The preparation method of QFN is similar, except that SR-717 was not added. To prepare DiD-labeled nanoparticles, DiD (60 μg) was dissolved in the oil phase without adding $FeCl_2$.

## Remove cholesterol in cells

CTLL2-PD1 cells were washed with PBS, and then cells were incubated with different concentrations of (2-hydroxypropyl)-β-cyclodextrin (β-CD) in serum-free RPMI-1640 medium (without serum) for 0.5 h at 37 °C (20 mM of β-CD was used in most experiments otherwise stated). The cell membrane was then prepared to prepare CISP. A similar method was used to remove cholesterol in CTLL2, B16F10, A375, and RAW264.7 cells.

## Derivation of cell membrane

To prepare cell membranes, previously reported methods were used with some modifications[7,56]. Briefly, CTLL2, CTLL2 treated with β-CD, CTLL2-PD1, and CTLL2-PD1 treated with β-CD were collected and disrupted in hypotonic lysing buffer (pH 7.4, 20 mM Tris-HCl, 10 mM KCl, 2 mM $MgCl_2$, Protease Inhibitor Cocktail) at 4 °C for 30 min. Destroyed cells in an ice bath using a probe ultrasonic instrument (150 W, 3 min). Then centrifuged at $20,000 \times g$ for 20 min to remove organelles and large particles. The supernatant was centrifuged at $100,000 \times g$ for 40 min to harvest cell membrane. The pellets were resuspended with buffer (10 mM Tris buffer, 1 mM EDTA (pH = 7.5)). The content of cholesterol in cell membrane was assayed using Cholesterol Assay Kit (ApplyGEN).

## Preparation of cell membrane-coated nanoparticles and cell membrane vesicles

nCSP, CSP, nCISP, CISP: QFN$_{717}$ (1 mL) was mixed with each kind of cell membrane (derived from $4 \times 10^7$ cells), and sonicated using a probe ultrasonic instrument (100 W, 1 min) in ice bath, then extruded through 400 nm and 200 nm polycarbonate membranes with an extruder for 20 passes. Nanoparticles were stored at 4 °C before use.

CIP: QFN (1 mL) was mixed with the low-cholesterol cell membrane (derived from $4 \times 10^7$ CTLL2-PD1 cells), and sonicated using a probe ultrasonic instrument (100 W, 1 min) in ice bath, then extruded through 400 nm and 200 nm polycarbonate membranes with an extruder for 20 passes. CIP was stored at 4 °C before use.

PEG-nCISP: To prepare PEG-nCISP, the previously reported method was used with some modifications[2]. Briefly, cell membrane (derived from $4 \times 10^7$ CTLL2-PD1 cells) and DSPE-PEG2000 (180 μg) were mixed and were physically extruded through a 200 nm polycarbonate membrane for 10 passes. The membrane vesicles were mixed with QFN$_{717}$ and sonicated using a probe ultrasonic instrument

(100 W, 1 min) in ice bath, then extruded through 400 nm and 200 nm polycarbonate membranes with an extruder for 20 passes.

nCBM, CBM, nCRM, CRM, nCAM, and CAM: PLGA nanoparticles (1 mL) was mixed with each kind of cell membrane (derived from $4 \times 10^7$ cells), and sonicated using a probe ultrasonic instrument (100 W, 1 min) in an ice bath, then extruded through 400 nm and 200 nm polycarbonate membranes with an extruder for 20 passes. Nanoparticles were stored at 4 °C before use.

nC-vesicle and C-vesicle: Cell membrane sourced from CTLL2-PD1 cells and β-CD-treated cell membrane (derived from $4 \times 10^7$ cells) were dispersed in pure water (1 mL), and sonicated using a probe ultrasonic instrument (100 W, 1 min) in ice bath, then extruded through 400 nm and 200 nm polycarbonate membranes with an extruder for 20 passes. To prepare DiD-labeled vesicles, CTLL2-PD1 cells were incubated with DiD (60 μg/mL) for 0.5 h, and the cell membrane was isolated to prepare vesicles. DiD in vesicles was quantified by a fluorescent photometer.

## Preparation of liposomes

Phospholipids (10 mg) and cholesterol (2 mg, 4 mg, or 6 mg) were added to a round bottom flask (50 mL), and dissolved them with dichloromethane (5 mL). Removed dichloromethane with a rotary evaporator, added pure water (5 mL), and shaken for 4 h to disperse the film in water, and sonicated it using a probe ultrasonic instrument (200 W, 8 min) in ice bath.

## Characterization of nanoparticles

After negatively staining with phosphotungstic acid solution (2%, w/v), TEM was used to observe the morphology of nanoparticles (QFN717, nCSP, CSP, nCISP, CISP, nCRM, CRM, liposomes, nC-vesicles, C-vesicles).

The proportion of nanoparticles (nCISP, CISP (20 mM), CISP (30 mM)) coated with cell membrane were detected based on TEM images. Prepared triplicate sample for each nanoparticle, and observe the morphology by TEM. Randomly selected a field of vision to observe nanoparticle, and recorded 40 particles for each sample, and finally calculated the proportion of particles coated with cell membrane.

Sizes and zeta potential of nanoparticles (QFN717, nCSP, CSP, nCISP, CISP, CIP) were detected by Malvern Zetasizer ZS90.

The encapsulation efficiency (EE%) of SR-717 in PLGA$_{717}$, QFN$_{717}$, nCSP, CSP, nCISP, and CISP was measured by high-performance liquid chromatography (HPLC) (Agilent 1260 Infinity, USA). Measurement was performed at 37 °C on a C18 column (5 μm, 150 × 4.6 mm, Kromasil) with a mobile phase of acetonitrile and water (50:50, v/v) and a detection wavelength of 290 nm. Nanoparticles were separated via ultracentrifuge ($20,000 \times g$, 30 min), and SR-717 in the supernatant and centrifuged nanoparticle was determined by HPLC. EE% was calculated according to Eq. (1).

$$EE\% = SR\text{-}717_{NP}/(SR\text{-}717_{supernatant} + SR\text{-}717_{NP}) \times 100\% \qquad (1)$$

## Water contact angle of nCISP and CISP

nCISP and CISP (20 mL) were made to powders through freeze-drying and then were tableting on glass slices to examine the water contact angle. The contact angle of liposomes was examined in the same way.

## Immunofluorescence staining

To stain PD-1 on the cell membrane, cells were washed with cold PBS, and then incubated with anti-PD-1 at 4 °C for 45 min. Cells were then observed using confocal laser scanning microscope.

To stain PD-L1, LAMP1 in melanoma, tumors were fixed with 4% paraformaldehyde for 72 h. Tumors were gradually dehydrated in 15% and 30% sucrose solution, and then frozen sections (10 μm) were prepared. Then the sections were permeabilized with 0.1 % Triton

X-100 and incubated with anti-PD-L1 (Alexa fluor 488) at 4 °C for 45 min. To stain LAMP1 in sections, sections were incubated with anti-LAMP1 for 1.5 h at room temperature, then washed with PBS three times. PE-conjugated secondary antibody was then added and incubated for 1.5 h at room temperature, then washed with PBS three times.

## Western blot

Cells, cell membranes, or tumor pieces were harvested and lysed with lysis buffer (with 1 mM, PMSF) in an ice bath for 0.5 h. Then sample buffer was added, and boiled for 8 min. Samples (containing 20–30 µg protein) were separated by SDS-PAGE gel (8%) and transferred to polyvinylidene difluoride (PVDF) membranes. PVDF membranes were blocked for 1 h at room temperature, and incubated with primary antibody at 4 °C overnight with shaking. PVDF membranes were incubated with HRP-conjugated secondary antibody for 1.5 h at room temperature. The uncropped and unprocessed scans of blots were shown in the Source Data file.

## Uptake of nCISP and CISP by B16F10 cells in vitro

B16F10 cells ($2 \times 10^5$) were inoculated on 12 well plates. 24 h later, cells were treated with nCISP and CISP (marked with DiD) for 2 h. Cells were collected and the uptake of nanoparticles by cells was detected by flow cytometry and confocal laser scanning microscope.

## Tumor cell killing induced by activated T cells in vitro

MojoSort™ Isolation Kits were used to isolate CD8+ T cells in the spleen of mouse[36]. T cells were cultured in Super CultureTM (DKW34-MCL1) with mouse CD3/CD28/CD2 T cell activator (BioLegend) and IL-2 (10 ng/mL) for one week to get activated T cells. B16F10 cells treated with nCSP, nCISP, and CISP (10 µg protein/mL) were cocultured with activated T cells for 48 h (B16F10 cells/activated T cells = 1/10). T cells were removed and the viability of B16F10 cell was detected by CCK-8 kit.

## Biodistribution of nanoparticles in mice bearing tumors detected by IVIS imaging

B16F10 tumor model: Subcutaneously inoculated B16F10 cells ($7 \times 10^5$ cells/mouse) in male C57BL/6 mice ($18 \pm 2$ g, 6–8 weeks). Mice were intravenously injected with nanoparticles (marked with DiD) on day 10. Fluorescence images of tumors in mice were then taken at different time points.

MC38 tumor model: Subcutaneously inoculated MC38 cells ($5 \times 10^5$ cells/mouse) in male C57BL/6 mice ($18 \pm 2$ g, 6–8 weeks). Mice were intravenously injected with nanoparticles (marked with DiD) on the day 10. Fluorescence images of tumors in mice were taken at 6 h post injection.

LLC tumor model: Subcutaneously inoculated LLC cells ($5 \times 10^5$ cells/mouse) in male C57BL/6 mice ($18 \pm 2$ g, 6–8 weeks). Mice were intravenously injected with nanoparticles (marked with DiD) on day 12. Fluorescence images of tumors in mice were taken at 6 h post-injection.

Metastatic melanoma model: B16F10 cells ($7 \times 10^5$ cells/mouse) were intravenously infused into male C57BL/6 mice ($18 \pm 2$ g, 6–8 weeks). Mice were intravenously injected with nanoparticles (marked with DiD) on day 10. Fluorescence images of the lung in mice were taken at 6 h post injection.

## Quantification of SR-717 in vivo

To quantify the biodistribution of SR-717 in mice, free SR-717 (10 mg/kg) and nanoparticles were injected into mice bearing melanoma (iv.) on day 10. Mice were sacrificed at different time after administration. Heart, liver, spleen, lung, kidney, tumors, and serum of mice were collected. Tissues were weighed (200 mg) and homogenized (added with 0.3 mL saline) on a lysis instrument (4 °C). Then, 1 mL of organic solvent (acetonitrile/methanol = 1/4) was added, and vortexed for 30 min.

Centrifuged at 12,000 rpm for 10 min and filtered (0.22 µm) for LC-MS/MS analysis. Serum (200 µL) was collected in EP tubes. Then, 1 mL of organic solvent (acetonitrile/methanol = 1/4) was added, and vortexed for 30 min. Centrifuged at 12,000 rpm for 10 min and then filtered (0.22 µm) for LC-MS/MS analysis. Pharmacokinetic parameters in terms of AUC were calculated by DAS2.0 software.

## Quantification of DiD in serum

To quantify fluorescence of nanoparticles in serum more accurate, nCISP and CISP (marked with DiD, 0.25 mg/kg) were injected into mice bearing melanoma (iv.) on day 10. Mice were sacrificed at different time points (0.25 h, 0.5 h, 1 h, 2 h, 4 h, 8 h, 12 h, and 24 h) to isolate serum. Serum (0.1 mL) was mixed with pure water (3 mL), and then was detected by a fluorescence spectrophotometer. Calculated DiD content in serum based on the calibration curve. Pharmacokinetic parameters in terms of AUC and $t_{1/2}$ were calculated by DAS2.0 software.

## Arthritis model

Male C57BL/6 mice ($18 \pm 2$ g, 6–8 weeks) were injected with a single 20 µL of Complete Freund's Adjuvant (10 mg/mL) on the hindfoot sole to induce adjuvant-induced arthritis (AIA) model[57]. On day 14, mice were injected with nanoparticles (marked with DiD, iv.).

## Uptake of nCISP and CISP by phagocytes in the blood, and C5a in blood

Subcutaneously inoculated B16F10 cells ($7 \times 10^5$ cells/mouse) in male C57BL/6 mice. At 10 days post cell inoculation, mice were intravenously injected with nCISP and CISP (marked with DiD, 6 µg/mice). Mice were sacrificed (at 0.5 h post injection) and the blood were collected using anticoagulant tube containing EDTA and nafamostat mesylate[9]. Monocyte, neutrophil, and lymphocyte in the blood were separated using Mouse Monocyte Extraction Kit, Mouse Neutrophil extraction kit, and Mouse Lymphocyte extraction kit. Then, the uptake of nCISP and CISP by monocytes was measured with flow cytometry. The plasma was collected and the fluorescence images were taken, and the C5a in the plasma was examined using ELISA Kit.

To study the uptake of nCISP and CISP in human blood, blood was obtained from healthy male donors. The collection of blood samples from healthy human was approved by the Ethics Committee of West China Fourth Hospital and West China School of Public health, Sichuan University. Approved number, Gwll2022117. Informed consent was obtained from the donors. Blood was collected in tubes containing the anticoagulant lepirudin, which does not affect the complement system[8]. Then, 0.5 mL of fresh blood and 50 µL of DiD-labeled nCISP and CISP were mixed and incubated in 37 °C water bath in dark for 0.5 h. Monocytes in blood were then separated using Human Monocyte Extraction Kit, and the uptake of nanoparticles by monocytes was measured by flow cytometry. The plasma was collected and the fluorescence images were taken by IVIS imaging. The experiment was designed without considering the sex of human, and blood of male human was selected to ensure gender uniformity.

## Bonding of complement C3 to CISP and nCISP

Serum of male C57BL/6 mice ($18 \pm 2$ g, 6–8 weeks) was collected, and 0.3 mL of serum was mixed with 0.1 mL of nCISP or CISP, and incubated at 37 °C for 0.5 h. Then centrifuged at $20,000 \times g$ for 30 min to separate nCISP and CISP, washed them with cold PBS twice. Western Blot was used to assay C3 absorbed on CISP and nCISP.

## Analysis of serum proteins absorbed on CISP and nCISP

Serum of male C57BL/6 mice ($18 \pm 2$ g, 6–8 weeks) was collected, and 0.3 mL of serum was mixed with 0.1 mL of nCISP or CISP, and incubated at 37 °C for 0.5 h. Then centrifuged at $20,000 \times g$ for 30 min to separate nCISP and CISP, washed them with cold PBS twice. The proteins of CISP and nCISP were determined by protein mass

spectrometry (Q-Exactive, Thermo Fisher), and the level of serum proteins adsorbed by nCISP and CISP were then analysis.

### In vivo anticancer treatment

B16F10 tumor model: Subcutaneously inoculated B16F10 cells ($7 \times 10^5$ cells/mouse) in male C57BL/6 mice ($18 \pm 2$ g, 6−8 weeks) ($n = 8$). Drugs were intravenously injected (SR-717 10 mg/kg) on days 8, 10 and 12. The tumor was irradiated with 808 nm laser for 5 minutes (1.0 W/cm$^2$) at 12 h post the first injection. Mice were sacrificed when the tumor volume[58] is >2000 mm$^3$. Tumor volume was calculated according to Eq. (2).

$$\text{Tumor volume} = \text{length} \times \text{width} \times \text{width}/2 \qquad (2)$$

MC38 tumor model: Subcutaneously inoculated MC38 cells ($5 \times 10^5$ cells/mouse) in male C57BL/6 mice ($18 \pm 2$ g, 6−8 weeks) ($n = 7$). Drugs were intravenously injected (SR-717 10 mg/kg) on days 6, 8, 10, 12, 14, and 16. Mice were sacrificed when the tumor volume is greater than 2000 mm$^3$. Tumor volume was calculated according to Eq. (2).

LLC tumor model: Subcutaneously inoculated LLC cells ($5 \times 10^5$ cells/mouse) in male C57BL/6 mice ($18 \pm 2$ g, 6−8 weeks) ($n = 8$). Drugs were intravenously injected (SR-717 10 mg/kg) on the days 12, 14, and 16. Mice were sacrificed when the tumor volume is greater than 2000 mm$^3$. Tumor volume was calculated according to Eq. (2).

Metastatic melanoma model: B16F10 cells ($7 \times 10^5$ cells/mouse) were intravenously infused into male C57BL/6 mice ($18 \pm 2$ g, 6−8 weeks) ($n = 5$). Drugs were intravenously injected (SR-717 10 mg/kg) on days 8, 10, 12, 14, and 16.

### Flow cytometry analysis of immune cells

Tumor, tumor-draining lymph node (TdLNs) and spleen were harvested immediately after mice sacrifice. TdLNs were squeezed to filter through a 70 μm cell sieve, cells were resuspended in a PBS buffer. Spleens were squeezed to filter through a 70 μm cell sieve, cells were resuspended in ammonium-chloridepotassium (ACK) lysing buffer for 30 min. Tumors were squeezed to filter through a 70 μm cell sieve, cells were resuspended in ACK lysing buffer for 30 min. The obtained cells were stained with the relevant antibodies, and cell were then detected on a BD flow cytometer. The obtained flow cytometry data was analyzed by Flowjo V10.

### Statistical analysis

Statistical analysis was performed using Graphpdad Prism (Version 8.0.2). All quantitative parameters were presented as mean with Standard Error of Mean. For two-group comparison, Student's two-sided $t$ test was performed for the statistical analysis. For multiple comparisons, the data was calculated by one-way or two-way ANOVA test. Survival was measured using the Kaplan−Meier method and statistical significance was calculated by log-rank test. $P < 0.05$ was considered statistically significant.

### Reporting summary

Further information on research design is available in the Nature Portfolio Reporting Summary linked to this article.

## Data availability

The Proteomics data generated in this study have been deposited in the PRIDE repository under the accession code [https://www.ebi.ac.uk/pride/archive/projects/PXD044151]. The source data underlying Figs. 1d, 1f, 1g, 2b, 2d−g, 3b, 3d−f, 4b, 4c, 4g, 4i, 4k, 4m, 4o, 5b, 5c, 5d, 5f, 5h, 6b−q, 7b−j, Supplementary Fig. 2, 3b−d, 4, 5, 6, 7a, 7b, 8a, 9b, 10b, 11b, 13b, 14, 16c, 16e, 17, 18b, 18d, 18f, 19b, 20d, 20f, 20h, 20j, 21b, 22b, 23, 24, 25c, 25e, 25g, 25i, 26, 27a, 27b, 28b, 30a−c, 31a, 31b, 32a−d, 33a, 33b, 35, 36b, 38a−c, 39a−d, 40a−c, 41b, 42a−e, 44, 45b, 45d−n, 47, 48b, 49b, 49e−h, 50a−c, 51a−f, 52a−f, 53a−f are provided as a Source Data file. The remaining data are available within the Article, Supplementary Information, or Source Data file. Source data are provided with this paper.

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

## Acknowledgements

We acknowledge the support from the National Science Fund for Excellent Young Scholars (No. 82022070 (L.Z.)), the Regional Innovation and Development Joint Fund (No. U20A20411 (Z.Z.)), and the Innovation Program of Med-X Center for Materials from Sichuan University (No. MCM202103 (L.Z.)).

## Author contributions

L.Z. and L.L. designed this research. L.L. and M.Z. performed and analyzed most experiments. Z.Z., L.F., Q.B., X.L., J. Li, T.L., J. Long, and Z.L. assisted in analyzing the results and doing experiments. S.H. assisted in checking some spelling errors in the manuscript. L.L., L.Z., and M.Z. prepared the manuscript. All authors have given approval to the final version of the manuscript.

## Competing interests

The authors declare no competing interests.
