## [Peer Review File · Nature Communications]

Cholesterol removal improves performance of a model biomimetic system to co-deliver a photothermal agent and a STING agonist for cancer immunotherapyREVIEWER COMMENTS

Reviewer #1 (Remarks to the Author): with expertise in cancer nanotherapy

In this manuscript, the authors developed a membrane-coated nano vehicle with enhanced circulating time and tumor-targeting capability by reducing cholesterol content with simple (2-hydroxypropyl)- β -cyclodextrin treatment. This biomimetic nano delivery system could achieve effective elimination of melanoma tumors with no recurrence for >100 days in >80% treated mice. Although there are some positive attributes, previous studies have reported that reducing cholesterol levels effectively enhances membrane permeability (e.g. doi.org/10.1002/anie.202203115; Nat Biomed Eng, 5, 1411-1425, 2021). Therefore, the article lacks enough novelty to be published in Nature communications. Also, there are others concerns that need to be appropriately addressed

1. The in vivo anticancer study lacks rigor and misses important immune mechanistic investigation. Only several basic and ordinary results are presented in this part, which could not make readers convinced that the CISP can exert a potent tumor immunoresponse.
2. The T cells analysis in this study is rather weak. What is the T cells status after vaccination? What are the T cells subsets in the tumor tissue after treatment? Memory T cells and immune suppressive cells should be thoroughly analyzed.
3. A control group without a STING agonist should be added to assess the SR717 effect.
4. Blood chemistry and liver functions before and after vaccination should be provided to assess the potential toxicity in vivo.
5. Activation of the cGAS-STING pathway produces type I interferons, so type I interferons expression levels should be measured in vivo. Moreover, the expression of key immune activation regulator including IL-1 β , IL-6, TNF- α and IL-12 should be quantitatively analyzed.
6. In clinical patients, the late stage of the tumor causes lung metastasis. Can this delivery system inhibit lung metastasis?

Reviewer #2 (Remarks to the Author): with expertise in cancer nanotherapy, immuno-engineering, STING

Summary: The manuscript by Li et al. describes a nanoparticle delivery for a small molecule STING agonist (SR717) to be used with photothermal therapy (PTT). The particle is design is quite complicated, using a core of quercetin and iron (for the PTT) that is loaded with SR717 and then coated with the cell membrane of murine T cells (CTLL-2). The primary variables being explored here are whether the particles are formed with T cells that overexpress PD-1 – to target and block PD-L1 -- and the amount of cholesterol that has been removed from these T cells prior to particle formation. The authors evaluate the different formulations with respect to tumor tropism, phagocyte clearance, pharmacokinetics, and antitumor efficacy. The primary conclusion is that particles assembled from T cell membranes with cholesterol depletion demonstrate a higher degree of tumor uptake and ultimately therapeutic efficacy when combined with PTT, which the authors attribute to reduced phagocyte clearance via complement pathway activation.

Evaluation: While the finding that assembly of these particles from cell membranes with reduced cholesterol is interesting and is an important finding for the design of cell-membrane wrapped nanoparticles, the work suffers in several major ways that limit its impact and suitability for publication in Nature Communications. First, the innovation is modest given that a very similar type of T cell membrane coating strategy has recently been reported (Zhai et al, Nature Nanotechnology, 2021) using a different interferon inducer. Second, as will be discussed more below, several important controls are missing from some experiments and conclusions are not always strongly supported by the data. Third, the immune analysis performed is somewhat superficial, the system is only evaluated in a single tumor model, and the potential for translational impact is low due to the complexity of the system as well as the need to combine with local PTT.

Major Points:

1. For all studies, show individual mice as data points on bar graphs and spider plots should be shown for tumor volume studies.
2. For studies in Figure 4C, the authors should quantify NP concentration using fluorescence

spectroscopy (for both blood and plasma) instead of IVIS imaging, which is not sufficiently quantitative without use of standards. They should also calculate a half-life and other relevant pharmacokinetic parameters (e.g., AUC) from their data. The data in 4C doesn't suggest a major difference in half-life between nCISP and CISP and so conclusions regarding this should be strengthened.

3. There are several issues with the immunophenotyping performed in Figure 5.

a. Some metric of T cell function/activation/proliferation should be included in the analysis as total T cell infiltrate, particularly for CD4 T cells, isn't particularly informative.

b. In 4H, are these CD11b+F4/80+CD80+ cells? Was CD11b+ included in the panel as well? Was CD86 also looked at – this would be a more accepted marker of M1-like macrophages.

c. Data in 4J isn't very convincing; quantification and additional assay to quantify INF-gamma levels is needed.

d. The flow data shown in Figure S16 does not appear to be properly compensated; there should be clearly resolved populations of CD4 and CD8 T cells. Plot CD4 vs. CD8 to ensure clearly resolved populations.

4. The therapeutic data in Figure K-L is encouraging but there are several issues to address:

a. Statistical comparisons need to be included for tumor growth and survival.

b. It is surprising that SR717 alone didn't have an effect considering that the original report that describes this molecule tested this agent in the B16 model. That group used SR717 at a dose of 30 mg/kg whereas it is reduced here to 10 mg/kg; thus, it is not clear that the carrier platform is enhancing the efficacy of the drug. This would be a remarkably complicated design if the same effect could be achieved by increasing the free drug dose 2-3x.

c. It would be ideal repeat this study in at least one more tumor model.

5. Related to above, several important controls are missing from the study in Figure 5. First, does the SR717 need to be loaded with the PTT agents? To assess this, free SR717 + 'empty' CISP+L would need to be tested. Second, it is not clear what role the PTT plays here as the CISP system is not tested w/o PTT; the impact of the work would be much higher if PTT was not needed. The current study lacks sufficient controls to understand the contribution of all of the various components of the system and their role in mediating antitumor effects.

6. Even with the best particle, most of the particle goes to the liver and the lung (Fig. S7, S11). Is STING activated in these tissues?

7. Further image analysis and quantification is necessary in Figure 3C to demonstrate that

the particles trigger lysosomal degradation of PD-L1; the data as presented is not convincing enough to draw that conclusion.

8. In Figure 5E, how is it known that these are tumor cells? Or is this just referring to all cells in the tumor (e.g., including cancer cells, myeloid cells, etc?)

9. In vivo, the addition of PD1 didn't improve tumor accumulation/efficacy unless the cholesterol was removed. However, CSP not included as a control in the in vivo studies and so ultimately the contribution of cholesterol depletion vs PD1 expression could not be clearly resolved. The authors should repeat a subset of the experiments with this control to determine if PD1 expression is a critical component of the design or if the improved distribution/PK properties associated with cholesterol depletion are sufficient.

Minor Points:

1. In Figure 5K-L, it would be helpful to include a timeline of when PTT was applied and when particle treatments were applied in comparison.

2. Figure 1H – the significance of this data isn't clear; this seems like something that should just be in the SI unless it serves to demonstrate expression pattern differences.

Reviewer #3 (Remarks to the Author): with expertise in cancer nanotherapy, immuno-engineering, STING

In this manuscript, the authors develop a biomimetic nanoparticle (CISP) that contains a photothermal agent (quercetin & iron), a STING agonist (SR-717), and is coated with an immune checkpoint blockade agent (PD-1). The quercetin, iron, and SR-717 is loaded into a PLGA nanoparticle that is then coated with a biological membrane sourced from CTLL-2 T cells overexpressing PD-1. The membrane is treated with β -cyclodextrin to lower the cholesterol content which was demonstrated to lower the uptake of the nanoparticle by the monocytes in the blood by 50%. The authors show that the CISP platform delivers SR-717 and quercetin/iron to tumor site to activate local STING pathway, while the PD-1 on CISP surface blocked the up-regulated PD-L1 in tumor cells induced by STING agonist. CISP treated mice showed increased anti-tumor effects with >80% of the mice showing no sign of recurrence with the combination of PTT. Authors conclude that the removal of cholesterol could also potentially be an effective membrane modification strategy for drug delivery

applications outside of cancer.

The following revisions are recommended:

1) What is the encapsulation efficiency or drug loading of quercetin, iron, and SR-717 in the nanoparticle? For iron and quercetin, if this is mentioned in the previous paper that can be referenced. However, since the SR-717 is new, it would be important to include this in Figure 1 (characterization of CISP) or at least in the supplemental figures.

2) A legend with all the different particles (nCSP, CIP, nCISP, etc.) and what they consist of would be helpful in Figure 1 to understand how it compares to CISP since there are many acronyms for each particle type.

3) The authors explained that while CISP and nCISP performed similarly in vitro with B16F10 tumor cells (Figure 2), CISP outperformed nCISP in vivo (Figure 3) and attribute this difference to monocyte clearance. More explanation or mechanistic understanding of the correlation between cholesterol and monocyte clearance would be helpful for the reader to be convinced of the reasoning behind the benefit of the cholesterol removal. For instance, do monocytes phagocytose cell membrane vesicles (without PLGA) differently with or without cholesterol? Is this due to membrane fluidity or due to specific interaction of cholesterol with receptors on monocytes?

4) In addition to the studies done with tumor cells, to supplement, the uptake of CISP and nCISP in dendritic cells could be evaluated for upregulation of maturation markers.

5) Just as human blood was used to assess the uptake of CISP and nCISP by monocytes, to be consistent with the previous study, it would be good to also test the uptake of in human tumor cell line to see if the same phenomenon holds true – that the performance of CISP and nCISP is similar when the blood flow factor is not considered.

6) The authors suggested that too low level of cholesterol may cause problem on membrane morphology maintenance and that studies have been done to optimize the β CD

concentration. These experiments should be included in the supplemental.

7) Methods section: What type of PLGA was used?

8) Schematic 1 uses the acronym "COM" for the membrane coated nanoparticle, but this is never referenced in the body of the text.

9) It would be good to show the dose used for each injection on the dosing scheme in Figure 5.

10) All gating schemes for flow cytometry should be provided in the supplemental information.

11) Statistical analysis: They used an unpaired two-tailed t-test was used to compare between two groups. They should consult with a biostatistician. One or two way ANOVA followed by a multiple comparison test should be used for analyzing most of their graphs.

Reviewer #4 (Remarks to the Author): with expertise in nanoparticle design, membrane coating

Cell membrane coating is one of the most promising ways to improve the performance of the nanoparticles/carriers for cancer therapy. The improved functionality obtained with cell membranes are related to increased systemic circulation time by reducing the effect of MPS and homologous targeting capability. The authors utilized cell membranes from T cell with overexpressed PD-1 to coat their NPs. These NPs were effective in photothermal conversion efficiency to realize PTT and NPs were also loaded with SR-717 to induce anti-tumoral immune response through STING activation. The idea of the authors was not to add anything in the delivery system but reduce the content of the cholesterol on the cell membrane used for the NP coating as cholesterol was previously observed to have negative effect on the targeting properties due to increased opsonization.

The authors were successful in proving their hypothesis with several and convincing results

they reported in the manuscript. Regarding the results, the authors presented appropriate statistical consideration. I see the results very significant for the scientists working with cell membrane coated delivery systems giving them one good tool to improve the performance of the delivery systems.

I have some comments to improve the quality of the manuscript:

- The authors should comments more closely what are the roles of PTT and STING activation to obtain effective antitumoral immune response i.e. good therapeutic outcome. Are they both needed?
- The first paragraph of Introduction is confusing. Are the authors writing about cell membrane coated NPs or other type of coatings?
- Chapter 2.5. What was the time point of laser irradiation?
- Lentivirus is indicated in Scheme 1. What is its role in the production of CTLL2-PD1?
- Figure 1A. What is n=5 referring to?
- Figure 1E. The authors should report some statistics what part of the NPs were coated with cell membranes and what is the quality of the coating (are the NPs fully or partly coated)?
- Figure 1F. It is very difficult to read the image i.e., to recognize the location of cell membrane on NPs.
- Figure 4A is messy and difficult to understand.

RESPONSE TO COMMENTS

We sincerely thank the comments made by the reviewers; many insightful comments have greatly helped us to improve our work.

The detailed point-to-point responses are addressed as followed:

REVIEWER COMMENTS

Reviewer #1 (Remarks to the Author): with expertise in cancer nanotherapy

In this manuscript, the authors developed a membrane-coated nano vehicle with enhanced circulating time and tumor-targeting capability by reducing cholesterol content with simple (2-hydroxypropyl)- β -cyclodextrin treatment. This biomimetic nano delivery system could achieve effective elimination of melanoma tumors with no recurrence for >100 days in >80% treated mice. Although there are some positive attributes, previous studies have reported that reducing cholesterol levels effectively enhances membrane permeability (e.g. doi.org/10.1002/anie.202203115; Nat Biomed Eng, 5, 1411-1425, 2021). Therefore, the article lacks enough novelty to be published in Nature communications. Also, there are others concerns that need to be appropriately addressed

Response: Thanks for pointing out the concern. Indeed, the first mentioned article (doi.org/10.1002/anie.202203115) showed that reduction of cholesterol levels could enhance membrane permeability, and increase activity of enzyme that coated by cell membrane. The second article showed that cancer cells treated with β -cyclodextrin became stiffened, thus augmented T-cell cytotoxicity and enhances the efficacy of adoptive T-cell therapy against solid tumors in mice (Nat Biomed Eng, 5, 1411-1425, 2021).

However, we would like to argue that our work still has proper novelty for several reasons. First, the enhancement of biological membranes performance (targeting ability) reported by us is not due to the aforementioned mechanisms (change in membrane permeability and stiffness). In fact, we have discovered a different mechanism (immune surveillance escaping) underlying the phenomenon and briefly revealed the reverse relationship between clearance time in blood and cholesterol level. We suppose that this is an important clue for future research on this topic and is carrying out following studies.

Second, by constructing proof-of-concept model system (i.e., CISP) and examine its effects

in vitro and *in vivo*, we verified the versatility of the cholesterol removal strategy. The loss of targeting ability of nanomedicines *in vivo* has been a big challenge, our finding may also have important applicability like enhancing enzyme activity (as mentioned in the first report). We would suggest that if the membrane-coated enzyme systems need to work in blood flow *in vivo*, the elongated circulation time caused by lower cholesterol contents definitely plays a part in its performance improvement.

Third, also based on our proof-of-concept system, we showed that this subtractive strategy did not affect the functionality of engineered surface PD-1 nor the tumor targeting capability. This characteristic may help avoid the dilemma for additive strategy that adding more and longer surface disguise molecules could improve circulation time but affect surface functional molecules at the same time. Thus, this strategy has potential to be applied in parallel to many different designs for many different purposes.

1. The *in vivo* anticancer study lacks rigor and misses important immune mechanistic investigation. Only several basic and ordinary results are presented in this part, which could not make readers convinced that the CISP can exert a potent tumor immunoresponse.

Response: Many thanks for your suggestion. We conducted further more detailed evaluations of the immune response in tumors again in B16F10 melanoma tumors, and the results are presented below:

With the combined therapeutic effects, CISP significantly increased the number of CD8 T cells, CD69⁺ CD8 T cells, Granzyme B⁺ CD8 T cells, Ki67⁺ CD8 T cells and CD4 T cells in tumors (Fig. 6d-h and Supplementary Fig. 29b, c). The maturation of DCs, M1 macrophages, memory T cells in tumors, and activated T cells in tumor draining lymph nodes (TdLNs) were also significantly increased (Supplementary Fig. 30a-c, Supplementary Fig. 31a, Supplementary Fig. 32a-d and Supplementary Fig. 33a, b). Besides, Treg cells and M2 macrophages in tumors were decreased (Fig. 6i, Supplementary Fig. 31b), and the level of IL-12, IL-6, IFN γ , IFN β , IFN α and TNF α in tumors were elevated (Fig. 6j-o). In contrast, other groups only showed limited effect on activating anti-tumor immune response (Fig. 6d-o). It is worth noting that although CIP (denotes low-Cholesterol membrane coated ICB agent and Photothermal agent) could also reduce PD-L1 in tumors, the final outcome is unsatisfactory, indicating that SR-717 is necessary

to boost the immune responses (Fig. 1j, Fig. 2f and Fig. 6c-o). Ultimately, ~85% of the mice in CISP group were cured with no tumor recurrence on day 80, which is the best outcome (Fig. 6p, q, Supplementary Fig. 34, Supplementary Fig. 35). In contrast, nCSP, CSP and nCISP displayed weaker therapeutic efficiency, with a survival rate of ~20%; and no mice survived with free SR-717 treatment, reflecting the importance of the delivery system. Noticeably, the tumor recurrence rate in CISP group was lower than CIP, again illustrated the important role of SR-717 in this system. Taken together, the CISP which utilized low-cholesterol membrane showed higher delivery performance and better therapeutic outcomes combined with PTT in mice bearing melanoma.

Fig. 6 CISP enhanced anti-tumor immunity in mice bearing melanoma. **a** Schematic illustration showing the experiment process. Red arrows indicate administration of nanomedicines. The dosage of SR-717 is 10 mg/kg. **b** Western blot analysis of p-TBK1, TBK1, p-IRF3 and IRF3 in tumors at 6 days post laser irradiation. **c** Flow-cytometric analysis of CD44⁺PD-L1⁺ cells in B16F10 melanoma tumors at 6 days post laser irradiation. Data represent mean \pm SEM, $n = 4$ independent animals. **d** Flow-cytometric analysis of CD3⁺CD8⁺ cells in B16F10 melanoma tumors. Data represent mean \pm SEM, $n = 4$ independent animals. **e** Flow-cytometric analysis of CD69⁺ cells in CD8 T cells in B16F10 melanoma tumors. Data represent mean \pm SEM, $n = 4$ independent animals. **f** Flow-cytometric analysis of Granzyme B⁺ cells in CD8 T cells in B16F10 melanoma tumors. Data represent mean \pm SEM, $n = 4$ independent animals. **g** Flow-cytometric analysis of Ki67⁺ cells in CD8 T cells in B16F10 melanoma tumors. Data represent mean \pm SEM, $n = 4$ independent animals. **h** Flow-cytometric analysis of CD3⁺CD4⁺ cells in B16F10 melanoma tumors. Data represent mean \pm SEM, $n = 4$ independent animals. **i** Flow-cytometric analysis of Foxp3⁺ cells in CD4 T cells in B16F10 melanoma tumors. Data represent mean \pm SEM, $n = 4$ independent animals. **j, k, l, m, n, o** IL-12 (**j**), L-6 (**k**), IFN γ (**l**), IFN β (**m**), IFN α (**n**) and TNF α (**o**) at 6 days post laser irradiation. Data represent mean \pm SEM, $n = 6$ independent animals. **p** Tumor growth curve of mice bearing melanoma. Red arrows indicate administration of nanomedicines. Data represent mean \pm SEM, $n = 8$ independent animals. **q** Survival curve of mice bearing melanoma, $n = 8$ independent animals. QFN, Quercetin-ferrum nanoparticles; QFN₇₁₇, QFN loaded with SR-717; nCSP, normal-Cholesterol cell membrane coated QFN₇₁₇; CSP, low-Cholesterol membrane coated QFN₇₁₇; nCISP, normal-Cholesterol cell membrane coated ICB agent and QFN₇₁₇; CISP, low-Cholesterol membrane coated ICB agent and QFN₇₁₇; CIP, low-Cholesterol membrane coated ICB agent and QFN.

Supplementary Figure 29. The antitumor immunity activation in B16F10 melanoma tumors. **a** Flow-cytometric analysis of CD44⁺PD-L1⁺ cells in tumors at 6 days post laser irradiation. $n = 4$ independent animals. **b** Flow-cytometric analysis of CD3⁺CD8⁺ cells in tumors at 6 days post laser irradiation. $n = 4$ independent animals. **c** Flow-cytometric analysis of CD3⁺CD4⁺ cells in tumors at 6 days post laser irradiation. $n = 4$ independent animals. L,

laser; QFN, Quercetin-ferrum nanoparticles; QFN₇₁₇, QFN loaded with SR-717; nCSP, normal-Cholesterol cell membrane coated QFN₇₁₇; CSP, low-Cholesterol membrane coated QFN₇₁₇; nCISP, normal-Cholesterol cell membrane coated ICB agent and QFN₇₁₇; CISP, low-Cholesterol membrane coated ICB agent and QFN₇₁₇; CIP, low-Cholesterol membrane coated ICB agent and QFN.

Supplementary Figure 30. Maturation of DCs in B16F10 melanoma tumors. **a, b, c** Flow-cytometric analysis of CD11c⁺CD40⁺ cells (**a**), CD11c⁺CD86⁺ cells (**b**) and CD11c⁺CD80⁺ cells (**c**) in B16F10 melanoma tumors at 6 days post laser irradiation. Data represent mean ± SEM, *n* = 4 independent animals. L, laser; QFN, Quercetin-ferrum nanoparticles; QFN₇₁₇, QFN loaded with SR-717; nCSP, normal-Cholesterol cell membrane coated QFN₇₁₇; CSP, low-Cholesterol membrane coated QFN₇₁₇; nCISP, normal-Cholesterol cell membrane coated ICB agent and QFN₇₁₇; CISP, low-Cholesterol membrane coated ICB agent and QFN₇₁₇; CIP, low-Cholesterol membrane coated ICB agent and QFN.

Supplementary Figure 31. Macrophages in B16F10 melanoma tumors. **a** Flow-cytometric analysis of CD80⁺CD86⁺ cells in CD11b⁺F4/80⁺ cells (M1) in B16F10 melanoma tumor at 6 days post laser irradiation. Data represent mean ± SEM, *n* = 4 independent animals. **b** Flow-cytometric analysis of CD206⁺F4/80⁺ cells in CD11b⁺ cells (M2) in B16F10 melanoma tumor at 6 days post laser irradiation. Data represent mean ± SEM, *n* = 4 independent animals. L, laser; QFN, Quercetin-ferrum nanoparticles; QFN₇₁₇, QFN loaded with SR-717; nCSP, normal-

Cholesterol cell membrane coated QFN₇₁₇; CSP, low-Cholesterol membrane coated QFN₇₁₇; nCISP, normal-Cholesterol cell membrane coated ICB agent and QFN₇₁₇; CISP, low-Cholesterol membrane coated ICB agent and QFN₇₁₇; CIP, low-Cholesterol membrane coated ICB agent and QFN.

Supplementary Figure 32. Memory T cells in B16F10 melanoma tumors. a, b, c, d Memory T cells including CD8⁺Tcm (a), CD8⁺Tem (b), CD4⁺Tcm (c) and CD4⁺Tem (d) in tumors were examined by flow cytometer. Tumors were harvested at 6 days post laser irradiation. Data represent mean ± SEM, *n* = 4 independent animals. L, laser; QFN, Quercetin-ferrum nanoparticles; QFN₇₁₇, QFN loaded with SR-717; nCSP, normal-Cholesterol cell membrane coated QFN₇₁₇; CSP, low-Cholesterol membrane coated QFN₇₁₇; nCISP, normal-Cholesterol cell membrane coated ICB agent and QFN₇₁₇; CISP, low-Cholesterol membrane coated ICB agent and QFN₇₁₇; CIP, low-Cholesterol membrane coated ICB agent and QFN.

Supplementary Figure 33. Activated T cells in TdLNs in mice bearing B16F10 melanoma tumor. a Flow-cytometric analysis of CD69⁺ cells in CD3⁺CD8⁺ cells in TdLNs at 6 days post laser irradiation. Data represent mean ± SEM, *n* = 4 independent animals. **b** Flow-cytometric analysis of CD69⁺ cells in CD3⁺CD4⁺ cells in TdLNs at 6 days post laser irradiation. Data represent mean ± SEM, *n* = 4 independent animals. L, laser, QFN, Quercetin-ferrum nanoparticles; QFN₇₁₇, QFN loaded with SR-717; nCSP, normal-Cholesterol cell membrane coated QFN₇₁₇; CSP, low-Cholesterol membrane coated QFN₇₁₇; nCISP, normal-Cholesterol cell membrane coated ICB agent and QFN₇₁₇; CISP, low-Cholesterol membrane coated ICB agent and QFN₇₁₇; CIP, low-Cholesterol membrane coated ICB agent and QFN; TdLNs, Tumor draining lymph nodes.

2. The T cells analysis in this study is rather weak. What is the T cells status after vaccination? What are the T cells subsets in the tumor tissue after treatment? Memory T cells and immune suppressive cells should be thoroughly analyzed.

Response: Thanks for your suggestion. We performed the *in vivo* experiments again and analyzed the requested T cells. The related contents are presented in response for comment 1.

3. A control group without a STING agonist should be added to assess the SR717 effect.

Response: We really appreciate your advice. According to your request, CIP (denotes low-Cholesterol membrane coated ICB agent and Photothermal agent) nanoparticles were fabricated and correspondent experiments were conducted. The results supported our rational-based design, and related text are copied below for your convenience:

Drugs were intravenously injected on the day 8, 10 and 12, and one laser irradiation was applied to activate photothermal therapy at 12 h post the first injection (Fig. 6a)¹. Western blot results demonstrated that CISP significantly increased the expression of p-TBK1 and p-IRF3 in tumors, indicating that STING signal pathway in tumor sites was activated (Fig. 6b). In comparison, free SR-717, QFN₇₁₇, nCSP, and nCISP all displayed limited effect (Fig. 6b). Similarly, CISP also significantly reduced the expression of PD-L1 in tumor cells while nCSP, CSP and nCISP showed weaker effects (Fig. 6c and Supplementary Fig. 29a). With the combined therapeutic effects, CISP significantly increased the number of CD8 T cells, CD69⁺ CD8 T cells, Granzyme B⁺ CD8 T cells, Ki67⁺ CD8 T cells and CD4 T cells in tumors (Fig. 6d-h and Supplementary Fig. 29b, c). The maturation of DCs, M1 macrophages, memory T cells in tumors, and activated T cells in tumor draining lymph nodes (TdLNs) were also significantly increased

(Supplementary Fig. 30a-c, Supplementary Fig. 31a, Supplementary Fig. 32a-d and Supplementary Fig. 33a, b). Besides, Treg cells and M2 macrophages in tumors were decreased (Fig. 6i, Supplementary Fig. 31b), and the level of IL-12, IL-6, IFN γ , IFN β , IFN α and TNF α in tumors were elevated (Fig. 6j-o). In contrast, other groups only showed limited effect on activating anti-tumor immune response (Fig. 6d-o). It is worth noting that although CIP (denotes low-Cholesterol membrane coated ICB agent and Photothermal agent) could also reduce PD-L1 in tumors, the final outcome is unsatisfactory, indicating that SR-717 is necessary to boost the immune responses (Fig. 1j, Fig. 2f and Fig. 6c-o). Ultimately, ~85% of the mice in CISP group were cured with no tumor recurrence on day 80, which is the best outcome (Fig. 6p, q, Supplementary Fig. 34, Supplementary Fig. 35). In contrast, nCSP, CSP and nCISP displayed weaker therapeutic efficiency, with a survival rate of ~20%; and no mice survived with free SR-717 treatment, reflecting the importance of the delivery system. Noticeably, the tumor recurrence rate in CISP group was lower than CIP, again illustrated the important role of SR-717 in this system.

Fig. 6 CISP enhanced anti-tumor immunity in mice bearing melanoma. **a** Schematic illustration showing the experiment process. Red arrows indicate administration of nanomedicines. The dosage of SR-717 is 10 mg/kg. **b** Western blot analysis of p-TBK1, TBK1, p-IRF3 and IRF3 in tumors at 6 days post laser irradiation. **c** Flow-cytometric analysis of CD44⁺PD-L1⁺ cells in B16F10 melanoma tumors at 6 days post laser irradiation. Data represent mean \pm SEM, $n = 4$ independent animals. **d** Flow-cytometric analysis of CD3⁺CD8⁺ cells in B16F10 melanoma tumors. Data represent mean \pm SEM, $n = 4$ independent animals. **e** Flow-cytometric analysis of CD69⁺ cells in CD8 T cells in B16F10 melanoma tumors. Data represent mean \pm SEM, $n = 4$ independent animals. **f** Flow-cytometric analysis of Granzyme B⁺ cells in CD8 T cells in B16F10 melanoma tumors. Data represent mean \pm SEM, $n = 4$ independent animals. **g** Flow-cytometric analysis of Ki67⁺ cells in CD8 T cells in B16F10 melanoma tumors. Data represent mean \pm SEM, $n = 4$ independent animals. **h** Flow-cytometric analysis of CD3⁺CD4⁺ cells in B16F10 melanoma tumors. Data represent mean \pm SEM, $n = 4$ independent

animals. **i** Flow-cytometric analysis of Foxp3⁺ cells in CD4 T cells in B16F10 melanoma tumors. Data represent mean \pm SEM, $n = 4$ independent animals. **j, k, l, m, n, o** IL-12 (**j**), L-6 (**k**), IFN γ (**l**), IFN β (**m**), IFN α (**n**) and TNF α (**o**) at 6 days post laser irradiation. Data represent mean \pm SEM, $n = 6$ independent animals. **p** Tumor growth curve of mice bearing melanoma. Red arrows indicate administration of nanomedicines. Data represent mean \pm SEM, $n = 8$ independent animals. **q** Survival curve of mice bearing melanoma, $n = 8$ independent animals. QFN, Quercetin-ferrum nanoparticles; QFN₇₁₇, QFN loaded with SR-717; nCSP, normal-Cholesterol cell membrane coated QFN₇₁₇; CSP, low-Cholesterol membrane coated QFN₇₁₇; nCISP, normal-Cholesterol cell membrane coated ICB agent and QFN₇₁₇; CISP, low-Cholesterol membrane coated ICB agent and QFN₇₁₇; CIP, low-Cholesterol membrane coated ICB agent and QFN.

Considering that PTT is not suitable for some cancers in clinical practice, we then examined the effect of CISP on some other tumor models. Here, colon cancer cells (MC38) (5×10^5 cells/mouse) were inoculated subcutaneously in C57BL/6 mice (Fig. 7a). Fluorescence images show that tumor accumulation of CISP was higher than that of nCISP, CSP and nCSP (Fig. 7b, c). Similarly, CISP and CIP also significantly reduced the expression of PD-L1 in tumor cells while nCSP, CSP and nCISP showed weaker effects (Fig. 7d and Supplementary Fig. 36a). CISP increased CD8 T cells in MC38 tumors, and CSP also improved it to some extent, but is weaker than CISP (Fig. 7e and Supplementary Fig. 36b). More specifically, CISP treatment resulted in the strongest increase of CD8 T cells, activated CD8 T cells, CD4 T cells, M1 Macrophages, NK cells and matured DCs in MC38 tumors (Fig. 7e-i, Supplementary Fig. 36c-g and Supplementary Fig. 37a-f). Maturation of DCs, activated CD4 T cells and CD8 T cells in TdLNs (Supplementary Fig. 38a-h); as well as activated CD4 T cells and CD8 T cells in splenocytes all displayed highest increase in CISP group (Supplementary Fig. 39a-c). CISP caused significant up-regulation of the CD8⁺/CD4⁺ ratio in TdLNs, which was reported to associated with the prognosis of a variety of cancers and the response to immunotherapy² (Supplementary Fig. 40a, b). The changes in IL-12, IL-6, IFN γ , IFN β , TNF α , and Treg cells in tumors were also the greatest in CISP group (Fig. 7j, Supplementary Fig. 41a-e). In contrast, other groups including CIP and CSP showed limited activation of anti-tumor immune response (Fig. 7e-j and Supplementary Fig. 41a-e). Ultimately, CISP caused the strongest MC38 tumor suppression, and ~40% of treated mice survived on day 80, while all mice in other groups died (Fig. 7k-m and Supplementary Fig. 42). Thus, CISP is also significantly more effective than other systems in treating MC38 colon cancer, even without the introduction of PTT.

Fig. 7 CISP enhanced anti-tumor immunity in mice bearing MC38 tumor. **a** Schematic illustration showing the experiment process. Red arrows indicate administration of nanomedicines. The dosage of SR-717 is 10 mg/kg. **b, c** Fluorescence images of MC38 tumors at 6 h post treated with of nCSP, CSP, nCISP and CISP (**b**) and mean fluorescence intensity (MFI) of DiD in tumors (**c**). Data represent mean \pm SEM, $n = 3$ independent animals. **d** Flow-cytometric analysis of CD44⁺PD-L1⁺ cells in MC38 tumors on day 18. Data represent mean \pm SEM, $n = 4$ independent animals. **e** Flow-cytometric analysis of CD3⁺CD8⁺ cells in MC38 tumors. Data represent mean \pm SEM, $n = 4$ independent animals. **f** Flow-cytometric analysis of CD69⁺ cells in CD8 T cells in MC38 tumors. Data represent mean \pm SEM, $n = 4$ independent animals. **g** Flow-cytometric analysis of Granzyme B⁺ cells in CD8 T cells in MC38 tumors. Data represent mean \pm SEM, $n = 4$ independent animals. **h** Flow-cytometric analysis of Ki67⁺ cells in CD8 T cells in MC38 tumors. Data represent mean \pm SEM, $n = 4$ independent animals. **i** Flow-cytometric analysis of CD3⁺CD4⁺ cells in MC38 tumors. Data represent mean \pm SEM, $n = 4$ independent animals. **j** Flow-cytometric analysis of Fopx3⁺ cells in CD4 T cells in MC38 tumors. Data represent mean \pm SEM, $n = 4$ independent animals. **k** Images of MC38 tumors at

14 days and 24 days post tumor inoculation. $n = 7$ independent animals. Scale bar = 0.5 cm. **I** Tumor growth curve of mice bearing MC38 tumors. Red arrows indicate administration of nanomedicines. Data represent mean \pm SEM, $n = 7$ independent animals. **m** Survival curve of mice bearing MC38 tumors. $n = 7$ independent animals. QFN, Quercetin-ferrum nanoparticles; QFN₇₁₇, QFN loaded with SR-717; nCSP, normal-Cholesterol cell membrane coated QFN₇₁₇; CSP, low-Cholesterol membrane coated QFN₇₁₇; nCISP, normal-Cholesterol cell membrane coated ICB agent and QFN₇₁₇; CISP, low-Cholesterol membrane coated ICB agent and QFN₇₁₇; CIP, low-Cholesterol membrane coated ICB agent and QFN.

4. Blood chemistry and liver functions before and after vaccination should be provided to assess the potential toxicity in vivo.

Response: Thanks for your suggestion. We have conducted correspondent experiments and related results are copied below:

Blood routine examination and blood biochemistry (including ALT, AST, LDH, CREA, UREA and CHE) of mice bearing MC38 tumor was all in the normal range treated with CISP (Supplementary Fig. 49a-f and Supplementary Fig. 50a-f). Blood biochemistry of mice bearing LLC tumor was also in the normal range treated with CISP (Supplementary Fig. 51a-f).

Supplementary Figure 50. Blood biochemistry of mice bearing MC38 tumor at 18 days post tumor inoculation. **a** The serum level of ALT in mice receiving the indicated treatment. Data represent mean \pm SEM, $n = 6$ independent animals. **b** The serum level of AST in mice receiving the indicated treatment. Data represent mean \pm SEM, $n = 6$ independent animals. **c** The serum level of LDH in mice receiving the indicated treatment. Data represent mean \pm SEM, $n = 6$ independent animals. **d** The serum level of CHE in mice receiving the indicated treatment. Data represent mean \pm SEM, $n = 6$ independent animals. **e** The serum level of UREA in mice

receiving the indicated treatment. Data represent mean \pm SEM, $n = 6$ independent animals. **f** The serum level of CREA in mice receiving the indicated treatment. Data represent mean \pm SEM, $n = 6$ independent animals. ALT, alanine aminotransferase; AST, aspartate aminotransferase; LDH, lactic dehydrogenase; CHE, Cholinesterase; CREA, Creatine. QFN, Quercetin-ferrum nanoparticles; QFN₇₁₇, QFN loaded with SR-717; nCSP, normal-Cholesterol cell membrane coated QFN₇₁₇; CSP, low-Cholesterol membrane coated QFN₇₁₇; nCISP, normal-Cholesterol cell membrane coated ICB agent and QFN₇₁₇; CISP, low-Cholesterol membrane coated ICB agent and QFN₇₁₇; CIP, low-Cholesterol membrane coated ICB agent and QFN.

Supplementary Figure 51. Blood biochemistry of mice bearing LLC tumor at 18 days post tumor inoculation. **a** The serum level of ALT in mice receiving the indicated treatment. Data represent mean \pm SEM, $n = 5$ independent animals. **b** The serum level of AST in mice receiving the indicated treatment. Data represent mean \pm SEM, $n = 5$ independent animals. **c** The serum level of LDH in mice receiving the indicated treatment. Data represent mean \pm SEM, $n = 5$ independent animals. **d** The serum level of CHE in mice receiving the indicated treatment. Data represent mean \pm SEM, $n = 5$ independent animals. **e** The serum level of UREA in mice receiving the indicated treatment. Data represent mean \pm SEM, $n = 5$ independent animals. **f** The serum level of CREA in mice receiving the indicated treatment. Data represent mean \pm SEM, $n = 5$ independent animals. ALT, alanine aminotransferase; AST, aspartate aminotransferase; LDH, lactic dehydrogenase; CHE, Cholinesterase; CREA, Creatine. QFN, Quercetin-ferrum nanoparticles; QFN₇₁₇, QFN loaded with SR-717; nCSP, normal-Cholesterol cell membrane coated QFN₇₁₇; CSP, low-Cholesterol membrane coated QFN₇₁₇; nCISP, normal-Cholesterol cell membrane coated ICB agent and QFN₇₁₇; CISP, low-Cholesterol membrane coated ICB agent and QFN₇₁₇; CIP, low-Cholesterol membrane coated ICB agent and QFN.

5. Activation of the cGAS-STING pathway produces type I interferons, so type I

interferons expression levels should be measured in vivo. Moreover, the expression of key immune activation regulator including IL-1 β , IL-6, TNF- α and IL-12 should be quantitatively analyzed.

Response: Many thanks for your suggestion. We have measured the mentioned molecules and the results are presented in the response for comment 1 (Fig. 6j-o).

6. In clinical patients, the late stage of the tumor causes lung metastasis. Can this delivery system inhibit lung metastasis?

Response: Thank you for rising the point, this is surely of clinical importance. We assessed the effect of CISP on lung metastasis using animal model, and results are presented below for your convenience:

Metastasis is a major cause of death for cancer patients and lung metastasis is common for melanoma³. Hence, we also examined the effect of CISP using lung metastatic melanoma mice model. Fluorescence images show that CISP could target to lung with metastasis sites (Supplementary Fig. 46a, b). Further examinations show that CISP performed better than CIP, nCISP, CSP and nCSP in reducing metastatic foci in lung (Supplementary Fig. 47a-c). Significantly higher CD8 T cells, matured DCs, M1 macrophages and IFN α could also be detected in CISP treated mice (Supplementary Fig. 47d-h). These results indicate that CISP could induce anti-tumor immune responses to inhibit metastatic melanoma.

Supplementary Figure 47. The therapeutic effects of CISP in mice bearing metastatic melanoma. **a** Schematic illustration of the experiment process. The dosage of SR-717 is 10 mg/kg. **b** Representative images of lung and the number of pulmonary metastasis foci at 18 days post B16F10 injection. Metastasis foci is pointed by arrow. Scale bar = 0.3 cm. Data represent mean \pm SEM, $n = 5$ independent animals. **c** Representative images of H&E staining of lung sections in mice. Metastasis foci is pointed by arrow. $n = 5$ independent animals. **d, e** Flow-cytometric analysis of CD3⁺CD8⁺ cell in lung at 18 days post B16F10 injection. Data represent mean \pm SEM, $n = 4$ independent animals. **f** Flow-cytometric analysis of CD11c⁺CD40⁺ cells in lung at 18 days post B16F10 injection. Data represent mean \pm SEM, $n = 4$ independent animals. **g** Flow-cytometric analysis of CD80⁺CD86⁺ cells in F4/80⁺CD11b⁺ cells in lung at 18 days post B16F10 injection. Data represent mean \pm SEM, $n = 4$ independent animals. **h** IFN α in lung at 18 days post B16F10 injection (iv.). Data represent mean \pm SEM, $n = 5$ independent animals. QFN, Quercetin-ferrum nanoparticles; QFN₇₁₇, QFN loaded with SR-717; nCSP, normal-Cholesterol cell membrane coated QFN₇₁₇; CSP, low-Cholesterol membrane coated QFN₇₁₇; nCISP, normal-Cholesterol cell membrane coated ICB agent and QFN₇₁₇; CISP,

low-Cholesterol membrane coated ICB agent and QFN₇₁₇; CIP, low-Cholesterol membrane coated ICB agent and QFN.

Reviewer #2 (Remarks to the Author): with expertise in cancer nanotherapy, immuno-engineering, STING

Summary: The manuscript by Li et al. describes a nanoparticle delivery for a small molecule STING agonist (SR717) to be used with photothermal therapy (PTT). The particle is design is quite complicated, using a core of quercetin and iron (for the PTT) that is loaded with SR717 and then coated with the cell membrane of murine T cells (CTLL-2). The primary variables being explored here are whether the particles are formed with T cells that overexpress PD-1 – to target and block PD-L1 -- and the amount of cholesterol that has been removed from these T cells prior to particle formation. The authors evaluate the different formulations with respect to tumor tropism, phagocyte clearance, pharmacokinetics, and antitumor efficacy. The primary conclusion is that particles assembled from T cell membranes with cholesterol depletion demonstrate a higher degree of tumor uptake and ultimately therapeutic efficacy when combined with PTT, which the authors attribute to reduced phagocyte clearance via complement pathway activation.

Evaluation: While the finding that assembly of these particles from cell membranes with reduced cholesterol is interesting and is an important finding for the design of cell-membrane wrapped nanoparticles, the work suffers in several major ways that limit its impact and suitability for publication in Nature Communications. First, the innovation is modest given that a very similar type of T cell membrane coating strategy has recently been reported (Zhai et al, Nature Nanotechnology, 2021) using a different interferon inducer. Second, as will be discussed more below, several important controls are missing from some experiments and conclusions are not always strongly supported by the data. Third, the immune analysis performed is somewhat superficial, the system is only evaluated in a single tumor model, and the potential for translational impact is low due to the complexity of the system as well as the need to combine with local PTT.

Response: We appreciate your insightful comments. First, we agree that the biological membrane wrapped nanoparticles have been investigated in many ways, and the literature

mentioned (i.e., Zhai et al, Nature Nanotechnology, 2021) has used a very similar membrane engineering as our proof-of-concept model system. However, our model system mainly worked as a proof to show that our subtractive cholesterol removal strategy is functional and operatable. The CISP system indeed provide evidences to support that the circulating time and targeting capability of biological membrane-coated nanovehicles could be significantly improved by reducing their cholesterol content with simple (2-hydroxypropyl)- β -cyclodextrin treatment. We suggest that the system reported by Zhai et al, could also benefit from this strategy. To provide other example of the strategy, in the revised article, we reported another system for arthritis treatment using a different type of cell membrane. The results demonstrate that this strategy is effective for this application as well. The related text is copied below for your convenience:

Macrophage membrane was reported to exhibit inflammatory site targeting capability⁴; thus, we constructed a macrophage membrane-wrapped PLGA nanosystem. We again found that the low-cholesterol RAW264.7 cell membrane-coated PLGA (e.g. CRM) possessed better inflammation site accumulation than particles coated with normal membrane (e.g. nCRM) and naked PLGA nanoparticles in arthritis model (Supplementary Fig. 20 a-d). These results imply that the cholesterol removal strategy is also applicable to other sources of biological membranes and other disease models.

Supplementary Figure 20. Targeting ability of CRM in arthritis model. **a** Representative image of ankle joints (Scale bar = 0.3 cm), H&E (Scale bar = 100 μ m) and Safranin-O and toluidine blue staining of ankle joints (Scale bar = 100 μ m). $n = 3$ independent animals. **b** Representative TEM image of nCRM and CRM. Scale bar = 50 nm. $n = 3$ independent samples. **c, d** Fluorescence images of model ankle joints at 6 h post injection of nCRM and CRM (**c**), and the fluorescence intensity of DiD in model ankle joints (**d**). Data represent mean \pm SEM, $n = 4$ independent animals. **e, f** Fluorescence images of plasma at 0.5 h post injection of particles (**e**), and the mean fluorescence intensity of DiD in the plasma (**f**). Data represent mean \pm SEM, $n = 3$ independent animals. **g, h** The uptake of nCRM and CRM by monocytes in the blood detected by flow-cytometry (**g**), and the relative fluorescence intensity of DiD in monocytes (**h**). Data represent mean \pm SEM, $n = 3$ independent animals. **i, j** Representative fluorescence images of major organs at 0.5 h post injection of nCRM and CRM (**i**), and the fluorescence intensity of DiD in the major organs (**j**). Data represent mean \pm SEM, $n = 3$ independent animals. nCRM, normal-Cholesterol cell membrane sourced from RAW264.7 cells coated PLGA nanoparticles. CRM, low-Cholesterol cell membrane sourced from RAW264.7 cells coated PLGA nanoparticles.

We also apologize for failing to meet your criteria for experimental detail like the immune-

related data, we have conducted new experiments (e.g., the macrophage membrane system above) and repeated many experiments for more detailed measurements and analysis. Hopefully, the revised version of our article is more satisfactory to you.

Finally, we would like to argue that our subtractive strategy has wide applicability and potential for clinical translation. We have tested the efficiency of CISP on colon cancer model without using PTT, and evaluated several other systems for different applications (e.g., other tumor types, different membranes, and arthritis) using the same cholesterol removal strategy. All experiments gave positive results. The key point is that this strategy is easily applicable to basically any biomimetic system that uses biological membrane as coating to improve their *in vivo* functionality. The biomimetic system developed for faster clinical translation is not necessary to integrate as many elements as the CISP system, which acted as a proof-of-concept here.

Major Points:

1. For all studies, show individual mice as data points on bar graphs and spider plots should be shown for tumor volume studies.

Response: Thanks for your suggestion. We have now revised the figures to show individual mice as data points on bar graphs and spider plots for tumor volume studies, and the revised figures are presented below:

Fig. 6 CISP enhanced anti-tumor immunity in mice bearing melanoma. **a** Schematic illustration showing the experiment process. Red arrows indicate administration of nanomedicines. The dosage of SR-717 is 10 mg/kg. **b** Western blot analysis of p-TBK1, TBK1, p-IRF3 and IRF3 in tumors at 6 days post laser irradiation. **c** Flow-cytometric analysis of CD44⁺PD-L1⁺ cells in B16F10 melanoma tumors at 6 days post laser irradiation. Data represent mean \pm SEM, $n = 4$ independent animals. **d** Flow-cytometric analysis of CD3⁺CD8⁺ cells in B16F10 melanoma tumors. Data represent mean \pm SEM, $n = 4$ independent animals. **e** Flow-cytometric analysis of CD69⁺ cells in CD8 T cells in B16F10 melanoma tumors. Data represent mean \pm SEM, $n = 4$ independent animals. **f** Flow-cytometric analysis of Granzyme B⁺ cells in CD8 T cells in B16F10 melanoma tumors. Data represent mean \pm SEM, $n = 4$ independent animals. **g** Flow-cytometric analysis of Ki67⁺ cells in CD8 T cells in B16F10 melanoma tumors. Data represent mean \pm SEM, $n = 4$ independent animals. **h** Flow-cytometric analysis of CD3⁺CD4⁺ cells in B16F10 melanoma tumors. Data represent mean \pm SEM, $n = 4$ independent

animals. **i** Flow-cytometric analysis of Foxp3⁺ cells in CD4 T cells in B16F10 melanoma tumors. Data represent mean \pm SEM, $n = 4$ independent animals. **j, k, l, m, n, o** IL-12 (**j**), L-6 (**k**), IFN γ (**l**), IFN β (**m**), IFN α (**n**) and TNF α (**o**) at 6 days post laser irradiation. Data represent mean \pm SEM, $n = 6$ independent animals. **p** Tumor growth curve of mice bearing melanoma. Red arrows indicate administration of nanomedicines. Data represent mean \pm SEM, $n = 8$ independent animals. **q** Survival curve of mice bearing melanoma, $n = 8$ independent animals. QFN, Quercetin-ferrum nanoparticles; QFN₇₁₇, QFN loaded with SR-717; nCSP, normal-Cholesterol cell membrane coated QFN₇₁₇; CSP, low-Cholesterol membrane coated QFN₇₁₇; nCISP, normal-Cholesterol cell membrane coated ICB agent and QFN₇₁₇; CISP, low-Cholesterol membrane coated ICB agent and QFN₇₁₇; CIP, low-Cholesterol membrane coated ICB agent and QFN.

Supplementary Figure 35. Tumor growth curve of mice bearing B16F10 melanoma tumors. $n = 8$ independent animals. L, laser; QFN, Quercetin-ferrum nanoparticles; QFN₇₁₇, QFN loaded with SR-717; nCSP, normal-Cholesterol cell membrane coated QFN₇₁₇; CSP, low-Cholesterol membrane coated QFN₇₁₇; nCISP, normal-Cholesterol cell membrane coated ICB agent and QFN₇₁₇; CISP, low-Cholesterol membrane coated ICB agent and QFN₇₁₇; CIP, low-Cholesterol membrane coated ICB agent and QFN.

Supplementary Figure 42. Tumor growth curve of mice bearing MC38 tumors. $n = 7$ independent animals. QFN, Quercetin-ferrum nanoparticles; QFN₇₁₇, QFN loaded with SR-717; nCSP, normal-Cholesterol cell membrane coated QFN₇₁₇; CSP, low-Cholesterol membrane coated QFN₇₁₇; nCISP, normal-Cholesterol cell membrane coated ICB agent and QFN₇₁₇; CISP, low-Cholesterol membrane coated ICB agent and QFN₇₁₇; CIP, low-Cholesterol membrane coated ICB agent and QFN.

Supplementary Figure 45. Tumor growth curve of mice bearing LLC tumors. $n = 8$ independent animals. QFN, Quercetin-ferrum nanoparticles; QFN₇₁₇, QFN loaded with SR-717; nCSP, normal-Cholesterol cell membrane coated QFN₇₁₇; CSP, low-Cholesterol membrane coated QFN₇₁₇; nCISP, normal-Cholesterol cell membrane coated ICB agent and QFN₇₁₇; CISP, low-Cholesterol membrane coated ICB agent and QFN₇₁₇; CIP, low-Cholesterol membrane coated ICB agent and QFN.

2. For studies in Figure 4C, the authors should quantify NP concentration using fluorescence spectroscopy (for both blood and plasma) instead of IVIS imaging, which is not sufficiently quantitative without use of standards. They should also calculate a half-life and other relevant pharmacokinetic parameters (e.g., AUC) from their data. The data in 4C doesn't suggest a major difference in half-life between nCISP and CISP and so conclusions regarding this should be strengthened.

Response: Thanks for your suggestion. The drug levels in serum were measured when determining pharmacokinetic parameters. Fluorescence spectrophotometer was used to detect DiD in serum, and LC-MS/MS was used to detect SR-717 in serum directly. Considering that blood cells (eg. red blood cells, platelet) may absorb the fluorescence emitted by DiD, which may affect the measurement accuracy of fluorescence spectrophotometer, we did not detect DiD in whole blood. Results are presented below:

Results show that the amount of CISP (marked with DiD) in plasma or serum was higher than that of nCISP (Fig. 4f, g and Supplementary Fig. 22a, b), with the Area Under Curve (AUC) of CISP being ~55% higher and the half life time ($t_{1/2}$) prolonged (Fig. 4h, i and Supplementary Fig. 23). The AUC of SR-717 in CISP group is also ~65% higher than that in nCISP group (Supplementary Fig. 24).

Fig. 4 The cholesterol-deficient cell membrane inhibited the clearance of CISP by monocytes in the blood. **h** Fluorescence images of serum in mice. Data represent mean \pm SEM, $n = 3$ independent animals. **i** Concentration of DiD in serum detected by fluorescence spectrophotometer. Data represent mean \pm SEM, $n = 3$ independent animals. MFI mean fluorescence intensity; QFN, Quercetin-ferrum nanoparticles; QFN₇₁₇, QFN loaded with SR-717; nCISP, normal-Cholesterol cell membrane coated ICB agent and QFN₇₁₇; CISP, low-Cholesterol membrane coated ICB agent and QFN₇₁₇.

Supplementary Figure 24. Concentration of SR-717 in serum. Mice with B16F10 melanoma tumor were treated with nCISP and CISP (10 mg/kg, iv.) and sacrificed at 0.25 h, 0.5 h, 2 h, 4 h, 8 h and 12 h. The concentration of SR-717 in serum was detected by LC-MS/MS. Data represent mean \pm SEM, $n = 5$ independent animals. QFN, Quercetin-ferrum nanoparticles; QFN₇₁₇, QFN loaded with SR-717; nCISP, normal-Cholesterol cell membrane coated ICB agent and QFN₇₁₇; CISP, low-Cholesterol membrane coated ICB agent and QFN₇₁₇.

3. There are several issues with the immunophenotyping performed in Figure 5.

a. Some metric of T cell function/activation/proliferation should be included in the analysis as total T cell infiltrate, particularly for CD4 T cells, isn't particularly

informative.

Response: Thanks for your suggestion. We examined the T cells according to your request again in a repeated experiment using mice bearing melanoma, and the results are presented below:

With the combined therapeutic effects, CISP significantly increased the number of CD8 T cells, CD69⁺ CD8 T cells, Granzyme B⁺ CD8 T cells, Ki67⁺ CD8 T cells and CD4 T cells in tumors (Fig. 6d-h and Supplementary Fig. 29b, c). The maturation of DCs, M1 macrophages, memory T cells in tumors, and activated T cells in tumor draining lymph nodes (TdLNs) were also significantly increased (Supplementary Fig. 30a-c, Supplementary Fig. 31a, Supplementary Fig. 32a-d and Supplementary Fig. 33a, b). Besides, Treg cells and M2 macrophages in tumors were decreased (Fig. 6i, Supplementary Fig. 31b), and the level of IL-12, IL-6, IFN γ , IFN β , IFN α and TNF α in tumors were elevated (Fig. 6j-o). In contrast, other groups only showed limited effect on activating anti-tumor immune response (Fig. 6d-o). It is worth noting that although CIP (denotes low-Cholesterol membrane coated ICB agent and Photothermal agent) could also reduce PD-L1 in tumors, the final outcome is unsatisfactory, indicating that SR-717 is necessary to boost the immune responses (Fig. 1j, Fig. 2f and Fig. 6c-o).

Fig. 6 CISP enhanced anti-tumor immunity in mice bearing melanoma. a Schematic illustration showing the experiment process. Red arrows indicate administration of nanomedicines. The dosage of SR-717 is 10 mg/kg. **b** Western blot analysis of p-TBK1, TBK1, p-IRF3 and IRF3 in tumors at 6 days post laser irradiation. **c** Flow-cytometric analysis of CD44⁺PD-L1⁺ cells in B16F10 melanoma tumors at 6 days post laser irradiation. Data represent mean ± SEM, *n* = 4 independent animals. **d** Flow-cytometric analysis of CD3⁺CD8⁺ cells in B16F10 melanoma tumors. Data represent mean ± SEM, *n* = 4 independent animals. **e** Flow-cytometric analysis of CD69⁺ cells in CD8 T cells in B16F10 melanoma tumors. Data represent mean ± SEM, *n* = 4 independent animals. **f** Flow-cytometric analysis of Granzyme B⁺ cells in CD8 T cells in B16F10 melanoma tumors. Data represent mean ± SEM, *n* = 4 independent animals. **g** Flow-cytometric analysis of Ki67⁺ cells in CD8 T cells in B16F10 melanoma tumors. Data represent mean ± SEM, *n* = 4 independent animals. **h** Flow-cytometric analysis of CD3⁺CD4⁺ cells in B16F10 melanoma tumors. Data represent mean ± SEM, *n* = 4 independent

animals. **i** Flow-cytometric analysis of Foxp3⁺ cells in CD4 T cells in B16F10 melanoma tumors. Data represent mean \pm SEM, $n = 4$ independent animals. **j, k, l, m, n, o** IL-12 (**j**), L-6 (**k**), IFN γ (**l**), IFN β (**m**), IFN α (**n**) and TNF α (**o**) at 6 days post laser irradiation. Data represent mean \pm SEM, $n = 6$ independent animals. **p** Tumor growth curve of mice bearing melanoma. Red arrows indicate administration of nanomedicines. Data represent mean \pm SEM, $n = 8$ independent animals. **q** Survival curve of mice bearing melanoma, $n = 8$ independent animals. QFN, Quercetin-ferrum nanoparticles; QFN₇₁₇, QFN loaded with SR-717; nCSP, normal-Cholesterol cell membrane coated QFN₇₁₇; CSP, low-Cholesterol membrane coated QFN₇₁₇; nCISP, normal-Cholesterol cell membrane coated ICB agent and QFN₇₁₇; CISP, low-Cholesterol membrane coated ICB agent and QFN₇₁₇; CIP, low-Cholesterol membrane coated ICB agent and QFN.

Supplementary Figure 32. Memory T cells in B16F10 melanoma tumors. a, b, c, d Memory T cells including CD8⁺Tcm (**a**), CD8⁺Tem (**b**), CD4⁺Tcm (**c**) and CD4⁺Tem (**d**) in tumors were examined by flow cytometer. Tumors were harvested at 6 days post laser irradiation. Data represent mean \pm SEM, $n = 4$ independent animals. L, laser; QFN, Quercetin-ferrum nanoparticles; QFN₇₁₇, QFN loaded with SR-717; nCSP, normal-Cholesterol cell membrane coated QFN₇₁₇; CSP, low-Cholesterol membrane coated QFN₇₁₇; nCISP, normal-Cholesterol cell membrane coated ICB agent and QFN₇₁₇; CISP, low-Cholesterol membrane coated ICB agent and QFN₇₁₇; CIP, low-Cholesterol membrane coated ICB agent and QFN.

Supplementary Figure 33. Activated T cells in TdLNs in mice bearing B16F10 melanoma tumor. a Flow-cytometric analysis of CD69⁺ cells in CD3⁺CD8⁺ cells in TdLNs at 6 days post laser irradiation. Data represent mean \pm SEM, $n = 4$ independent animals. **b** Flow-cytometric analysis of CD69⁺ cells in CD3⁺CD4⁺ cells in TdLNs at 6 days post laser irradiation. Data represent mean \pm SEM, $n = 4$ independent animals. L, laser, QFN, Quercetin-ferrum nanoparticles; QFN₇₁₇, QFN loaded with SR-717; nCSP, normal-Cholesterol cell membrane coated QFN₇₁₇; CSP, low-Cholesterol membrane coated QFN₇₁₇; nCISP, normal-Cholesterol cell membrane coated ICB agent and QFN₇₁₇; CISP, low-Cholesterol membrane coated ICB agent and QFN₇₁₇; CIP, low-Cholesterol membrane coated ICB agent and QFN; TdLNs, Tumor draining lymph nodes.

b. In 4H, are these +F4/80+CD80+ cells? Was CD11b+ included in the panel as well? Was CD86 also looked at – this would be a more accepted marker of M1-like macrophages.

Response: Thanks for your suggestion. We assessed the M1 markers (CD11b⁺CD86⁺CD80⁺)⁵ in the repeated experiment. These data were presented in Sup. Fig. 30 and 37, which are copied below for your convenience:

Supplementary Figure 31. Macrophages in B16F10 melanoma tumors. a Flow-cytometric analysis of CD80⁺CD86⁺ cells in CD11b⁺F4/80⁺ cells (M1) in B16F10 melanoma tumor at 6 days post laser irradiation. Data represent mean \pm SEM, $n = 4$ independent animals. **b** Flow-cytometric analysis of CD206⁺F4/80⁺ cells in CD11b⁺ cells (M2) in B16F10 melanoma tumor

at 6 days post laser irradiation. Data represent mean \pm SEM, $n = 4$ independent animals. L, laser; QFN, Quercetin-ferrum nanoparticles; QFN₇₁₇, QFN loaded with SR-717; nCSP, normal-Cholesterol cell membrane coated QFN₇₁₇; CSP, low-Cholesterol membrane coated QFN₇₁₇; nCISP, normal-Cholesterol cell membrane coated ICB agent and QFN₇₁₇; CISP, low-Cholesterol membrane coated ICB agent and QFN₇₁₇; CIP, low-Cholesterol membrane coated ICB agent and QFN.

Supplementary Figure 37. Macrophages, NK cells and DCs in MC38 tumors. **a** Flow-cytometric analysis of CD80⁺CD86⁺ cells in CD11b⁺F4/80⁺ cells in MC38 tumors at 18 days post tumor inoculation. Data represent mean \pm SEM, $n = 4$ independent animals. **b** Flow-cytometric analysis of NK1.1⁺ cells in in MC38 tumors at 18 days post tumor inoculation. Data represent mean \pm SEM, $n = 4$ independent animals. **c** Flow-cytometric analysis of CD11c⁺CD86⁺ cells in MC38 tumors at 18 days post tumor inoculation. Data represent mean \pm SEM, $n = 4$ independent animals. **d** Representative flow cytometry dot plot of CD80⁺CD86⁺ cells in CD11b⁺F4/80⁺ cells in MC38 tumors. $n = 4$ independent animals. **e** Representative flow cytometry dot plot of NK1.1⁺ cells in MC38 tumors. $n = 4$ independent animals. **f**

Representative flow cytometry dot plot of CD11c⁺CD86⁺ in MC38 tumors. *n* = 4 independent animals. QFN, Quercetin-ferrum nanoparticles; QFN₇₁₇, QFN loaded with SR-717; nCSP, normal-Cholesterol cell membrane coated QFN₇₁₇; CSP, low-Cholesterol membrane coated QFN₇₁₇; nCISP, normal-Cholesterol cell membrane coated ICB agent and QFN₇₁₇; CISP, low-Cholesterol membrane coated ICB agent and QFN₇₁₇; CIP, low-Cholesterol membrane coated ICB agent and QFN.

c. Data in 4J isn't very convincing; quantification and additional assay to quantify INF-gamma levels is needed.

Response: Thanks for pointing out this flaw. The level of IFN- γ in tumors was quantified by ELISA and Figure 6j-o was revised to show the data, the figure is copied at response to comment 3a.

d. The flow data shown in Figure S16 does not appear to be properly compensated; there should be clearly resolved populations of CD4 and CD8 T cells. Plot CD4 vs. CD8 to ensure clearly resolved populations.

Response: Thanks for your suggestion. These data were presented in Sup. Fig. 40 in the revised manuscript, which are copied below for your convenience:

CISP caused significant up-regulation of the CD8⁺/CD4⁺ ratio in TdLNs, which was reported to associated with the prognosis of a variety of cancers and the response to immunotherapy² (Supplementary Fig. 40a, b).

Supplementary Figure 40. Proportions of CD8⁺ and CD4⁺ T cells in TdLNs in mice bearing MC38 tumor. a, b Representative flow cytometry dot plot gated on CD3⁺ T cells (a)

and the proportions of CD8⁺/CD4⁺ (b). Data represent mean ± SEM, *n* = 4 independent animals. QFN, Quercetin-ferrum nanoparticles; QFN₇₁₇, QFN loaded with SR-717; nCSP, normal-Cholesterol cell membrane coated QFN₇₁₇; CSP, low-Cholesterol membrane coated QFN₇₁₇; nCISP, normal-Cholesterol cell membrane coated ICB agent and QFN₇₁₇; CISP, low-Cholesterol membrane coated ICB agent and QFN₇₁₇; CIP, low-Cholesterol membrane coated ICB agent and QFN.

4. The therapeutic data in Figure K-L is encouraging but there are several issues to address:

a. Statistical comparisons need to be included for tumor growth and survival.

Response: Thanks for your suggestion. We have performed the requested analysis. Related results are presented below for your convenience:

Ultimately, ~85% of the mice in CISP group were cured with no tumor recurrence on day 80, which is the best outcome (Fig. 6p, q, Supplementary Fig. 34, Supplementary Fig. 35). In contrast, nCSP, CSP and nCISP displayed weaker therapeutic efficiency, with a survival rate of ~20%; and no mice survived with free SR-717 treatment, reflecting the importance of the delivery system. Noticeably, the tumor recurrence rate in CISP group was lower than CIP, again illustrated the important role of SR-717 in this system. (Due to the varying recurrence times of mouse bearing B16F10 melanoma tumor, the statistical comparisons for tumor growth are not did in this model).

Fig. 6 CISP enhanced anti-tumor immunity in mice bearing melanoma. p Tumor growth curve of mice bearing melanoma. Red arrows indicate administration of nanomedicines. Data represent mean ± SEM, *n* = 8 independent animals. **q** Survival curve of mice bearing melanoma, *n* = 8 independent animals. QFN, Quercetin-ferrum nanoparticles; QFN₇₁₇, QFN loaded with SR-717; nCSP, normal-Cholesterol cell membrane coated QFN₇₁₇; CSP, low-Cholesterol membrane coated QFN₇₁₇; nCISP, normal-Cholesterol cell membrane coated ICB agent and QFN₇₁₇; CISP, low-Cholesterol membrane coated ICB agent and QFN₇₁₇; CIP, low-Cholesterol membrane coated ICB agent and QFN.

Ultimately, CISP caused the strongest MC38 tumor suppression, and ~40% of treated mice survived on day 80, while all mice in other groups died (Fig. 7k-m and Supplementary Fig. 42). Thus, CISP is also significantly more effective than other systems in treating MC38 colon cancer, even without the introduction of PTT.

Fig. 7 CISP enhanced anti-tumor immunity in mice bearing MC38 tumor. l Tumor growth curve of mice bearing MC38 tumors. Red arrows indicate administration of nanomedicines. Data represent mean \pm SEM, $n = 7$ independent animals. **m** Survival curve of mice bearing MC38 tumors. $n = 7$ independent animals. QFN, Quercetin-ferrum nanoparticles; QFN₇₁₇, QFN loaded with SR-717; nCSP, normal-Cholesterol cell membrane coated QFN₇₁₇; CSP, low-Cholesterol membrane coated QFN₇₁₇; nCISP, normal-Cholesterol cell membrane coated ICB agent and QFN₇₁₇; CISP, low-Cholesterol membrane coated ICB agent and QFN₇₁₇; CIP, low-Cholesterol membrane coated ICB agent and QFN.

Ultimately, only CISP inhibited the growth of LLC tumors, while other groups basically failed to show therapeutic effect in this tumor model (Supplementary Fig. 43m, n, Supplementary Fig. 44 and Supplementary Fig. 45).

Supplementary Figure 43. The therapeutic effects of CISP in mice bearing LLC tumor. m Tumor growth curve of mice bearing LLC tumors. Red arrows indicate administration of nanomedicines. Data represent mean \pm SEM, $n = 8$ independent animals. **n** Survival curve of mice bearing LLC tumors. $n = 8$ independent animals. QFN, Quercetin-ferrum nanoparticles; QFN₇₁₇, QFN loaded with SR-717; nCSP, normal-Cholesterol cell membrane coated QFN₇₁₇; CSP, low-Cholesterol membrane coated QFN₇₁₇; nCISP, normal-Cholesterol cell membrane coated ICB agent and QFN₇₁₇; CISP, low-Cholesterol membrane coated ICB agent and QFN₇₁₇; CIP, low-Cholesterol membrane coated ICB agent and QFN.

b. It is surprising that SR717 alone didn't have an effect considering that the original

report that describes this molecule tested this agent in the B16 model. That group used SR717 at a dose of 30 mg/kg whereas it is reduced here to 10 mg/kg; thus, it is not clear that the carrier platform is enhancing the efficacy of the drug. This would be a remarkably complicated design if the same effect could be achieved by increasing the free drug dose 2-3x.

Response: Thanks very much for your comment. Indeed, in the original report, SR-717 was administrated daily (30 mg/kg, I.P.). However, at this dosage, SR-717 treatment upregulated alanine aminotransferase (ALT) levels on day 4 after treatment and displayed certain level of adverse effects⁶. Therefore, due to safety considerations and the expensive price of SR-717, we reduced the dosage of SR-717 to 10 mg/kg (iv.). Our data show that CISP could enhance its tumor accumulation to activate STING signals like free SR-717 at 30 mg/kg; while both ALT and AST were all in the normal range after treatment (Supplementary Figure 50a-f, Supplementary Figure 51a-f). According to these results, we suggest that the CISP delivery system may reduce the dosage of SR-717 needed while enhancing its effect on tumors, implying that the delivery system worked as proposed. The related results are copied below:

CISP could significantly increase the level of SR-717 in tumors, and CSP delivered less SR-717 to tumors (Fig. 3e).

Fig. 3 CISP targeted to tumors in vivo. e Concentration of SR-717 in tumors detected by LC-MS/MS. Data represent mean \pm SEM, $n = 4$ independent animals.

Western blot results demonstrated that CISP significantly increased the expression of p-TBK1 and p-IRF3 in tumors, indicating that STING signal pathway in tumor sites was activated (Fig. 6b). In comparison, free SR-717, QFN₇₁₇, nCSP, and nCISP all displayed limited effect (Fig. 6b).

Fig. 6 CISP enhanced anti-tumor immunity in mice bearing melanoma. b Western blot analysis of p-TBK1, TBK1, p-IRF3 and IRF3 in tumors at 6 days post laser irradiation.

Blood routine examination and blood biochemistry (including ALT, AST, LDH, CREA, UREA and CHE) of mice bearing MC38 tumor was all in the normal range treated with CISP (Supplementary Fig. 49a-f and Supplementary Fig. 50a-f). Blood biochemistry of mice bearing LLC tumor was also in the normal range treated with CISP (Supplementary Fig. 51a-f).

Supplementary Figure 50. Blood biochemistry of mice bearing MC38 tumor at 18 days post tumor inoculation. a The serum level of ALT in mice receiving the indicated treatment. Data represent mean \pm SEM, $n = 6$ independent animals. **b** The serum level of AST in mice receiving the indicated treatment. Data represent mean \pm SEM, $n = 6$ independent animals. **c** The serum level of LDH in mice receiving the indicated treatment. Data represent mean \pm SEM, $n = 6$ independent animals. **d** The serum level of CHE in mice receiving the indicated treatment. Data represent mean \pm SEM, $n = 6$ independent animals. **e** The serum level of UREA in mice

receiving the indicated treatment. Data represent mean \pm SEM, $n = 6$ independent animals. **f** The serum level of CREA in mice receiving the indicated treatment. Data represent mean \pm SEM, $n = 6$ independent animals. ALT, alanine aminotransferase; AST, aspartate aminotransferase; LDH, lactic dehydrogenase; CHE, Cholinesterase; CREA, Creatine. QFN, Quercetin-ferrum nanoparticles; QFN₇₁₇, QFN loaded with SR-717; nCSP, normal-Cholesterol cell membrane coated QFN₇₁₇; CSP, low-Cholesterol membrane coated QFN₇₁₇; nCISP, normal-Cholesterol cell membrane coated ICB agent and QFN₇₁₇; CISP, low-Cholesterol membrane coated ICB agent and QFN₇₁₇; CIP, low-Cholesterol membrane coated ICB agent and QFN.

Supplementary Figure 51. Blood biochemistry of mice bearing LLC tumor at 18 days post tumor inoculation. **a** The serum level of ALT in mice receiving the indicated treatment. Data represent mean \pm SEM, $n = 5$ independent animals. **b** The serum level of AST in mice receiving the indicated treatment. Data represent mean \pm SEM, $n = 5$ independent animals. **c** The serum level of LDH in mice receiving the indicated treatment. Data represent mean \pm SEM, $n = 5$ independent animals. **d** The serum level of CHE in mice receiving the indicated treatment. Data represent mean \pm SEM, $n = 5$ independent animals. **e** The serum level of UREA in mice receiving the indicated treatment. Data represent mean \pm SEM, $n = 5$ independent animals. **f** The serum level of CREA in mice receiving the indicated treatment. Data represent mean \pm SEM, $n = 5$ independent animals. ALT, alanine aminotransferase; AST, aspartate aminotransferase; LDH, lactic dehydrogenase; CHE, Cholinesterase; CREA, Creatine. QFN, Quercetin-ferrum nanoparticles; QFN₇₁₇, QFN loaded with SR-717; nCSP, normal-Cholesterol cell membrane coated QFN₇₁₇; CSP, low-Cholesterol membrane coated QFN₇₁₇; nCISP, normal-Cholesterol cell membrane coated ICB agent and QFN₇₁₇; CISP, low-Cholesterol membrane coated ICB agent and QFN₇₁₇; CIP, low-Cholesterol membrane coated ICB agent and QFN.

c. It would be ideal repeat this study in at least one more tumor model.

Response: Thank you for the advice, we agree that more tumor model and maybe other disease model could provide more support for our claim. According to your suggestion, we tested CISP in 4 different tumor models (melanoma, MC38 tumor, LLC tumor and metastatic melanoma), all with positive final outcomes. Besides, an arthritis model was also tested using the same subtractive membrane engineering strategy but with different membrane. This inflammatory disease model also gave positive results. The figures of these *in vivo* tests are copied below for your convenience:

Fig. 6 CISP enhanced anti-tumor immunity in mice bearing melanoma. a Schematic illustration showing the experiment process. Red arrows indicate administration of nanomedicines. The dosage of SR-717 is 10 mg/kg. **b** Western blot analysis of p-TBK1, TBK1,

p-IRF3 and IRF3 in tumors at 6 days post laser irradiation. **c** Flow-cytometric analysis of CD44⁺PD-L1⁺ cells in B16F10 melanoma tumors at 6 days post laser irradiation. Data represent mean \pm SEM, $n = 4$ independent animals. **d** Flow-cytometric analysis of CD3⁺CD8⁺ cells in B16F10 melanoma tumors. Data represent mean \pm SEM, $n = 4$ independent animals. **e** Flow-cytometric analysis of CD69⁺ cells in CD8 T cells in B16F10 melanoma tumors. Data represent mean \pm SEM, $n = 4$ independent animals. **f** Flow-cytometric analysis of Granzyme B⁺ cells in CD8 T cells in B16F10 melanoma tumors. Data represent mean \pm SEM, $n = 4$ independent animals. **g** Flow-cytometric analysis of Ki67⁺ cells in CD8 T cells in B16F10 melanoma tumors. Data represent mean \pm SEM, $n = 4$ independent animals. **h** Flow-cytometric analysis of CD3⁺CD4⁺ cells in B16F10 melanoma tumors. Data represent mean \pm SEM, $n = 4$ independent animals. **i** Flow-cytometric analysis of Foxp3⁺ cells in CD4 T cells in B16F10 melanoma tumors. Data represent mean \pm SEM, $n = 4$ independent animals. **j, k, l, m, n, o** IL-12 (**j**), L-6 (**k**), IFN γ (**l**), IFN β (**m**), IFN α (**n**) and TNF α (**o**) at 6 days post laser irradiation. Data represent mean \pm SEM, $n = 6$ independent animals. **p** Tumor growth curve of mice bearing melanoma. Red arrows indicate administration of nanomedicines. Data represent mean \pm SEM, $n = 8$ independent animals. **q** Survival curve of mice bearing melanoma, $n = 8$ independent animals. QFN, Quercetin-ferrum nanoparticles; QFN₇₁₇, QFN loaded with SR-717; nCSP, normal-Cholesterol cell membrane coated QFN₇₁₇; CSP, low-Cholesterol membrane coated QFN₇₁₇; nCISP, normal-Cholesterol cell membrane coated ICB agent and QFN₇₁₇; CISP, low-Cholesterol membrane coated ICB agent and QFN₇₁₇; CIP, low-Cholesterol membrane coated ICB agent and QFN.

Fig. 7 CISP enhanced anti-tumor immunity in mice bearing MC38 tumor. **a** Schematic illustration showing the experiment process. Red arrows indicate administration of nanomedicines. The dosage of SR-717 is 10 mg/kg. **b, c** Fluorescence images of MC38 tumors at 6 h post treated with of nCSP, CSP, nCISP and CISP (**b**) and mean fluorescence intensity (MFI) of DiD in tumors (**c**). Data represent mean \pm SEM, $n = 3$ independent animals. **d** Flow-cytometric analysis of CD44⁺PD-L1⁺ cells in MC38 tumors on day 18. Data represent mean \pm SEM, $n = 4$ independent animals. **e** Flow-cytometric analysis of CD3⁺CD8⁺ cells in MC38 tumors. Data represent mean \pm SEM, $n = 4$ independent animals. **f** Flow-cytometric analysis of CD69⁺ cells in CD8 T cells in MC38 tumors. Data represent mean \pm SEM, $n = 4$ independent animals. **g** Flow-cytometric analysis of Granzyme B⁺ cells in CD8 T cells in MC38 tumors. Data represent mean \pm SEM, $n = 4$ independent animals. **h** Flow-cytometric analysis of Ki67⁺ cells in CD8 T cells in MC38 tumors. Data represent mean \pm SEM, $n = 4$ independent animals. **i** Flow-cytometric analysis of CD3⁺CD4⁺ cells in MC38 tumors. Data represent mean \pm SEM, $n = 4$ independent animals. **j** Flow-cytometric analysis of FcγR3⁺ cells in CD4 T cells in MC38 tumors. Data represent mean \pm SEM, $n = 4$ independent animals. **k** Images of MC38 tumors at

14 days and 24 days post tumor inoculation. $n = 7$ independent animals. Scale bar = 0.5 cm. **l** Tumor growth curve of mice bearing MC38 tumors. Red arrows indicate administration of nanomedicines. Data represent mean \pm SEM, $n = 7$ independent animals. **m** Survival curve of mice bearing MC38 tumors. $n = 7$ independent animals. QFN, Quercetin-ferrum nanoparticles; QFN₇₁₇, QFN loaded with SR-717; nCSP, normal-Cholesterol cell membrane coated QFN₇₁₇; CSP, low-Cholesterol membrane coated QFN₇₁₇; nCISP, normal-Cholesterol cell membrane coated ICB agent and QFN₇₁₇; CISP, low-Cholesterol membrane coated ICB agent and QFN₇₁₇; CIP, low-Cholesterol membrane coated ICB agent and QFN.

Supplementary Figure 43. The therapeutic effects of CISP in mice bearing LLC tumor. a, b Fluorescence images of LLC tumors at 6 h post injection of nCISP and CISP (**a**), and the fluorescence intensity of DiD in tumors (**b**). Data represent mean \pm SEM, $n = 3$ independent animals. **c** Schematic illustration showing the experiment process. Red arrows indicate administration of nanomedicines. The dosage of SR-717 is 10 mg/kg. **d** Flow-cytometric analysis of CD44⁺PD-L1⁺ cells in LLC tumors at 18 days post tumor inoculation. Data represent mean \pm SEM, $n = 4$ independent animals. **e** Flow-cytometric analysis of CD3⁺CD8⁺ cells in LLC tumors at 18 days post tumor inoculation. Data represent mean \pm SEM, $n = 4$ independent animals. **f** Flow-cytometric analysis of NK1.1⁺ cells in LLC tumors at 18 days post tumor

inoculation. Data represent mean \pm SEM, $n = 4$ independent animals. **g** Flow-cytometric analysis of CD80⁺CD86⁺ cells in F4/80⁺CD11b⁺ cells in LLC tumors at 18 days post tumor inoculation. Data represent mean \pm SEM, $n = 4$ independent animals. **h, i, j, k, l** IFN α (**h**), IFN β (**i**), TNF α (**j**), IFN γ (**k**) and IL6 (**l**) in LLC tumors at 18 days post tumor inoculation. Data represent mean \pm SEM, $n = 5$ independent animals. **m** Tumor growth curve of mice bearing LLC tumors. Red arrows indicate administration of nanomedicines. Data represent mean \pm SEM, $n = 8$ independent animals. **n** Survival curve of mice bearing LLC tumors. $n = 8$ independent animals. QFN, Quercetin-ferrum nanoparticles; QFN₇₁₇, QFN loaded with SR-717; nCSP, normal-Cholesterol cell membrane coated QFN₇₁₇; CSP, low-Cholesterol membrane coated QFN₇₁₇; nCISP, normal-Cholesterol cell membrane coated ICB agent and QFN₇₁₇; CISP, low-Cholesterol membrane coated ICB agent and QFN₇₁₇; CIP, low-Cholesterol membrane coated ICB agent and QFN.

Supplementary Figure 47. The therapeutic effects of CISP in mice bearing metastatic melanoma. a Schematic illustration of the experiment process. The dosage of SR-717 is 10

mg/kg. **b** Representative images of lung and the number of pulmonary metastasis foci at 18 days post B16F10 injection. Metastasis foci is pointed by arrow. Scale bar = 0.3 cm. Data represent mean \pm SEM, $n = 5$ independent animals. **c** Representative images of H&E staining of lung sections in mice. Metastasis foci is pointed by arrow. $n = 5$ independent animals. **d, e** Flow-cytometric analysis of CD3⁺CD8⁺ cell in lung at **18 days post B16F10 injection**. Data represent mean \pm SEM, $n = 4$ independent animals. **f** Flow-cytometric analysis of CD11c⁺CD40⁺ cells in lung at 18 days post B16F10 injection. Data represent mean \pm SEM, $n = 4$ independent animals. **g** Flow-cytometric analysis of CD80⁺CD86⁺ cells in F4/80⁺CD11b⁺ cells in lung at 18 days post B16F10 injection. Data represent mean \pm SEM, $n = 4$ independent animals. **h** IFN α in lung at 18 days post B16F10 injection (iv.). Data represent mean \pm SEM, $n = 5$ independent animals. QFN, Quercetin-ferrum nanoparticles; QFN₇₁₇, QFN loaded with SR-717; nCSP, normal-Cholesterol cell membrane coated QFN₇₁₇; CSP, low-Cholesterol membrane coated QFN₇₁₇; nCISP, normal-Cholesterol cell membrane coated ICB agent and QFN₇₁₇; CISP, low-Cholesterol membrane coated ICB agent and QFN₇₁₇; CIP, low-Cholesterol membrane coated ICB agent and QFN.

Supplementary Figure 20. Targeting ability of CRM in arthritis model. a Representative image of ankle joints (Scale bar = 0.3 cm), H&E (Scale bar = 100 μ m) and Safranin-O and toluidine blue staining of ankle joints (Scale bar = 100 μ m). $n = 3$ independent animals. **b**

Representative TEM image of nCRM and CRM. Scale bar = 50 nm. $n = 3$ independent samples. **c, d** Fluorescence images of model ankle joints at 6 h post injection of nCCM and CCM (**c**), and the fluorescence intensity of DiD in model ankle joints (**d**). Data represent mean \pm SEM, $n = 4$ independent animals. **e, f** Fluorescence images of plasma at 0.5 h post injection of particles (**e**), and the mean fluorescence intensity of DiD in the plasma (**f**). Data represent mean \pm SEM, $n = 3$ independent animals. **g, h** The uptake of nCRM and CRM by monocytes in the blood detected by flow-cytometry (**g**), and the relative fluorescence intensity of DiD in monocytes (**h**). Data represent mean \pm SEM, $n = 3$ independent animals. **i, j** Representative fluorescence images of major organs at 0.5 h post injection of nCRM and CRM (**i**), and the fluorescence intensity of DiD in the major organs (**j**). Data represent mean \pm SEM, $n = 3$ independent animals. nCRM, normal-Cholesterol cell membrane sourced from RAW264.7 cells coated PLGA nanoparticles. CRM, low-Cholesterol cell membrane sourced from RAW264.7 cells coated PLGA nanoparticles.

5. Related to above, several important controls are missing from the study in Figure 5. First, does the SR717 need to be loaded with the PTT agents? To assess this, free SR717 + 'empty' CISP+L would need to be tested. Second, it is not clear what role the PTT plays here as the CISP system is not tested w/o PTT; the impact of the work would be much higher if PTT was not needed. The current study lacks sufficient controls to understand the contribution of all of the various components of the system and their role in mediating antitumor effects.

Response: Thank you very much for your suggestion, we apologize for not providing sufficient data. First of all, it can be seen that CISP could enhance the tumor accumulation of SR-717, which is an important function of the delivery system (Fig. 3e). According to your comment, CIP (denotes low-Cholesterol membrane coated ICB agent and Photothermal agent) + free SR-717 group was investigated using LLC tumor model (Supplementary Fig. 43) without laser irradiation. Related results are copied below for your convenience:

CISP could significantly increase the level of SR-717 in tumors, and CSP delivered less SR-717 to tumors (Fig. 3e).

Fig. 3 CISP targeted to tumors in vivo. e Concentration of SR-717 in tumors detected by LC-MS/MS. Data represent mean \pm SEM, $n = 4$ independent animals.

Additionally, we also detected the effect of CISP in mice bearing Lewis lung cancer. Results show that CISP could also target to LLC tumors (Supplementary Fig. 43a, b). To examine whether loading SR-717 into CISP lead to better outcome, CIP administrated with free SR-717 (CIP + SR-717) was used as a comparison in this tumor model (Supplementary Fig. 43c). Measurements show that CIP + SR-717 along with CISP and CIP significantly reduced the expression of PD-L1 in LLC tumor cells while nCSP, CSP and nCISP showed weaker effects (Supplementary Fig.43 d). However, only CISP resulted in the strongest increase of CD8 T cells, NK cells, M1 macrophage and anti-tumor cytokines including IL-6, IFN γ , IFN β , TNF α and IFN α in LLC tumors (Supplementary Fig. 43e-l). Ultimately, only CISP inhibited the growth of LLC tumors, while other groups basically failed to show therapeutic effect in this tumor model (Supplementary Fig. 43m, n, Supplementary Fig. 44 and Supplementary Fig. 45). These results indicate that CISP exhibited good therapeutic outcome in LLC tumors, and support the necessary of loading SR-717 into CISP to activate anti-tumor immune responses (Fig. 3e).

Supplementary Figure 43. The therapeutic effects of CISP in mice bearing LLC tumor. a, b Fluorescence images of LLC tumors at 6 h post injection of nCISP and CISP (a), and the fluorescence intensity of DiD in tumors (b). Data represent mean \pm SEM, $n = 3$ independent animals. **c** Schematic illustration showing the experiment process. Red arrows indicate administration of nanomedicines. The dosage of SR-717 is 10 mg/kg. **d** Flow-cytometric analysis of CD44⁺PD-L1⁺ cells in LLC tumors at 18 days post tumor inoculation. Data represent mean \pm SEM, $n = 4$ independent animals. **e** Flow-cytometric analysis of CD3⁺CD8⁺ cells in LLC tumors at 18 days post tumor inoculation. Data represent mean \pm SEM, $n = 4$ independent animals. **f** Flow-cytometric analysis of NK1.1⁺ cells in LLC tumors at 18 days post tumor inoculation. Data represent mean \pm SEM, $n = 4$ independent animals. **g** Flow-cytometric analysis of CD80⁺CD86⁺ cells in F4/80⁺CD11b⁺ cells in LLC tumors at 18 days post tumor inoculation. Data represent mean \pm SEM, $n = 4$ independent animals. **h, i, j, k, l** IFN α (h), IFN β (i), TNF α (j), IFN γ (k) and IL6 (l) in LLC tumors at 18 days post tumor inoculation. Data represent mean \pm SEM, $n = 5$ independent animals. **m** Tumor growth curve of mice bearing LLC tumors. Red arrows indicate administration of nanomedicines. Data represent mean \pm SEM, $n = 8$ independent animals. **n** Survival curve of mice bearing LLC tumors. $n = 8$ independent animals. QFN, Quercetin-ferrum nanoparticles; QFN₇₁₇, QFN loaded with SR-717; nCSP, normal-Cholesterol cell membrane coated QFN₇₁₇; CSP, low-Cholesterol membrane

coated QFN₇₁₇; nCISP, normal-Cholesterol cell membrane coated ICB agent and QFN₇₁₇; CISP, low-Cholesterol membrane coated ICB agent and QFN₇₁₇; CIP, low-Cholesterol membrane coated ICB agent and QFN.

6. Even with the best particle, most of the particle goes to the liver and the lung (Fig. S7, S11). Is STING activated in these tissues?

Response: Thanks for your comment, off-target effect of nanomedicine is indeed undesirable. In the responses to comment 4b and 5, we provided data to show that CISP can improve the tumor targeting of SR-717 and reduce the necessary administrate dosage from 30mg/kg to 10 mg/kg (i.e., a 66% drop). The blood indexes remained in normal range after treatment of CISP with overall dosage reduction of SR-717. Though SR-717 is present in liver, lung and kidney as measured by LC-MS (Supplementary Fig. 14), no identifiable hepatotoxicity sign was detected (Supplementary Fig. 50a-f, Supplementary Fig 51a-f, Supplementary Fig 52, Supplementary Fig 53).

Supplementary Figure 14. Concentration of SR-717 in the major organs. Mice with B16F10 melanoma tumor were treated with nCISP and CISP (10 mg/kg, iv.), and sacrificed at 8 h post injection. The concentration of SR-717 in major organs was detected by LC-MS/MS. Data represent mean \pm SEM, $n = 4$ independent animals. QFN, Quercetin-ferrum nanoparticles; QFN₇₁₇, QFN loaded with SR-717; nCISP, normal-Cholesterol cell membrane coated ICB agent and QFN₇₁₇; CISP, low-Cholesterol membrane coated ICB agent and QFN₇₁₇.

Blood routine examination and blood biochemistry (including ALT, AST, LDH, CREA, UREA and CHE) of mice bearing MC38 tumor was all in the normal range treated with CISP (Supplementary Fig. 49a-f and Supplementary Fig. 50a-f). Blood biochemistry of mice bearing LLC tumor was also in the normal range treated with CISP (Supplementary Fig. 51a-f).

Supplementary Figure 50. Blood biochemistry of mice bearing MC38 tumor at 18 days post tumor inoculation. **a** The serum level of ALT in mice receiving the indicated treatment. Data represent mean \pm SEM, $n = 6$ independent animals. **b** The serum level of AST in mice receiving the indicated treatment. Data represent mean \pm SEM, $n = 6$ independent animals. **c** The serum level of LDH in mice receiving the indicated treatment. Data represent mean \pm SEM, $n = 6$ independent animals. **d** The serum level of CHE in mice receiving the indicated treatment. Data represent mean \pm SEM, $n = 6$ independent animals. **e** The serum level of UREA in mice receiving the indicated treatment. Data represent mean \pm SEM, $n = 6$ independent animals. **f** The serum level of CREA in mice receiving the indicated treatment. Data represent mean \pm SEM, $n = 6$ independent animals. ALT, alanine aminotransferase; AST, aspartate aminotransferase; LDH, lactic dehydrogenase; CHE, Cholinesterase; CREA, Creatine. QFN, Quercetin-ferrum nanoparticles; QFN₇₁₇, QFN loaded with SR-717; nCSP, normal-Cholesterol cell membrane coated QFN₇₁₇; CSP, low-Cholesterol membrane coated QFN₇₁₇; nCISP, normal-Cholesterol cell membrane coated ICB agent and QFN₇₁₇; CISP, low-Cholesterol membrane coated ICB agent and QFN₇₁₇; CIP, low-Cholesterol membrane coated ICB agent and QFN.

Supplementary Figure 51. Blood biochemistry of mice bearing LLC tumor at 18 days post tumor inoculation. **a** The serum level of ALT in mice receiving the indicated treatment. Data represent mean \pm SEM, $n = 5$ independent animals. **b** The serum level of AST in mice receiving the indicated treatment. Data represent mean \pm SEM, $n = 5$ independent animals. **c** The serum level of LDH in mice receiving the indicated treatment. Data represent mean \pm SEM, $n = 5$ independent animals. **d** The serum level of CHE in mice receiving the indicated treatment. Data represent mean \pm SEM, $n = 5$ independent animals. **e** The serum level of UREA in mice receiving the indicated treatment. Data represent mean \pm SEM, $n = 5$ independent animals. **f** The serum level of CREA in mice receiving the indicated treatment. Data represent mean \pm SEM, $n = 5$ independent animals. ALT, alanine aminotransferase; AST, aspartate aminotransferase; LDH, lactic dehydrogenase; CHE, Cholinesterase; CREA, Creatine. QFN, Quercetin-ferrum nanoparticles; QFN₇₁₇, QFN loaded with SR-717; nCSP, normal-Cholesterol cell membrane coated QFN₇₁₇; CSP, low-Cholesterol membrane coated QFN₇₁₇; nCISP, normal-Cholesterol cell membrane coated ICB agent and QFN₇₁₇; CISP, low-Cholesterol membrane coated ICB agent and QFN₇₁₇; CIP, low-Cholesterol membrane coated ICB agent and QFN.

Supplementary Figure 52. H&E staining of major organs in mice bearing MC38 tumor. Mice bearing MC38 tumor receiving the indicated treatment were sacrificed at 18 days post tumor inoculation. Scale bar = 100 μ m. $n = 3$ independent animals. QFN, Quercetin-ferrum nanoparticles; QFN₇₁₇, QFN loaded with SR-717; nCSP, normal-Cholesterol cell membrane coated QFN₇₁₇; CSP, low-Cholesterol membrane coated QFN₇₁₇; nCISP, normal-Cholesterol cell membrane coated ICB agent and QFN₇₁₇; CISP, low-Cholesterol membrane coated ICB agent and QFN₇₁₇; CIP, low-Cholesterol membrane coated ICB agent and QFN.

Supplementary Figure 53. H&E staining of major organs in mice bearing LLC tumor. Mice bearing LLC tumor receiving the indicated treatment were sacrificed at 18 days post tumor inoculation. Scale bar = 100 μ m. $n = 3$ independent animals. QFN, Quercetin-ferrum nanoparticles; QFN₇₁₇, QFN loaded with SR-717; nCSP, normal-Cholesterol cell membrane coated QFN₇₁₇; CSP, low-Cholesterol membrane coated QFN₇₁₇; nCISP, normal-Cholesterol cell membrane coated ICB agent and QFN₇₁₇; CISP, low-Cholesterol membrane coated ICB agent and QFN₇₁₇; CIP, low-Cholesterol membrane coated ICB agent and QFN.

7. Further image analysis and quantification is necessary in Figure 3C to demonstrate that the particles trigger lysosomal degradation of PD-L1; the data as presented is not convincing enough to draw that conclusion.

Response: Thanks for your suggestion. Based on your comment, we reanalyzed the co-localization of QFN, PD-L1 and LAMP1 (Fig. 3f). We also evaluated cytotoxicity of T cells to

tumor cells *in vitro* (Fig. 2e, f). As it is still not solid enough to conclude that PD-L1 is degraded in lysosomes, we corrected the interpretation of the results. The related contents are presented below:

It appears that, consistent with *in vitro* experiments, CISP bound PD-L1 on tumor cells and brought it to lysosomes, while CSP failed to show such ability (Fig. 2e and Fig. 3f).

Fig. 3 CISP targeted to tumors *in vivo*. **f** Representative images of QFN core, PD-L1 and LAMP1 in tumors and the fluorescence intensity on the arrow line. Scale bar = 10 μm . $n = 3$ independent samples. MFI, mean fluorescence intensity; QFN, Quercetin-ferrum nanoparticles; QFN₇₁₇, QFN loaded with SR-717; nCSP, normal-Cholesterol cell membrane coated QFN₇₁₇; CSP, low-Cholesterol membrane -coated QFN₇₁₇; nCISP, normal-Cholesterol cell membrane coated ICB agent and QFN₇₁₇; CISP, low-Cholesterol membrane coated ICB agent and QFN₇₁₇; CISP (20 mM), low-Cholesterol membrane (treated with 20 mM β -CD) coated ICB agent and QFN₇₁₇; CISP (30 mM), low-Cholesterol membrane (treated with 30 mM β -CD) coated ICB agent and QFN₇₁₇.

Moreover, compared with nCSP, CISP and nCISP enhanced the killing effect of CD8 T cells against tumor cells *in vitro* (Fig. 2g, h and Supplementary Fig. 8a, b)⁷.

Fig. 2 CISP blocked the elevated PD-L1 in tumor cells induced by STING agonists *in vitro*. **g, h** Cell viability of B16F10 detected with CCK-8 kit (**g**), and images of B16F10 cells (**h**). B16F10 cells were treated with nCSP, nCISP and CISP, and then incubated with activated CD8⁺ T cells for 48 h. Scale bar = 100 μm . Data represent mean \pm SEM, $n = 4$ independent

samples. MFI, mean fluorescence intensity; QFN, Quercetin-ferrum nanoparticles; QFN₇₁₇, QFN loaded with SR-717; nCSP, normal-Cholesterol cell membrane coated QFN₇₁₇; CSP, low-Cholesterol membrane coated QFN₇₁₇; nCISP, normal-Cholesterol cell membrane coated ICB agent and QFN₇₁₇; CISP, low-Cholesterol membrane coated ICB agent and QFN₇₁₇; CIP, low-Cholesterol membrane coated ICB agent and QFN.

Supplementary Figure 8. Characterization of purified CD8⁺ T cells. **a** Flow-cytometric analysis of CD8⁺ T cells in splenocytes (Control) and purified splenocytes (Purified group). MojoSort™ Isolation Kits was used to isolate CD8⁺ T cells in the spleen of mouse Data represent mean ± SEM, *n* = 3 independent samples. **b** Representative images of CD8⁺ T cells in splenocytes (Control) and purified splenocytes (Purified group). Scale bar = 50 μm.

8. In Figure 5E, how is it known that these are tumor cells? Or is this just referring to all cells in the tumor (e.g., including cancer cells, myeloid cells, etc?)

Response: Thanks for the comment. Since CD44 is overexpressed in B16F10 melanoma tumors⁸, and the cells we identified are CD44⁺PD-L1⁺ cells in B16F10 melanoma tumors at 6 days post laser irradiation, we denoted them as tumor cells. The related data are presented in Fig. 6c in response to comment 1 and 3.

9. In vivo, the addition of PD1 didn't improve tumor accumulation/efficacy unless the cholesterol was removed. However, CSP not included as a control in the in vivo studies and so ultimately the contribution of cholesterol depletion vs PD1 expression could not be clearly resolved. The authors should repeat a subset of the experiments with this control

to determine if PD1 expression is a critical component of the design or if the improved distribution/PK properties associated with cholesterol depletion are sufficient.

Response: Thank you very much for your advice. We have repeated the experiments and CSP (denotes low-Cholesterol membrane coated STING agonist and Photothermal agent) group was added as a control group. Related text and data are copied below:

CSP (denotes low-Cholesterol membrane coated STING agonist and Photothermal agent) was fabricated and used as control as well (Fig. 1j and Supplementary Fig. 15). Results show that CSP had weaker tumor targeting capability than CISP (Fig. 3c, d). CISP could significantly increase the level of SR-717 in tumors, and CSP delivered less SR-717 to tumors (Fig. 3e). It appears that, consistent with *in vitro* experiments, CISP bound PD-L1 on tumor cells and brought it to lysosomes, while CSP failed to show such ability (Fig. 2e and Fig. 3f). These results indicate that cholesterol-deficient membrane overexpressed PD-1 could promote the tumor targeting ability of nanoparticles *in vivo*.

Fig. 3 CISP targeted to tumors *in vivo*. **c, d** Fluorescence images of tumor at 6 h post injection of nCSP, nCISP (20 mM) and nCISP (30 mM) (**c**), and the mean fluorescence intensity of DiD in tumors (**d**). Data represent mean \pm SEM, $n = 5$ independent animals. **e** Concentration of SR-717 in tumors detected by LC-MS/MS. Data represent mean \pm SEM, $n = 4$ independent animals. **f** Representative images of QFN core, PD-L1 and LAMP1 in tumors and the fluorescence intensity on the arrow line. Scale bar = 10 μ m. $n = 3$ independent samples. MFI, mean fluorescence intensity; QFN, Quercetin-ferrum nanoparticles; QFN₇₁₇, QFN loaded with SR-717; nCSP, normal-Cholesterol cell membrane coated QFN₇₁₇; CSP, low-Cholesterol membrane

-coated QFN₇₁₇; nCISP, normal-Cholesterol cell membrane coated ICB agent and QFN₇₁₇; CISP, low-Cholesterol membrane coated ICB agent and QFN₇₁₇; CISP (20 mM), low-Cholesterol membrane (treated with 20 mM β -CD) coated ICB agent and QFN₇₁₇; CISP (30 mM), low-Cholesterol membrane (treated with 30 mM β -CD) coated ICB agent and QFN₇₁₇.

Ultimately, ~85% of the mice in CISP group were cured with no tumor recurrence on day 80, which is the best outcome (Fig. 6p, q, Supplementary Fig. 34, Supplementary Fig. 35). In contrast, nCSP, CSP and nCISP displayed weaker therapeutic efficiency, with a survival rate of ~20%; and no mice survived with free SR-717 treatment, reflecting the importance of the delivery system. Noticeably, the tumor recurrence rate in CISP group was lower than CIP, again illustrated the important role of SR-717 in this system.

Fig. 6 CISP enhanced anti-tumor immunity in mice bearing melanoma. p Tumor growth curve of mice bearing melanoma. Red arrows indicate administration of nanomedicines. Data represent mean \pm SEM, $n = 8$ independent animals. **q** Survival curve of mice bearing melanoma, $n = 8$ independent animals. QFN, Quercetin-ferrum nanoparticles; QFN₇₁₇, QFN loaded with SR-717; nCSP, normal-Cholesterol cell membrane coated QFN₇₁₇; CSP, low-Cholesterol membrane coated QFN₇₁₇; nCISP, normal-Cholesterol cell membrane coated ICB agent and QFN₇₁₇; CISP, low-Cholesterol membrane coated ICB agent and QFN₇₁₇; CIP, low-Cholesterol membrane coated ICB agent and QFN.

Fluorescence images show that tumor accumulation of CISP was higher than that of nCISP, CSP and nCSP (Fig. 7b, c). Similarly, CISP and CIP also significantly reduced the expression of PD-L1 in tumor cells while nCSP, CSP and nCISP showed weaker effects (Fig. 7d and

Supplementary Fig. 36a).

Fig. 7 CISP enhanced anti-tumor immunity in mice bearing MC38 tumor. **a** Schematic illustration showing the experiment process. Red arrows indicate administration of nanomedicines. The dosage of SR-717 is 10 mg/kg. **b, c** Fluorescence images of MC38 tumors at 6 h post treated with of nCSP, CSP, nCISP and CISP (**b**) and mean fluorescence intensity (MFI) of DiD in tumors (**c**). Data represent mean \pm SEM, $n = 3$ independent animals. **d** Flow-cytometric analysis of CD44⁺PD-L1⁺ cells in MC38 tumors on day 18. Data represent mean \pm SEM, $n = 4$ independent animals. QFN, Quercetin-ferrum nanoparticles; QFN₇₁₇, QFN loaded with SR-717; nCSP, normal-Cholesterol cell membrane coated QFN₇₁₇; CSP, low-Cholesterol membrane coated QFN₇₁₇; nCISP, normal-Cholesterol cell membrane coated ICB agent and QFN₇₁₇; CISP, low-Cholesterol membrane coated ICB agent and QFN₇₁₇; CIP, low-Cholesterol membrane coated ICB agent and QFN.

Macrophage membrane was reported to exhibit inflammatory site targeting capability⁴; thus, we constructed a macrophage membrane-wrapped PLGA nanosystem. We again found that the low-cholesterol RAW264.7 cell membrane-coated PLGA (e.g. CRM) possessed better inflammation site accumulation than particles coated with normal membrane (e.g. nCRM) and naked PLGA nanoparticles in arthritis model (Supplementary Fig. 20 a-d). These results imply that the cholesterol removal strategy is also applicable to other sources of biological membranes and other disease models. Interestingly, low-cholesterol membrane sourced from CTLL2-PD1 failed to show improved targeting ability in arthritis model (Supplementary Fig. 21a, b), indicating that the functionality of the engineered membrane itself is essential, which is consistent with the results mentioned above (Fig. 3c).

Supplementary Figure 20. Targeting ability of CRM in arthritis model. **a** Representative image of ankle joints (Scale bar = 0.3 cm), H&E (Scale bar = 100 μ m) and Safranin-O and toluidine blue staining of ankle joints (Scale bar = 100 μ m). $n = 3$ independent animals. **b** Representative TEM image of nCRM and CRM. Scale bar = 50 nm. $n = 3$ independent samples. **c, d** Fluorescence images of model ankle joints at 6 h post injection of nCRM and CRM (**c**), and the fluorescence intensity of DiD in model ankle joints (**d**). Data represent mean \pm SEM, $n = 4$ independent animals. **e, f** Fluorescence images of plasma at 0.5 h post injection of particles (**e**), and the mean fluorescence intensity of DiD in the plasma (**f**). Data represent mean \pm SEM, $n = 3$ independent animals. **g, h** The uptake of nCRM and CRM by monocytes in the blood detected by flow-cytometry (**g**), and the relative fluorescence intensity of DiD in monocytes (**h**). Data represent mean \pm SEM, $n = 3$ independent animals. **i, j** Representative fluorescence images of major organs at 0.5 h post injection of nCRM and CRM (**i**), and the fluorescence intensity of DiD in the major organs (**j**). Data represent mean \pm SEM, $n = 3$ independent animals. nCRM, normal-Cholesterol cell membrane sourced from RAW264.7 cells coated PLGA nanoparticles. CRM, low-Cholesterol cell membrane sourced from RAW264.7 cells coated PLGA nanoparticles.

Supplementary Figure 21. Targeting ability of CCM in arthritis model. a, b Fluorescence images of ankle joints at 6 h post injection of nCCM and CCM (**a**), and the fluorescence intensity of DiD in ankle joints with arthritis (**b**). Data represent mean \pm SEM, $n = 3$ independent animals. nCCM, normal-Cholesterol cell membrane sourced from CTLL2-PD1 cells coated PLGA nanoparticles. CCM, low-Cholesterol cell membrane sourced from CTLL2-PD1 cells coated PLGA nanoparticles.

Minor Points:

1. In Figure 5K-L, it would be helpful to include a timeline of when PTT was applied and when particle treatments were applied in comparison.

Response: Thanks for your suggestion. We added timeline of when PTT was applied and when particle treatments to the tumor growth curve. Figures are presented below:

Fig. 6 CISP enhanced anti-tumor immunity in mice bearing melanoma. p Tumor growth curve of mice bearing melanoma. Red arrows indicate administration of nanomedicines. Data represent mean \pm SEM, $n = 8$ independent animals. **q** Survival curve of mice bearing melanoma, $n = 8$ independent animals. QFN, Quercetin-ferrum nanoparticles; QFN₇₁₇, QFN loaded with SR-717; nCSP, normal-Cholesterol cell membrane coated QFN₇₁₇; CSP, low-Cholesterol membrane coated QFN₇₁₇; nCISP, normal-Cholesterol cell membrane coated ICB agent and QFN₇₁₇; CISP, low-Cholesterol membrane coated ICB agent and QFN₇₁₇; CIP, low-Cholesterol

membrane coated ICB agent and QFN.

Fig. 7 CISP enhanced anti-tumor immunity in mice bearing MC38 tumor. l Tumor growth curve of mice bearing MC38 tumors. Red arrows indicate administration of nanomedicines. Data represent mean \pm SEM, $n = 7$ independent animals. **m** Survival curve of mice bearing MC38 tumors. $n = 7$ independent animals. QFN, Quercetin-ferrum nanoparticles; QFN₇₁₇, QFN loaded with SR-717; nCSP, normal-Cholesterol cell membrane coated QFN₇₁₇; CSP, low-Cholesterol membrane coated QFN₇₁₇; nCISP, normal-Cholesterol cell membrane coated ICB agent and QFN₇₁₇; CISP, low-Cholesterol membrane coated ICB agent and QFN₇₁₇; CIP, low-Cholesterol membrane coated ICB agent and QFN.

2. Figure 1H – the significance of this data isn't clear; this seems like something that should just be in the SI unless it serves to demonstrate expression pattern differences.

Response: Thanks for your suggestion. We moved it into the supplementary information.

Supplementary Figure 4. Coomassie brilliant blue staining of SDS-PAGE. nCSP, nCISP and CISP with 20 μ g proteins were loaded into the SDS-PAGE. QFN, Quercetin-ferrum nanoparticles; QFN₇₁₇, QFN loaded with SR-717; nCSP, normal-Cholesterol cell membrane coated QFN₇₁₇; nCISP, normal-Cholesterol cell membrane coated ICB agent and QFN₇₁₇; CISP,

low-Cholesterol membrane coated ICB agent and QFN₇₁₇.

Reviewer #3 (Remarks to the Author): with expertise in cancer nanotherapy, immuno-engineering, STING

In this manuscript, the authors develop a biomimetic nanoparticle (CISP) that contains a photothermal agent (quercetin & iron), a STING agonist (SR-717), and is coated with an immune checkpoint blockade agent (PD-1). The quercetin, iron, and SR-717 is loaded into a PLGA nanoparticle that is then coated with a biological membrane sourced from CTLL-2 T cells overexpressing PD-1. The membrane is treated with β -cyclodextrin to lower the cholesterol content which was demonstrated to lower the uptake of the nanoparticle by the monocytes in the blood by 50%. The authors show that the CISP platform delivers SR-717 and quercetin/iron to tumor site to activate local STING pathway, while the PD-1 on CISP surface blocked the up-regulated PD-L1 in tumor cells induced by STING agonist. CISP treated mice showed increased anti-tumor effects with >80% of the mice showing no sign of recurrence with the combination of PTT. Authors conclude that the removal of cholesterol could also potentially be an effective membrane modification strategy for drug delivery applications outside of cancer. The following revisions are recommended:

1) What is the encapsulation efficiency or drug loading of quercetin, iron, and SR-717 in the nanoparticle? For iron and quercetin, if this is mentioned in the previous paper that can be referenced. However, since the SR-717 is new, it would be important to include this in Figure 1 (characterization of CISP) or at least in the supplemental figures.

Response: Many thanks for your suggestion. The characterization of QFN had been done in previous study¹, to save space we did not include the contents here. In the current study, QFN was used as a carrier for SR-717. Thus, we only tested the encapsulation efficiency of SR-717. Related contents are presented below:

The encapsulation rate of SR-717 in QFN₇₁₇, nCSP, nCISP and CISP was ~90%, while it was ~70% in PLGA nanoparticles loaded with SR-717 (e.g. PLGA₇₁₇), indicating QFN performed well to load SR-717 (Supplementary Fig. 3d).

Supplementary Figure 3: Characterization of Nanoparticles. d The encapsulation efficiency (EE%) of SR-717 in PLGA₇₁₇, QFN₇₁₇, nCSP, CSP, nCISP and CISP. Data represent mean \pm SEM, $n = 3$ independent samples. QFN, Quercetin-ferrum nanoparticles; QFN₇₁₇, QFN loaded with SR-717; PLGA₇₁₇, PLGA nanoparticles loaded with SR-717; nCSP, normal-Cholesterol cell membrane coated QFN₇₁₇; CSP, low-Cholesterol membrane coated QFN₇₁₇; nCISP, normal-Cholesterol cell membrane coated ICB agent and QFN₇₁₇; CISP, low-Cholesterol membrane coated ICB agent and QFN₇₁₇; CIP, low-Cholesterol membrane coated ICB agent and QFN.

2) A legend with all the different particles (nCSP, CIP, nCISP, etc.) and what they consist of would be helpful in Figure 1 to understand how to it compares to CISP since there are many acronyms for each particle type.

Response: Thanks for your suggestion. We have made the figure accordingly.

Fig. 1 Preparation and characterization of CISP. j Schematic illustration showing nCSP, CSP, nCISP, CISP and CIP used in this study. MFI, mean fluorescence intensity; QFN, Quercetin-ferrum nanoparticles; QFN₇₁₇, QFN loaded with SR-717; nCSP, normal-Cholesterol cell membrane coated QFN₇₁₇; CSP, low-Cholesterol membrane coated QFN₇₁₇; nCISP, normal-Cholesterol cell membrane coated ICB agent and QFN₇₁₇; CISP, low-Cholesterol membrane coated ICB agent and QFN₇₁₇; CIP, low-Cholesterol membrane coated ICB agent and QFN.

3) The authors explained that while CISP and nCISP performed similarly in vitro with B16F10 melanoma tumor cells (Figure 2), CISP outperformed nCISP in vivo (Figure 3)

and attribute this difference to monocyte clearance. More explanation or mechanistic understanding of the correlation between cholesterol and monocyte clearance would be helpful for the reader to be convinced of the reasoning behind the benefit of the cholesterol removal. For instance, do monocytes phagocytose cell membrane vesicles (without PLGA) differently with or without cholesterol? Is this due to membrane fluidity or due to specific interaction of cholesterol with receptors on monocytes?

Response: Many thanks for your comments, which inspired us greatly. We did additional experiments to study the mechanisms, and found that nCISP absorbed more complement proteins (C3, C5, C1s1, CD4b, C8a, etc.) in serum. Inhibiting the functions of complement proteins using EDTA reduced the uptake of nCISP by monocytes (Fig. 5e, f). Similar patterns were also found on vesicles of cell membranes without wrapping a core nanoparticle (Supplementary Fig. 16). These results indicate that the complement proteins absorbed on biological membrane wrapped nanoparticles may enhanced their clearance by monocytes in the blood, and reduced adsorption and activation of complement system by cholesterol removal possibly impedes this clearance.

Besides, other mechanisms such as membrane fluidity of cell membrane and the interaction of cholesterol with receptors on monocytes may also involve in this process, and we will explore it in future research.

Related text and figures are copied below for your convenience.

It was reported that complement proteins deposit on the surface of nanoparticles prime them for removal by immune cells like monocytes⁹⁻¹². Indeed, we found that the adsorption of complement C3 on CISP in mice serum is positively correlated with cholesterol level in the membrane, and the content of activated complement C5a in blood was also lower (Fig. 1e, Fig. 5a, b and Supplementary Fig. 26). The proteomics results show that nCISP absorbed more complement proteins (C3, C5, C1s1, CD4b, C8a, etc.) in serum compared with CISP (Fig. 5c, d). When liposomes with different levels of cholesterol were incubated with serum of mice, higher level of cholesterol led to stronger C3 absorption in serum (Supplementary Fig. 27a, b), which is consistent with previous report¹³. These results indicate the level of cholesterol in cell membrane-coated nanoparticles may affect complement proteins absorbed on them. Thus, we used ethylenediaminetetraacetic acid (EDTA), a known inhibitor of complement¹⁴, to test the

relationship between complement protein function and blood clearance. Results show that EDTA indeed reduced the uptake of nCISP by monocytes in the blood, and more nCISP were remained in plasma (Fig. 5e-h). These results indicate that the complement proteins absorbed on biological membrane-wrapped nanoparticles may enhance their clearance by monocytes in the blood, and reduced adsorption and activation of complement system by removing cholesterol may impede this clearance¹³.

The hydrophobicity of the nanoparticle has a significant impact on the adsorption of proteins such as complements¹⁵. We found that CISP and liposomes with lower level of cholesterol have better hydrophilicity (Supplementary Fig. 27c, d and Supplementary Fig. 28a, b). However, the detailed mechanism of CISP in reducing complement adsorption is yet to be investigated in the future.

Fig. 5 CISP absorbed less complement C3 in serum. **a** Schematic illustration showing the process to detect protein corona absorbed on CISP. **b** Western blot analysis of C3 absorbed on CISP and nCISP. **c** Volcano plot showing serum proteins absorbed on CISP and nCISP. Fold_change means nCISP/CISP. $n = 4$ independent samples. **d** Heat map showing serum proteins absorbed on CISP and nCISP with significant difference (fold change > 1.2, $p < 0.05$). $n = 4$ independent samples. **e, f** Uptake of nCISP by monocytes detect by flow-cytometry (**e**) and the relative MFI of DiD in monocytes (**f**). Blood of mice was collected and incubated with EDTA (2 mM) for 20 min, and nCISP was added into blood and incubated for 30 min. **g, h** Fluorescence images of plasma (**g**) and MFI of DiD in plasma (**h**). Data represent mean \pm SEM, $n = 3$ independent samples. C3, complement 3; QFN, Quercetin-ferrum nanoparticles; QFN₇₁₇, QFN loaded with SR-717; nCISP, normal-Cholesterol cell membrane coated ICB agent and QFN₇₁₇; CISP, low-Cholesterol membrane coated ICB agent and QFN₇₁₇.

Supplementary Figure 27. Characterization of liposomes. **a** Images of coomassie brilliant blue staining of SDS-PAGE gel. Liposomes (0.5 mg/mL) with different content of cholesterol were incubated with serum of mice, and the protein absorbed on them were assessed then. **b** Western blot analysis of complement C3 absorbed on liposomes. **c** Water contact angle of Liposome 2/10 and Liposome 6/10 at 0.08s. Data represent mean \pm SEM, $n = 3$ independent samples. **d** TEM images of Liposome 2/10, Liposome 4/10 and Liposome 6/10. Scale bar = 100 nm. Lip 2/10, liposomes made from cholesterol (2 mg) and phospholipids (10 mg); Lip 4/10, liposomes made from cholesterol (4 mg) and phospholipids (10 mg); Lip 6/10, liposomes made from cholesterol (6 mg) and phospholipids (10 mg).

Supplementary Figure 28. Water contact angle of nCISP and CISP tableting on glass slices. **a, b** Representative images are shown in (a), and water contact angle of nCISP and CISP at 0.08s are shown in (b). Data represent mean \pm SEM, $n = 6$ independent samples. QFN, Quercetin-ferrous nanoparticles; QFN₇₁₇, QFN loaded with SR-717; nCISP, normal-Cholesterol cell

membrane coated ICB agent and QFN₇₁₇; CISP, low-Cholesterol membrane coated ICB agent and QFN₇₁₇.

Cell membrane vesicle isolated from CTLL2-PD1 cells treated with β -CD also showed improved tumor targeting ability, indicating that this strategy is also applicable to cell membrane vesicle and the relatively stiffer inner core is not solely responsible for their clearance (Supplementary Fig. 16a-c).

Supplementary Figure 16. Tumor targeting ability of cell membrane vesicle. a Representative TEM image of nC-vesicle (normal cell membrane vesicle sourced from CTLL2-PD1 cells) and C-vesicle (cell membrane vesicle sourced from CTLL2-PD1 cells that treated with β -CD). Scale bar = 50 nm. **b, c** Fluorescence images of B16F10 melanoma at 6 h post injection of nC-vesicle and C-vesicle (1 μ g DiD/mouse, iv.) (**b**), and the mean fluorescence intensity of DiD in tumors (**c**). Data represent mean \pm SEM, $n = 3$ independent animals. **d, e** The uptake of C-vesicle and nC-vesicle by monocytes in the blood at 0.5 h post injection detected by flow cytometry (**d**) and the relative mean fluorescence intensity (MFI) of DiD in monocytes (**e**). Data represent mean \pm SEM, $n = 3$ independent animals.

4) In addition to the studies done with tumor cells, to supplement, the uptake of CISP and nCISP in dendritic cells could be evaluated for upregulation of maturation markers.

Response: Many thanks for your suggestion. We have conducted request experiments and the results are presented below:

More CISP could be detected in dendritic cell (DCs) in tumors (Supplementary Fig. 11a, b).

Supplementary Figure 11. Flow-cytometric analysis of the uptake of nCISP and CISP by DCs in tumors. **a, b** Representative flow cytometry plot (**a**) and the proportion of DiD⁺ DCs in tumors (**b**). Mice bearing B16F10 melanoma tumors were treated with nCISP and CISP (marked with DiD, 6 µg/mouse, iv.), and cells in tumors were isolated at 6 h post injection. Cells were incubated with red blood cell lysis buffer and stained with CD11c-FITC. Data represent mean ± SEM, *n* = 3 independent animals. QFN, Quercetin-ferrum nanoparticles; QFN₇₁₇, QFN loaded with SR-717; nCISP, normal-Cholesterol cell membrane coated ICB agent and QFN₇₁₇; CISP, low-Cholesterol membrane coated ICB agent and QFN₇₁₇; DCs, dendritic cells.

Supplementary Figure 30. Maturation of DCs in B16F10 melanoma tumors. **a, b, c** Flow-cytometric analysis of CD11c⁺CD40⁺ cells (**a**), CD11c⁺CD86⁺ cells (**b**) and CD11c⁺CD80⁺ cells (**c**) in B16F10 melanoma tumors at 6 days post laser irradiation. Data represent mean ± SEM, *n* = 4 independent animals. L, laser; QFN, Quercetin-ferrum nanoparticles; QFN₇₁₇, QFN loaded with SR-717; nCSP, normal-Cholesterol cell membrane coated QFN₇₁₇; CSP, low-Cholesterol membrane coated QFN₇₁₇; nCISP, normal-Cholesterol cell membrane coated ICB agent and QFN₇₁₇; CISP, low-Cholesterol membrane coated ICB agent and QFN₇₁₇; CIP, low-Cholesterol membrane coated ICB agent and QFN.

5) Just as human blood was used to assess the uptake of CISP and nCISP by monocytes, to be consistent with the previous study, it would be good to also test the uptake of in

human tumor cell line to see if the same phenomenon holds true – that the performance of CISP and nCISP is similar when the blood flow factor is not considered.

Response: Thanks for your suggestion. Due to the structural differences between PD-1 in mouse and human T cells, CISP may not recognize PD-L1 in human cancer cells. As tumor cell membrane coated nanoparticles were reported to target the same tumor¹⁶, human melanoma cells A375 was used in new experiment. The results are presented below:

Cell membrane sourced from human melanoma cells A375 was also used here¹⁶, results show that low-cholesterol membrane-coated PLGA nanoparticles (e.g. CAM) and their normal counterpart (e.g. nCAM) had similar targeting ability to A375 cells *in vitro*, and more CAM remained in plasma after incubation with human blood (Supplementary Fig. 25f-i). Hence, the cholesterol removal strategy displayed good potential for clinical translation.

Supplementary Figure 25. Uptake of nanoparticles in human blood. f, g Representative fluorescence images of A375 cells treated with nCAM and CAM (Red) (f), and the relative MFI

of DiD in cells (g). Scale bar = 20 μ m. Data represent mean \pm SEM, $n = 5$ independent samples. **h, i** Fluorescence images of plasma at 0.5 h post incubation (h), and MFI of DiD in plasma (i). Data represent mean \pm SEM, $n = 3$ independent samples. QFN, Quercetin-ferrum nanoparticles; QFN₇₁₇, QFN loaded with SR-717; nCISP, normal-Cholesterol cell membrane coated ICB agent and QFN₇₁₇; CISP, low-Cholesterol membrane coated ICB agent and QFN₇₁₇; nCAM, normal-Cholesterol cell membrane sourced from A375 cells coated PLGA nanoparticles. CAM, low-Cholesterol cell membrane sourced from A375 cells coated PLGA nanoparticles.

6) The authors suggested that too low level of cholesterol may cause problem on membrane morphology maintenance and that studies have been done to optimize the β CD concentration. These experiments should be included in the supplemental.

Response: Many thanks for your suggestion. We have included the data which are presented below:

Additionally, high concentration of β -CD (30 mM) was found to reduce the viability of CTLL2-PD1 cells after incubation (Supplementary Fig. 17), and the encapsulation efficiency decreased to ~40% when QFN₇₁₇ were coated with membrane source from T cells treated with β -CD (30 mM) (Supplementary Fig. 1 and Supplementary Fig. 2). CISP (30 mM) failed to demonstrate the excellent tumor targeting effect of CISP (20 mM) (Fig. 3c). Hence, it appears that there is an optimal concentration of β -CD to be used in this strategy.

Supplementary Figure 17. Viability of CTLL2-PD1 cells treated with β -CD. Cells were washed with PBS once and incubated with different concentration of (2-hydroxypropyl)- β -cyclodextrin (5 mM, 10 mM, 20 mM and 30 mM) for 30 min, and then washed with PBS once.

Cell viability was detected by using CCK-8 kit. Data represent mean \pm SEM, $n = 3$ independent samples.

Supplementary Figure 1. Representative transmission electron microscope (TEM) images of nCISP, CISP (20 mM) and CISP (30 mM). $n = 3$ independent samples. QFN, Quercetin-

ferrum nanoparticles; QFN₇₁₇, QFN loaded with SR-717; nCISP, normal-Cholesterol cell membrane coated ICB agent and QFN₇₁₇; CISP (20 mM), low-Cholesterol membrane (treated with 20 mM β -CD) coated ICB agent and QFN₇₁₇; CISP (30 mM), low-Cholesterol membrane (treated with 30 mM β -CD) coated ICB agent and QFN₇₁₇.

Supplementary Figure 2. The proportion of nanoparticles coated with cell membrane detected by TEM. Data represent mean \pm SEM, $n = 3$ independent samples. QFN, Quercetin-ferrum nanoparticles; QFN₇₁₇, QFN loaded with SR-717; nCISP, normal-Cholesterol cell membrane coated ICB agent and QFN₇₁₇; CISP (20 mM), low-Cholesterol membrane (treated with 20 mM β -CD) coated ICB agent and QFN₇₁₇; CISP (30 mM), low-Cholesterol membrane (treated with 30 mM β -CD) coated ICB agent and QFN₇₁₇.

Fig. 3 CISP targeted to tumors *in vivo*. c, d Fluorescence images of tumor at 6 h post injection of nCISP, nCISP (20 mM) and nCISP (30 mM) (c), and the mean fluorescence intensity of DiD in tumors (d). Data represent mean \pm SEM, $n = 5$ independent animals. MFI, mean fluorescence intensity; QFN, Quercetin-ferrum nanoparticles; QFN₇₁₇, QFN loaded with SR-717; nCISP, normal-Cholesterol cell membrane coated QFN₇₁₇; CSP, low-Cholesterol membrane -coated QFN₇₁₇; nCISP, normal-Cholesterol cell membrane coated ICB agent and QFN₇₁₇; CISP, low-Cholesterol membrane coated ICB agent and QFN₇₁₇; CISP (20 mM), low-Cholesterol membrane (treated with 20 mM β -CD) coated ICB agent and QFN₇₁₇; CISP (30 mM), low-Cholesterol membrane (treated with 30 mM β -CD) coated ICB agent and

QFN₇₁₇.

7) Methods section: What type of PLGA was used?

Response: Thanks for your comments, we have added correspondent text.

0.67 dL/g carboxy-terminated 50:50 poly(lactic-co-glycolic acid) was obtained from Chongqing Yusi Pharmaceutical Technology in China.

8) Schematic 1 uses the acronym “COM” for the membrane coated nanoparticle, but this is never referenced in the body of the text.

Response: Thanks for spotting the mistake, we had made the corrections in the revised figure and text.

Scheme 1. **a** Schematic illustration of the preparation of CISP. **b** CISP enhanced the tumor-targeted delivery of SR-717. PD-1, programmed death-1; PD-L1, programmed cell death-Ligand 1; SR, SR-717; β -CD, (2-hydroxypropyl)- β -cyclodextrin; Que, Quercetin; PLGA, poly(lactic-co-glycolic acid); QFN₇₁₇, quercetin-ferrum nanoparticles (QFN) encapsulated SR-717; CISP, low-Cholesterol membrane coated ICB agent and QFN₇₁₇; nCISP, normal Cholesterol membrane coated ICB agent and QFN₇₁₇; C3, complement 3; CR, complement receptor; IFN-I, type I interferon.

9) It would be good to show the dose used for each injection on the dosing scheme in Figure 5.

Response: Many thanks for your suggestion, we have added the dosing information to make the figures more readable.

Fig. 6 CISP enhanced anti-tumor immunity in mice bearing melanoma. a Schematic illustration showing the experiment process. Red arrows indicate administration of nanomedicines. The dosage of SR-717 is 10 mg/kg.

Fig. 7 CISP enhanced anti-tumor immunity in mice bearing MC38 tumor. a Schematic illustration showing the experiment process. Red arrows indicate administration of nanomedicines. The dosage of SR-717 is 10 mg/kg.

Supplementary Figure 43. The therapeutic effects of CISP on mice bearing LLC tumor. c

Schematic illustration showing the experiment process. Red arrows indicate administration of nanomedicines. The dosage of SR-717 is 10 mg/kg.

10) All gating schemes for flow cytometry should be provided in the supplemental information.

Response: Many thanks for your suggestion. We have added the gating schemes in the supplemental information.

Supplementary Methods

Gating strategies

A forward/side scatter (FSC/SSC) dot plot was used to gate the main cell population, and the FSC-A/FSC-H dot plot was used to gate the single cell population. Antigen-positive cells were gated according to the cells stained with single antibody. The detail gating strategy used in this study is presented below:

Analysis of PD-1 level on the membrane of CTLL2-PD1 cells. Fig. 1b, c.

Analysis of the uptake of nanoparticles by cells. Fig. 2a, b, Supplementary Figure 18a-d.

Analysis of the uptake of nanoparticles by lymphocytes, neutrophils and monocytes in the blood. Fig. 4j-o. Fig. 5f.

Analysis of CD44⁺PD-L1⁺ cells in tumors. Fig. 6c. Fig. 7d. Supplementary Figure 43d.

Analysis of CD3⁺CD8⁺ cells in tumors. Fig. 6d. Fig. 7e. Supplementary Figure 43e

Analysis of CD69⁺ cells in CD8 T cells in tumors. Fig. 6e. Fig. 7f.

Analysis of Granzyme B⁺ cells in CD8 T cells in tumors. Fig. 6f. Fig. 7g.

Analysis of Ki67⁺ cells in CD8 T cells in tumors. Fig. 6g. Fig. 7h.

Analysis of CD3⁺CD4⁺ cells in tumors. Fig. 6h. Fig. 7i.

Analysis of Foxp3⁺ cells in CD4 T cells in tumors. Fig. 6i. Fig. 7j.

Analysis of the proportions of CD8⁺ and CD4⁺ T cells in TdLNs in mice. Supplementary Figure 40a, b.

Analysis of CD8⁺ T cells in splenocytes and the purified splenocytes. Supplementary Figure 8a.

Analysis of the uptake of nanoparticles by DCs in tumors. Supplementary Figure 12a, b.

Analysis of CD11c⁺CD40⁺ cells in tumors. Supplementary Figure 30a.

Analysis of CD11c⁺CD86⁺ cells in tumors. Supplementary Figure 30b. Supplementary Figure 37c.

Analysis of CD11c⁺CD80⁺ cells in tumors. Supplementary Figure 30c.

Analysis of CD80⁺CD86⁺ cells in CD11b⁺F4/80⁺ cells in tumors. Supplementary Figure 31a.

Supplementary Figure 37a. Supplementary Figure 43g. Supplementary Figure 47g.

Analysis of CD206⁺F4/80⁺ cells in CD11b⁺ cells in tumors. Supplementary Figure 31b.

Analysis of CD69⁺ cells in CD8 T cells TdLNs. Supplementary Figure 33a. Supplementary Figure 38c.

Analysis of CD69⁺ cells in CD4 T cells TdLNs. Supplementary Figure 33b. Supplementary Figure 38d.

Analysis of NK1.1⁺ cells tumors. Supplementary Figure 37b. Supplementary Figure 43f.

Analysis of CD11c⁺CD40⁺ cells in TdLNs. Supplementary Figure 38a.

Analysis of CD11c⁺CD86⁺ cells in TdLNs. Supplementary Figure 38b.

Analysis of CD69⁺ cells in CD4 T cells in splenocytes. Supplementary Figure 39a.

Analysis of CD69⁺ cells in CD8 T cells in splenocytes. Supplementary Figure 39b.

Analysis of Ki67⁺ cells in CD8 T cells in splenocytes. Supplementary Figure 39c.

Analysis of memory T cells in tumors. Supplementary Figure 32.

11) **Statistical analysis:** They used an unpaired two-tailed t-test was used to compare between two groups. They should consult with a biostatistician. One or two way ANOVA followed by a multiple comparison test should be used for analyzing most of their graphs.

Response: Many thanks for your suggestion, and we are sorry for making such mistake.

Statistical analysis

Statistical analysis was performed using Graphpad Prism (Version 8.0.2). All quantitative parameters were presented as mean with Standard Error of Mean. For two-group comparison, Student's two-sided *t* test was performed for the statistical analysis. For multiple comparisons, the data were analyzed using a two-way analysis of variance (ANOVA). $P < 0.05$ was considered statistically significant. ns, no significant difference.

Reviewer #4 (Remarks to the Author): with expertise in nanoparticle design, membrane coating Cell membrane coating is one of the most promising ways to improve the performance of the nanoparticles/carriers for cancer therapy. The improved functionality obtained with cell membranes are related to increased systemic circulation time by reducing the effect of MPS and homologous targeting capability. The authors utilized cell membranes from T cell with overexpressed PD-1 to coat their NPs. These NPs were effective in photothermal conversion efficiency to realize PTT and NPs were also loaded with SR-717 to induce anti-tumoral immune response through STING activation. The idea of the authors was not to add anything in the delivery system but reduce the content of the cholesterol on the cell membrane used for the NP

coating as cholesterol was previously observed to have negative effect on the targeting properties due to increased opsonization.

The authors were successful in proving their hypothesis with several and convincing results they reported in the manuscript. Regarding the results, the authors presented appropriate statistical consideration. I see the results very significant for the scientists working with cell membrane coated delivery systems giving them one good tool to improve the performance of the delivery systems.

I have some comments to improve the quality of the manuscript:

1) The authors should comments more closely what are the roles of PTT and STING activation to obtain effective antitumoral immune response i.e. good therapeutic outcome. Are they both needed?

Response: Many thanks for your comments. Immunotherapy is highly promising tumor therapy that already achieved amazing clinical outcome in many cases. STING agonists have been reported to promote immune cell activation at tumor sites to treat cancer by itself or enhance the efficacy of many immunotherapies including ICB in different ways (with detailed mechanisms still in exploration). As the activation and proliferation of immune cells need a period of time, we found it is hard to inhibit fast growing tumors like melanoma (it only takes about 5 days for the tumor to grow from 100 mm³ to 1000 mm³) by STING agonists alone. In this case, using PTT along with STING activation resulted in much better final outcome. In comparison, for MC38 tumors that grow relatively slowly, CISP inhibited its growth without need of PTT, with ~40% of them survived at last. Since different types of cancer and the exact gene type of tumor cells will largely influence the outcome likely by showing different sensitivity to STING agonists and immunotherapy, there is no clear answer and is hard to predict clinical results. Results are presented below:

Similarly, CISP also significantly reduced the expression of PD-L1 in tumor cells while nCSP, CSP and nCISP showed weaker effects (Fig. 6c and Supplementary Fig. 29a). With the combined therapeutic effects, CISP significantly increased the number of CD8 T cells, CD69⁺ CD8 T cells, Granzyme B⁺ CD8 T cells, Ki67⁺ CD8 T cells and CD4 T cells in tumors (Fig. 6d-h and Supplementary Fig. 29b, c). The maturation of DCs, M1 macrophages, memory T cells in tumors, and activated T cells in tumor draining lymph nodes (TdLNs) were also significantly

increased (Supplementary Fig. 30a-c, Supplementary Fig. 31a, Supplementary Fig. 32a-d and Supplementary Fig. 33a, b). Besides, Treg cells and M2 macrophages in tumors were decreased (Fig. 6i, Supplementary Fig. 31b), and the level of IL-12, IL-6, IFN γ , IFN β , IFN α and TNF α in tumors were elevated (Fig. 6j-o). In contrast, other groups only showed limited effect on activating anti-tumor immune response (Fig. 6d-o). It is worth noting that although CIP (denotes low-Cholesterol membrane coated ICB agent and Photothermal agent) could also reduce PD-L1 in tumors, the final outcome is unsatisfactory, indicating that SR-717 is necessary to boost the immune responses (Fig. 1j, Fig. 2f and Fig. 6c-o). Ultimately, ~85% of the mice in CISP group were cured with no tumor recurrence on day 80, which is the best outcome (Fig. 6p, q, Supplementary Fig. 34, Supplementary Fig. 35). In contrast, nCSP, CSP and nCISP displayed weaker therapeutic efficiency, with a survival rate of ~20%; and no mice survived with free SR-717 treatment, reflecting the importance of the delivery system. Noticeably, the tumor recurrence rate in CISP group was lower than CIP, again illustrated the important role of SR-717 in this system. Taken together, the CISP which utilized low-cholesterol membrane showed higher delivery performance and better therapeutic outcomes combined with PTT in mice bearing melanoma.

Fig. 6 CISP enhanced anti-tumor immunity in mice bearing melanoma. **a** Schematic illustration showing the experiment process. Red arrows indicate administration of nanomedicines. The dosage of SR-717 is 10 mg/kg. **b** Western blot analysis of p-TBK1, TBK1, p-IRF3 and IRF3 in tumors at 6 days post laser irradiation. **c** Flow-cytometric analysis of CD44⁺PD-L1⁺ cells in B16F10 melanoma tumors at 6 days post laser irradiation. Data represent mean \pm SEM, $n = 4$ independent animals. **d** Flow-cytometric analysis of CD3⁺CD8⁺ cells in B16F10 melanoma tumors. Data represent mean \pm SEM, $n = 4$ independent animals. **e** Flow-cytometric analysis of CD69⁺ cells in CD8 T cells in B16F10 melanoma tumors. Data represent mean \pm SEM, $n = 4$ independent animals. **f** Flow-cytometric analysis of Granzyme B⁺ cells in CD8 T cells in B16F10 melanoma tumors. Data represent mean \pm SEM, $n = 4$ independent animals. **g** Flow-cytometric analysis of Ki67⁺ cells in CD8 T cells in B16F10 melanoma tumors. Data represent mean \pm SEM, $n = 4$ independent animals. **h** Flow-cytometric analysis of CD3⁺CD4⁺ cells in B16F10 melanoma tumors. Data represent mean \pm SEM, $n = 4$ independent

animals. **i** Flow-cytometric analysis of Foxp3⁺ cells in CD4 T cells in B16F10 melanoma tumors. Data represent mean \pm SEM, $n = 4$ independent animals. **j, k, l, m, n, o** IL-12 (**j**), L-6 (**k**), IFN γ (**l**), IFN β (**m**), IFN α (**n**) and TNF α (**o**) at 6 days post laser irradiation. Data represent mean \pm SEM, $n = 6$ independent animals. **p** Tumor growth curve of mice bearing melanoma. Red arrows indicate administration of nanomedicines. Data represent mean \pm SEM, $n = 8$ independent animals. **q** Survival curve of mice bearing melanoma, $n = 8$ independent animals. QFN, Quercetin-ferrum nanoparticles; QFN₇₁₇, QFN loaded with SR-717; nCSP, normal-Cholesterol cell membrane coated QFN₇₁₇; CSP, low-Cholesterol membrane coated QFN₇₁₇; nCISP, normal-Cholesterol cell membrane coated ICB agent and QFN₇₁₇; CISP, low-Cholesterol membrane coated ICB agent and QFN₇₁₇; CIP, low-Cholesterol membrane coated ICB agent and QFN.

Considering that PTT is not suitable for some cancers in clinical practice, we then examined the effect of CISP on some other tumor models. Here, colon cancer cells (MC38) (5×10^5 cells/mouse) were inoculated subcutaneously in C57BL/6 mice (Fig. 7a). Fluorescence images show that tumor accumulation of CISP was higher than that of nCISP, CSP and nCSP (Fig. 7b, c). Similarly, CISP and CIP also significantly reduced the expression of PD-L1 in tumor cells while nCSP, CSP and nCISP showed weaker effects (Fig. 7d and Supplementary Fig. 36a). CISP increased CD8 T cells in MC38 tumors, and CSP also improved it to some extent, but is weaker than CISP (Fig. 7e and Supplementary Fig. 36b). More specifically, CISP treatment resulted in the strongest increase of CD8 T cells, activated CD8 T cells, CD4 T cells, M1 Macrophages, NK cells and matured DCs in MC38 tumors (Fig. 7e-i, Supplementary Fig. 36c-g and Supplementary Fig. 37a-f). Maturation of DCs, activated CD4 T cells and CD8 T cells in TdLNs (Supplementary Fig. 38a-h); as well as activated CD4 T cells and CD8 T cells in splenocytes all displayed highest increase in CISP group (Supplementary Fig. 39a-c). CISP caused significant up-regulation of the CD8⁺/CD4⁺ ratio in TdLNs, which was reported to associated with the prognosis of a variety of cancers and the response to immunotherapy² (Supplementary Fig. 40a, b). The changes in IL-12, IL-6, IFN γ , IFN β , TNF α , and Treg cells in tumors were also the greatest in CISP group (Fig. 7j, Supplementary Fig. 41a-e). In contrast, other groups including CIP and CSP showed limited activation of anti-tumor immune response (Fig. 7e-j and Supplementary Fig. 41a-e). Ultimately, CISP caused the strongest MC38 tumor suppression, and ~40% of treated mice survived on day 80, while all mice in other groups died (Fig. 7k-m and Supplementary Fig. 42). Thus, CISP is also significantly more effective than other systems in treating MC38 colon cancer, even without the introduction of PTT.

Fig. 7 CISP enhanced anti-tumor immunity in mice bearing MC38 tumor. **a** Schematic illustration showing the experiment process. Red arrows indicate administration of nanomedicines. The dosage of SR-717 is 10 mg/kg. **b, c** Fluorescence images of MC38 tumors at 6 h post treated with of nCSP, CSP, nCISP and CISP (**b**) and mean fluorescence intensity (MFI) of DiD in tumors (**c**). Data represent mean \pm SEM, $n = 3$ independent animals. **d** Flow-cytometric analysis of CD44⁺PD-L1⁺ cells in MC38 tumors on day 18. Data represent mean \pm SEM, $n = 4$ independent animals. **e** Flow-cytometric analysis of CD3⁺CD8⁺ cells in MC38 tumors. Data represent mean \pm SEM, $n = 4$ independent animals. **f** Flow-cytometric analysis of CD69⁺ cells in CD8 T cells in MC38 tumors. Data represent mean \pm SEM, $n = 4$ independent animals. **g** Flow-cytometric analysis of Granzyme B⁺ cells in CD8 T cells in MC38 tumors. Data represent mean \pm SEM, $n = 4$ independent animals. **h** Flow-cytometric analysis of Ki67⁺ cells in CD8 T cells in MC38 tumors. Data represent mean \pm SEM, $n = 4$ independent animals. **i** Flow-cytometric analysis of CD3⁺CD4⁺ cells in MC38 tumors. Data represent mean \pm SEM, $n = 4$ independent animals. **j** Flow-cytometric analysis of Foxp3⁺ cells in CD4 T cells in MC38

tumors. Data represent mean \pm SEM, $n = 4$ independent animals. **k** Images of MC38 tumors at 14 days and 24 days post tumor inoculation. $n = 7$ independent animals. Scale bar = 0.5 cm. **l** Tumor growth curve of mice bearing MC38 tumors. Red arrows indicate administration of nanomedicines. Data represent mean \pm SEM, $n = 7$ independent animals. **m** Survival curve of mice bearing MC38 tumors. $n = 7$ independent animals. QFN, Quercetin-ferrum nanoparticles; QFN₇₁₇, QFN loaded with SR-717; nCSP, normal-Cholesterol cell membrane coated QFN₇₁₇; CSP, low-Cholesterol membrane coated QFN₇₁₇; nCISP, normal-Cholesterol cell membrane coated ICB agent and QFN₇₁₇; CISP, low-Cholesterol membrane coated ICB agent and QFN₇₁₇; CIP, low-Cholesterol membrane coated ICB agent and QFN.

2) The first paragraph of Introduction is confusing. Are the authors writing about cell membrane coated NPs or other type of coatings?

Response: Thanks for your comments. We have rewritten the introduction and the first sentence of the intro is copied below.

As an important type of biomimetic drug delivery system, cell membrane coated nanoparticles have been widely studied in recent years and displayed various advantages in targeted delivery of drugs.

3) Chapter 2.5. What was the time point of laser irradiation?

Response: Thanks very much for your comments, we have remade the treatment scheme graph.

Drugs were intravenously injected on the day 8, 10 and 12, and one laser irradiation was applied to activate photothermal therapy at 12 h post the first injection (Fig. 6a)

Fig. 6 CISP enhanced anti-tumor immunity in mice bearing melanoma. a Schematic illustration showing the experiment process. Red arrows indicate administration of nanomedicines. The dosage of SR-717 is 10 mg/kg.

4) Lentivirus is indicated in Scheme 1. What is its role in the production of CTLL2-PD1?

Response: Thanks for your comments. Lentivirus, as a vector, can effectively integrate foreign genes into the host genome, and induce the stable and long-term overexpression of PD-1 in CTLL2 cells. Therefore, Lentivirus can be used to establish the PD-1 stable overexpression cell line CTLL2-PD1.

5) Figure 1A. What is n=5 referring to?

Response: Thanks for your comments. In order to detect the expression of PD-1 in cells, we used anti-mouse PD-1-APC antibody to stain CTLL2 cells and CTLL2-PD1 cells. n = 5 means 5 biological replicates per group.

6) Figure 1E. The authors should report some statistics what part of the NPs were coated with cell membranes and what is the quality of the coating (are the NPs fully or partly coated)?

Response: Thanks very much for your comments, we have revised the correspondent text and figures. The revised parts are copied below:

Transmission electron microscope (TEM) images show membrane-like structures on the surface of nCSP, nCISP, CISP (Fig. 1f and Supplementary Fig. 1), and more than 80% QFN₇₁₇ were coated with cell membrane in both nCISP and CISP (Supplementary Fig. 2).

Supplementary Figure 1. Representative transmission electron microscope (TEM) images of nCISP, CISP (20 mM) and CISP (30 mM). $n = 3$ independent samples. QFN, Quercetin-ferrum nanoparticles; QFN₇₁₇, QFN loaded with SR-717; nCISP, normal-Cholesterol cell membrane coated ICB agent and QFN₇₁₇; CISP (20 mM), low-Cholesterol membrane (treated with 20 mM β -CD) coated ICB agent and QFN₇₁₇; CISP (30 mM), low-Cholesterol membrane (treated with 30 mM β -CD) coated ICB agent and QFN₇₁₇.

Supplementary Figure 2. The proportion of nanoparticles coated with cell membrane detected by TEM. Data represent mean \pm SEM, $n = 3$ independent samples. QFN, Quercetin-ferrum nanoparticles; QFN₇₁₇, QFN loaded with SR-717; nCISP, normal-Cholesterol cell membrane coated ICB agent and QFN₇₁₇; CISP (20 mM), low-Cholesterol membrane (treated with 20 mM β -CD) coated ICB agent and QFN₇₁₇; CISP (30 mM), low-Cholesterol membrane (treated with 30 mM β -CD) coated ICB agent and QFN₇₁₇.

7) Figure 1F. It is very difficult to read the image i.e., to recognize the location of cell membrane on NPs.

Response: Many thanks for your comments. We reanalyzed the co-localization of QFN and cell membrane, and revised the image. The data is shown blow:

Fig. 1 Preparation and characterization of CISP. g The localization of cell membrane (marked with DiD) and QFN₇₁₇ core (marked with coumarin) in CISP, and the fluorescence intensity of DiD and coumarin on the arrow line. B16F10 cells were incubated with CISP for 4 h. Scale bar = 5 μ m. $n = 3$ independent samples.

8) Figure 4A is messy and difficult to understand.

Response: Many thanks for your comments. We have reprocessed the image to only display the tumor sites of the mice. Revised figure is presented below:

Fig. 4 The cholesterol-deficient cell membrane inhibited the clearance of CISP by monocytes in the blood. a, b Fluorescence images of tumors in mice treated with CISP (a) and the MFI of DiD in tumors (b). Membrane of CISP were sourced from CTLL2-PD1 that were treated with different concentration of β -CD. Data represent mean \pm SEM, $n = 3$ independent animals. MFI mean fluorescence intensity; QFN, Quercetin-ferrum nanoparticles; QFN₇₁₇, QFN loaded with SR-717; nCISP, normal-Cholesterol cell membrane coated ICB agent and QFN₇₁₇; CISP, low-Cholesterol membrane coated ICB agent and QFN₇₁₇.

Reference

1. Li, L. *et al.* Quercetin-ferrum nanoparticles enhance photothermal therapy by modulating the tumor immunosuppressive microenvironment. *Acta Biomaterialia* (2022).
2. Liang, J. *et al.* Nanoparticle-enhanced chemo-immunotherapy to trigger robust antitumor immunity. *Sci Adv* **6**, eabc3646 (2020).
3. Biermann, J. *et al.* Dissecting the treatment-naïve ecosystem of human melanoma brain metastasis. *Cell* **185**, 2591-2608 e2530 (2022).
4. Li, R. *et al.* Route to Rheumatoid Arthritis by Macrophage-Derived Microvesicle-Coated Nanoparticles. *Nano Lett* **19**, 124-134 (2019).
5. Kulkarni, A. *et al.* A designer self-assembled supramolecule amplifies macrophage immune responses against aggressive cancer. *Nat Biomed Eng* **2**, 589-599 (2018).
6. Chin, E.N. *et al.* Antitumor activity of a systemic STING-activating non-nucleotide cGAMP mimetic. *Science* **369**, 993-999 (2020).
7. Cha, J.H. *et al.* Metformin Promotes Antitumor Immunity via Endoplasmic-Reticulum-Associated Degradation of PD-L1. *Mol Cell* **71**, 606-620 e607 (2018).
8. Shi, S. *et al.* Synergistic active targeting of dually integrin α v β 3/CD44-targeted nanoparticles to B16F10 tumors located at different sites of mouse bodies. *J Control Release* **235**, 1-13 (2016).
9. Wibroe, P.P. *et al.* Bypassing adverse injection reactions to nanoparticles through shape modification and attachment to erythrocytes. *Nat Nanotechnol* **12**, 589-594 (2017).

10. Chen, F. *et al.* Complement proteins bind to nanoparticle protein corona and undergo dynamic exchange in vivo. *Nat Nanotechnol* **12**, 387-393 (2017).
11. Szebeni, J. *et al.* A porcine model of complement-mediated infusion reactions to drug carrier nanosystems and other medicines. *Adv Drug Deliv Rev* **64**, 1706-1716 (2012).
12. Moghimi, S.M., Simberg, D., Papini, E. & Farhangrazi, Z.S. Complement activation by drug carriers and particulate pharmaceuticals: Principles, challenges and opportunities. *Adv Drug Deliv Rev* **157**, 83-95 (2020).
13. Ishida, T., Funato, K., Kojima, S., Yoda, R. & Kiwada, H. Enhancing effect of cholesterol on the elimination of liposomes from circulation is mediated by complement activation. *International Journal of Pharmaceutics* **156**, 27-37 (1997).
14. Vu, V.P. *et al.* Immunoglobulin deposition on biomolecule corona determines complement opsonization efficiency of preclinical and clinical nanoparticles. *Nat Nanotechnol* **14**, 260-268 (2019).
15. Saha, K., Moyano, D.F. & Rotello, V.M. Protein coronas suppress the hemolytic activity of hydrophilic and hydrophobic nanoparticles. *Mater Horiz* **1**, 102-105 (2014).
16. Chen, Z. *et al.* Cancer Cell Membrane-Biomimetic Nanoparticles for Homologous-Targeting Dual-Modal Imaging and Photothermal Therapy. *ACS Nano* **10**, 10049-10057 (2016).

REVIEWERS' COMMENTS

Reviewer #1 (Remarks to the Author):

The authors added an extensive body of work in the revision stage to address most of the concerns, which is very impressive. However, my concern on the novelty of this work is still there since the cholesterol removal strategy has been reported and further investigation on this mechanism and demonstration of more applications were not quite significant advances, but overall, the manuscript could be considered publication.

Reviewer #2 (Remarks to the Author):

The authors have conducted a number of new experiments and have added additional controls to address my comments. Based on these new data, the manuscript is significantly improved and merits consideration for publication in Nature Communications.

Reviewer #3 (Remarks to the Author):

They have adequately addressed the previous reviews.

Reviewer #4 (Remarks to the Author):

The authors have mainly addressed the issues of the ms I raised. I have few further comments:

- Fig.6. In the figure the laser should point at the 12 h time point.
- Q6. As the sample preparation of the TEM samples might cause some artefacts (Chem. Eur. J., e202200947) and TEM is basically a qualitative and non-statistical imaging tool, it is difficult to study the overall quality of the coating in a sample. The authors should consider applying the recent method to study the integrity of the cell membrane coating of NPs (Nature Comm., 12:5726).

RESPONSE TO COMMENTS

We sincerely thank the kind comments made by the reviewers. The detailed point-to-point responses are addressed as followed:

REVIEWERS' COMMENTS

Reviewer #1 (Remarks to the Author):

The authors added an extensive body of work in the revision stage to address most of the concerns, which is very impressive. However, my concern on the novelty of this work is still there since the cholesterol removal strategy has been reported and further investigation on this mechanism and demonstration of more applications were not quite significant advances, but overall, the manuscript could be considered publication.

Reviewer #2 (Remarks to the Author):

The authors have conducted a number of new experiments and have added additional controls to address my comments. Based on these new data, the manuscript is significantly improved and merits consideration for publication in Nature Communications.

Reviewer #3 (Remarks to the Author):

They have adequately addressed the previous reviews.

Reviewer #4 (Remarks to the Author):

The authors have mainly addressed the issues of the ms I raised. I have few further comments:

1) Fig.6. In the figure the laser should point at the 12 h time point.

Response: Thanks for your suggestion. We adjusted the position of arrow so that it points to the time after 8 days, and the accurate time point for laser irradiation is shown in the figure legends. The related text and figures are copied below for your convenience:

Fig. 6 CISP enhanced anti-tumor immunity in mice bearing melanoma. **a** Schematic illustration showing the experiment process. Red arrows indicate administration of nanomedicines. The dosage of SR-717 is 10 mg/kg, and tumors were irradiated with 808nm laser for 5 min (1.0 W/cm^2) at 12 h post the first injection. **p** Tumor growth curve of mice bearing melanoma. Red arrows indicate administration of nanomedicines ($n = 8$ independent animals). Tumors were irradiated with 808nm laser at 12 h post the first injection.

2) Q6. As the sample preparation of the TEM samples might cause some artefacts (Chem. Eur. J., e202200947) and TEM is basically a qualitative and non-statistical imaging tool, it is difficult to study the overall quality of the coating in a sample. The authors should consider applying the recent method to study the integrity of the cell membrane coating of NPs (Nature Comm., 12:5726).

Response: Thanks for your suggestion. We agree that studying the coating quality of membrane-wrapped nanoparticles is a meaningful topic as it is one of the aspects of quality control and is necessary for the translation of this type of delivery systems.

This study (Nature Comm., 12:5726)¹ mentioned here provided a new method for detecting the encapsulation efficiency of cell membranes, which relies on the fluorescence quenching function of dithionite (DT) to nitro-2,1,3-benzoxadiazol-4-yl (NBD) in nanoparticles. DT is a water-soluble small molecule, which can react with NBD when there are defects or gaps in the membrane to let it pass. Therefore, if a particle is not completely wrapped by a membrane, the fluorescent will be quenched. In another word, only perfectly wrapped particles will be detected. This is reflected by the results in the article where the proportion of encapsulation never exceeded 20%. However, it is likely that partially wrapped nanoparticles are also partially functional in many cases. Thus, this criterion seems to be a bit too high at this moment. Nevertheless, to reach high portion of 100% wrapped nanoparticles is an important technique challenge and requires much more efforts in related investigation, though it is not the main purpose of current study.

Besides, as reported in the article, the author also used TEM method to identify nanoparticles with encapsulation rates ranging from 30% to 100% for experiments. Thus, TEM qualification is still a relatively classic method for identifying cell membrane encapsulation efficiency.

Based on these rationales, here we did not apply this method in our work at present. Nonetheless, as mentioned above, this wrapping issue is important and this new method is worth applying, hence we plan to put more effort in this aspect in studies that focus on formulations in the future.

The related text and figures are copied below for your convenience:

These results were consistent with previous report show that the cell membrane integrity affects the targeting ability of membrane coated nanoparticles¹, and the method established in this work may also be used to detect the membrane coating integrity on CISP in the future.

Reference

1. Liu, L. *et al.* Cell membrane coating integrity affects the internalization mechanism of biomimetic nanoparticles. *Nat Commun* **12**, 5726 (2021).